



# Dust Constraints from joint Observational-Modelling-experiMental analysis (DustCOMM): Comparison with measurements and model simulations

Adeyemi A. Adebiyi[1], Jasper F. Kok[1], Yang Wang[1], Akinori Ito[2], David A. Ridley[3], Pierre Nabat[4], Chun Zhao[5]

[1]Department of Atmospheric and Oceanic Sciences, University of California Los Angeles, CA, USA
[2]Yokohama Institute for Earth Sciences, JAMSTEC, Yokohama, Kanagawa, 236-0001, Japan
[3]Monitoring & Laboratory Division, California Air Resources Board, Sacramento, CA, USA
[4]Centre National de Recherches Météorologiques, Université de Toulouse, Météo-France, CNRS, Toulouse, France
[5]School of Earth and Space Sciences, University of Science and Technology of China, Hefei, Anhui 230026, China.

*Correspondence to*: Adeyemi A. Adebiyi (aadebiyi@ucla.edu)

**Abstract**. Mineral dust is the most abundant aerosol specie by mass in the atmosphere, and it impacts global climate, biogeochemistry, and human health. Understanding these varied impacts on the Earth system requires accurate knowledge of dust abundance, size, and optical properties, and how they vary in space and time. However, current global models show substantial biases against measurements of these dust properties. For instance, recent studies suggest that atmospheric dust is substantially coarser and more aspherical than accounted for in models, leading to persistent biases in modelled impacts of dust on the Earth system. Here, we facilitate more accurate constraints on dust impacts by developing a new dataset: Dust Constraints from joint Observational-Modelling-experiMental analysis (DustCOMM). This dataset leverages observational and experimental constraints on dust size distribution and shape to obtain more accurate constraints on three-dimensional (3-D) atmospheric dust properties than is possible from global model simulations alone. Specifically, we present annual and seasonal climatologies of the 3-D dust size distribution, 3-D dust mass extinction efficiency at 550 nm, and two-dimensional atmospheric dust loading. Comparisons with independent measurements taken over several locations, heights, and seasons show that DustCOMM estimates consistently outperform conventional global model simulations. In particular, DustCOMM achieves a substantial reduction in the bias relative to measured dust size distributions in the 0.5-20 μm diameter range. Furthermore, DustCOMM reproduces measurements of dust mass extinction efficiency to almost within the experimental uncertainties, whereas global models generally overestimate the mass extinction efficiency. DustCOMM thus provides more accurate constraints on 3-D dust properties, and as such can be used to improve global models or serve as an alternative to global model simulations in constraining dust impacts on the Earth system.

## 1. Introduction

Even though mineral dust accounts for substantial fraction of the total mass of aerosol particles in the atmosphere and produces important impacts on the Earth system, global models are unable to accurately reproduce dust abundance, size, and optical properties (Kinne et al., 2006; Huneus et al., 2011). Model difficulties in reproducing these atmospheric dust properties are largely associated with their inability to accurately simulate important dust processes, such as dust emission, transport, and deposition (e.g. Ginoux et al., 2001; Shao, 2001; Zender et al., 2003; Huneeus et



al., 2011; Kok et al., 2017). Dust aerosols are emitted from source regions such as the Sahara, the Middle-East, and Asian deserts, and are deposited after they are transported for thousands of kilometres (Duce et al., 1980; Prospero et al., 1981; Weinzierl et al., 2017). Their abundance and long-range transport allow them to play a significant role in the processes that impact global climate (Boucher et al., 2013), biogeochemistry (e.g. Mahowald et al., 2008, 2009;

Ito et al., 2019), and human health (e.g. Giannadaki et al., 2014). Specifically, dust affects global climate directly by influencing the amount of radiation that can reach or leave the atmosphere and the surface (Haywood et al., 2003; Kok et al., 2017), or indirectly by changing the amount, reflectivity and lifetime of clouds (e.g. Lohmann & Diehl, 2006; Doherty & Evan, 2014; Amiri-Farahani et al., 2017). In addition, dust also impacts global biogeochemistry through deposition of iron and phosphorous-rich micro-nutrients (Mahowald et al., 2008; 2009; Ito et al., 2019), both of which

are linked to the ability of ocean and land ecosystems to absorb atmospheric carbon dioxide (e.g. Watson et al., 2000; Blain et al., 2007). Finally, dust particles are easily inhaled by humans, with smaller dust particles penetrating deep into the lungs and leading to cardiopulmonary disease, lung cancer, and eventually death (e.g. Giannadaki et al., 2014). Therefore, obtaining accurate constraints on the many impacts of dust on the Earth system requires accurate knowledge of the sizes, abundance, and optical properties of atmospheric dust particles (Mahowald et al., 2014).

Uncertainties in dust aerosol properties directly translates into uncertainties in estimating their impact on the Earth system, such as dust radiative impacts (e.g. Huneeus et al., 2011; Zhao et al., 2013; Albani et al., 2014). Several studies have associated a large part of these uncertainties to the uncertainty in simulating the dust size distributions (e.g. Huneeus et al., 2011; Kok, 2011; Evan et al., 2014). Specifically, global models simulate too much fine-mode dust

($\sim D \leq 2.5\,\mu m$) and too little coarse-mode dust ($\sim D \geq 5\,\mu m$), both at emission and during transport in the atmosphere (e.g. Kok 2011; Kok et al., 2017). This bias is particularly problematic because fine dust cools the climate system by extinguishing shortwave (SW) radiation whereas coarse dust warms it by also extinguishing longwave (LW) radiation  (e.g. Tegen & Lacis, 1996; Dufresne et al., 2002). Whereas previous modelling studies affected by the size bias found that the combined (SW+LW) effect of dust is to cool the climate system (e.g. Tegen & Lacis, 1996;

Tegen et al., 1996; Colarco et al., 2014), it is unclear whether the dust LW warming effect may overcome the dust SW cooling effect when the underestimation of coarse-mode particles is corrected (Kok et al., 2017). Since the dust radiative effect is sensitive to the representation of size distribution in global models, constraining the dust size distribution, and how it varies spatially, is thus important.

In addition to the sensitivity of dust size distribution, dust radiative effects are also sensitive to the shape of dust particles (e.g. Kalashnikova & Sokolik, 2004). Global models generally assume that dust particles are spherical (Ginoux et al., 2001; Miller et al., 2006; Huneeus et al., 2011), even though observations suggest that they are highly non-spherical (Okada et al., 2001; Potenza et al., 2016). This idealization in the representation of dust shape in global models is used to simplify model physics (e.g. Miller et al., 2006) and the calculation of their optical properties, but

recent studies show that neglecting the asphericity of dust in models causes an underestimation of about 30% of dust aerosol optical depth (AOD) or extinction produced per unit mass of dust (Potenza et al., 2016; Kok et al., 2017). This is largely caused by the greater surface-to-volume ratio of non-spherical particles, compared to that of equal-volume spherical particles (e.g. Kalashnikova & Sokolik, 2002, 2004). The assumption of spherical dust in climate models is also problematic because the resulting underestimation of dust AOD largely masks the positive bias associated with





the fine dust particles in models, which results in an overestimation of dust AOD and extinction at remote regions when the dust emissions are scaled to match the observation of AOD near the source regions (e.g. Kok et al., 2017). Hence, to properly constrain dust impacts on radiation, observational constraints must be applied to both the dust size distribution and dust shape.

Global model simulations of the global dust cycle are thus subject to numerous important biases, which have obscured a detailed understanding of the impacts of dust on the Earth system. To address this problem, we propose a methodology to more accurately obtain 3-D dust properties than is possible from global model simulations alone. Specifically, we propose a new product called the Dust Constraints from joint Observational-Modelling-experiMental

analysis (DustCOMM), which combines an ensemble of global model simulations with observational and experimental constraints on dust size distribution and shape. The resulting product constrains the climatology of 3-D global atmospheric dust properties on seasonal and annual timescales. Below, sections 2 and 3 respectively describe the details of the methodology, as well as the data used. In section 4, we present the constrained spatial distribution of the dust size distribution, mass extinction efficiency and the atmospheric dust loading, which we evaluate using

independent *in-situ* measurements of dust size distributions and mass extinction efficiencies. Section 6 summarizes the paper. Finally, we note that all the DustCOMM dust aerosol properties (dark shaded boxes in Fig. 1) presented in this study are publicly available (Adebiyi et al., 2019a).

## 2. Methodology

Our aim is to create a new product – DustCOMM – that constrains the spatial variability of three major properties of

atmospheric dust which determine many of its impacts on the Earth system, namely (1) the atmospheric dust size distribution, (2) the dust mass extinction efficiency, and (3) the column-integrated atmospheric dust loading. We do so by combining observational, experimental and theoretical constraints on dust properties and abundance with global model simulations of the size-resolved spatially-varying dust concentration (Fig. 1). After we present a general overview of the methodology here, we describe the details of the methodology and the calculation of the associated

uncertainty estimates in the following sub-sections.

We obtain the first constrained product in our dust climatology, the dust size distribution, by bias correcting six global model simulations (section 2.1; left panel of Fig. 1). Specifically, we bias correct these model simulations using the constraint on the globally-averaged dust size distribution from Kok et al. (2017), which was obtained from

measurements of the emitted dust size distribution and model simulations of the globally-averaged dust lifetime. Model simulations of the size-resolved dust lifetimes were used because this cannot be readily constrained with observations or measurements. Similarly, we use the constraints on the globally-averaged size distribution from Kok et al. (2017) to correct modelled size distributions because dust size distribution measurements are insufficient to constrain the dust size distribution for every location. After correcting the model simulations of the dust size

distribution, we combine them into a single constraint on the 3-D dust size distribution. To do this, we estimate the sub-bin distributions by fitting the dust size distribution after the bias correction with a generalized analytical function



based on brittle fragmentation theory (Kok, 2011). We then use the resulting distributions from the multiple models to obtain a constraint on the atmospheric dust size distribution, for each horizontal location and height level.

We use these constrained size distributions to obtain our second product, namely the size-integrated 3-D dust mass extinction efficiency (section 2.2; middle panel in Fig. 1). Specifically, we combine the constrained 3-D dust size distribution with the constraint on the size-resolved globally-averaged single-particle dust extinction efficiency at 550 nm obtained from Kok et al. (2017). This size-resolved single-particle dust extinction efficiency leverages measurements of dust index of refraction and also accounts for the non-spherical shape of dust particles. As we did for the size distribution, we use the globally-averaged dust extinction efficiency here because measurements of dust shapes and index of refraction are currently insufficient to constrain this for every location. As with the size distribution, we also constrain the mass extinction efficiency over each horizontal location and height level.

We obtain our third product – the column-integrated atmospheric dust loading – by combining the constraint on dust mass extinction efficiency with dust aerosol optical depth from multiple reanalysis products (section 2.3; right panel in Fig. 1). Using four state-of-the-art reanalysis products, we calculate the ensemble average of dust aerosol optical depth, accounting for systematic and random errors. We propagate the errors in the dust mass extinction efficiency and dust aerosol optical depth to obtain the mean and the uncertainty of the column-integrated atmospheric dust loading over each horizontal location.

## 2.1 Constraining the 3-D atmospheric dust size distribution.

We constrain the spatially-varying atmospheric dust size distributions by combining constraints on the globally-averaged dust size distribution with an ensemble of simulations of the 3-D spatial variability of the dust size distribution (Fig. 1). We obtain the globally-averaged atmospheric size distribution, $\left[\frac{d\hat{V}(D)}{dD}\right]_g$, from Kok et al. (2017; see their Fig. 2a), which was obtained by combining constraints on the size distribution of emitted dust particles with simulations of the size-resolved dust lifetime. That is:

$$\left[\frac{d\hat{V}(D)}{dD}\right]_g = \left[\frac{d\hat{V}_{\text{emit}}(D)}{dD}\right]_g \cdot \left[\frac{\tilde{T}(D)}{\bar{\tilde{T}}}\right]_g \tag{1}$$

where the long-square parentheses $[\quad]_g$ indicate quantities that are globally averaged, quantities with $\hat{}$ accents are partially constrained by observations, and quantities with $\tilde{}$ accents are obtained from model simulations. The constrained globally-averaged size distribution of emitted dust particles is based on an analysis of different measurements of the emitted dust size distribution, and it is denoted here by $\left[\frac{d\hat{V}_{\text{emit}}(D)}{dD}\right]_g$. Conversely, the size-resolved globally-averaged dust lifetime is based on an ensemble of global model simulations, and is denoted by $[\tilde{T}(D)]_g$; $\left[\bar{\tilde{T}}\right]_g$ is the mass-weighted mean of $[\tilde{T}(D)]_g$. The constrained globally-averaged size distribution is normalized such that $\int_0^{D_{\max}} \left[\frac{d\hat{V}(D)}{dD}\right]_g dD = 1$. Where $D_{\max}$ represents the maximum geometric diameter above which the contribution to extinction is negligible ($D_{\max} = 20\mu m$, see section 2.1.1).



We use this constrained globally-averaged atmospheric dust size distribution (Eqn. 1) to bias-correct our spatially-varying model simulations of the annually-averaged dust size distribution. This is necessary because models generally under-estimate coarse dust particles, largely because they assume too much fine dust in the emitted dust size distribution (Kok, 2011). We thus force the annually-averaged, globally-averaged dust size distribution of each simulation in our ensemble to match the Kok et al. (2017) constraint on the globally-averaged size distribution (see supplmentart Fig. S-1). Specifically, we correct each particle bin of a model's simulated 3-D size distribution by multiplying it with a correction factor. This correction factor ($\alpha$) is defined by the ratio of the Kok et al. (2017) constraint on the fractional contribution of the particle bin to the global dust loading with the simulated fractional contribution of the particle bin to the global dust loading. That is:

$$\hat{f}_{k,i}(x,y,z,D_{k,i}) = \bar{f}_{k,i}(x,y,z,D_{k,i}) \cdot \alpha_{k,i} \tag{2}$$

$$where \quad \alpha_{k,i} = \frac{\int_{D_{k,i-}}^{D_{k,i+}} \left[\frac{d\hat{V}(D)}{dD}\right]_g dD}{\left[\bar{\bar{f}}_{k,i}(D_{k,i})\right]_g}$$

The annually-averaged 3-D distribution of the dust size distribution for each particle bin $i$ simulated by model $k$ is $\tilde{f}_{k,i}(x,y,z,D_{k,i})$, and the corresponding simulated globally-averaged dust mass fraction is $\left[\bar{\bar{f}}_{k,i}(D_{k,i})\right]_g$; Further, $D_{k,i-}$ and $D_{k,i+}$ respectively denote the lower and upper geometric diameter limits of particle bin $i$ of model $k$, and $i = 1, 2, ..., N_k$ where $N_k$ is the total number of dust particle bins for a given model simulation $k$. From Eqn. 2, we derive the resulting corrected spatially-varying dust size distribution, $\hat{f}_{k,i}(x,y,z,D_{k,i})$, where the discrete sum over each location and height equals unity, that is: $\sum_{i=1_k}^{N_k} \hat{f}_{k,i}(x,y,z,D_{k,i}) = 1$.

Each model simulation in the ensemble has a particle size range and spacing that differs from other models (see Table 1 and section 3.1 for details). In order to combine the corrected size distributions from the different models into a single estimate, and to quantify the uncertainty across the different models, each corrected size distribution must be in a consistent size range and spacing with other models. We therefore process the corrected size distributions over a given location as follows: (1) we correct and scale each model's lower and upper diameter limits to the common diameter range of 0.2 – 20 µm (see section 2.1.1); and (2) we estimate the sub-bin distribution for each model's bias-corrected size distribution by fitting a generalized analytical function, extending the Kok et al, (2017) theoretical expression of dust size distribution to the 3-D dataset (see section 2.1.2).

### 2.1.1   Correcting model simulations to a common diameter range

For all simulations in the model ensemble, we set the lower and upper diameter limits to common limits defined by $D_{\min} = 0.2$ µm and $D_{\max} = 20$ µm, respectively. The lower diameter limit ($D_{\min}$) is based on the lowest common diameter included in all the model simulations used in our analysis (Table 1). In addition, possible contaminations by other aerosol species are significantly more likely below 0.2 µm in measurements of dust aerosol particles (e.g. Dubovik et al., 2000). For these reasons, we set the lower diameter limit to $D_{\min} = 0.2$µm, consistent with previous



studies (e.g. Mahowald et al., 2014, Kok et al., 2017). The upper diameter limit ($D_{\mathrm{max}}$) represents the maximum geometric diameter above which the contribution to extinction at 550 nm is negligible (e.g. Otto et al., 2011; Kok et al., 2017). Although coarser dust could be locally important (e.g. Ryder et al., 2013), dust particles with $D > D_{max}$ generally stay only for a short period in the atmosphere before they are deposited. Moreover, measurements of dust

with $D > D_{max}$ are scarce, such that a constraint on dust particles with $D > D_{max}$ would be very uncertain (e.g. Mahowald et al., 2014).

To correct each model simulation to the common diameter range of $[D_{\mathrm{min}}, D_{\mathrm{max}}]$, we first create a new particle bin for the lower and/or upper diameter limit, and then we use the constraints on the globally-averaged size distribution

(Eqn. 1) to estimate the equivalent fraction of dust mass in that bin. This dust mass fraction is estimated in a way that is consistent with the size distribution obtained earlier from Eqn. 2. Specifically, for simulations with a lower diameter limit ($D_{k,1_{k^-}}$) less than $D_{\mathrm{min}}$, we estimate the equivalent dust mass fraction for the bin between $D_{\mathrm{min}}$ and $D_{k,1_{k^+}}$ (where $D_{k,1_{k^+}}$ is the upper diameter limit of bin 1; such that $D_{k,1_{k^+}} > D_{\mathrm{min}}$ ) by scaling the mass in the nearest bin with a factor that depends on the globally-averaged size distribution. For instance, the first particle bin of the CESM

model (Table 1) has a range of $[D_{k,1_{k^-}}, D_{k,1_{k^+}}] = 0.1 - 1.0$ µm, such that we create a new particle bin defined by $[D_{\mathrm{min}}, D_{k,1_{k^+}}] = 0.2 - 1.0$ µm, and estimate the equivalent dust mass fraction in that new bin. For all model simulations, we can denote this procedure mathematically as:

$$\hat{f}_k\big(x, y, z, [D_{\mathrm{min}}, D_{k,1_{k^+}}]\big) \; = \; \hat{f}_k\big(x, y, z, [D_{k,1_{k^-}}, D_{k,1_{k^+}}]\big) \cdot \delta_{D_{\mathrm{min}}} \tag{3}$$

$$where \quad \delta_{D_{\mathrm{min}}} = \frac{\int_{D_{\mathrm{min}}}^{D_{k,1_{k^+}}} \left[\dfrac{d\hat{V}(D)}{dD}\right]_g dD}{\int_{D_{k,1_{k^-}}}^{D_{k,1_{k^+}}} \left[\dfrac{d\hat{V}(D)}{dD}\right]_g dD}$$

The modelled dust size distribution is relatively invariant for fine particles because of the consistent emitted dust size distribution (Kok, 2011a & b), and because removal processes for fine dust (wet deposition) do not strongly depend on particle size (e.g. Zender et al., 2003). Therefore, we simply estimate $\delta_{D_{\mathrm{min}}}$ in Eqn. 3 as the ratio between the fractional values of the globally-averaged size distribution in the desired new bin $[D_{\mathrm{min}}, D_{k,1_{k^+}}]$ and in the model's original bin $[D_{k,1_{k^-}}, D_{k,1_{k^+}}]$.

We also create a new bin with the upper diameter equal to $D_{\mathrm{max}}$ for model simulations with an upper diameter limit ($D_{k,N_{k^+}}$) that differs from $D_{\mathrm{max}}$. We do so by scaling the nearest bin by a factor ($\delta_{D_{\mathrm{max}}}$) that also depends, in part, on the globally-averaged size distribution. Because the main removal process for large dust particles ($D > 10\mu m$) is dry deposition, which depends strongly on particle size, the relative contribution to the size distribution of different particle

bins of large particles has substantial spatial variability. To account for this, we use simulations of bins with $D > D_{k,N_{k^+}}$ from other model simulations in order to estimate what model $k$ would have predicted for a hypothetical $[D_{k,N_{k^+}}, D_{\mathrm{max}}]$ particle bin. That is:

$$\hat{f}_k\big(x, y, z, [D_{k,N_{k^+}}, D_{\mathrm{max}}]\big) \; = \; \hat{f}_k\big(x, y, z, [D_{k,N_{k^-}}, D_{k,N_{k^+}}]\big) \cdot \delta_{D_{\mathrm{max}}}(x, y, z) \tag{4a}$$


$$where \quad \delta_{D_{\max}}(x,y,z) = \frac{\int_{D_{k,N_k+}}^{D_{\max}} \left[\frac{d\hat{V}(D)}{dD}\right]_g dD}{\int_{D_{k,N_k-}}^{D_{k,N_k+}} \left[\frac{d\hat{V}(D)}{dD}\right]_g dD} \cdot \beta_r(x,y,z)$$

The factor $\beta_r$ thus quantifies the ratio of the mass fractions between the model's largest particle bin ($\left[D_{k,N_k-}, D_{k,N_k+}\right]$) and the newly created particle bin to extend the simulation to $D_{\max} = 20\ \mu m$ ($\left[D_{k,N_k+}, D_{\max}\right]$), as estimated from the GISS and ARPEGE-Climat simulations, which have particle bins extending to $D_{\max}$ (Table 1). We denote these latter

model simulations with a subscript $r$ for the purpose of clarity, and to separate them from the model simulation that is being adjusted to the $[D_{\min}, D_{\max}]$ size range, which is denoted by a subscript $k$ in Eqn. 4a above. We thus estimate $\beta_r$ as:

$$\beta_r(x,y,z) = \frac{\hat{f}_r\left(x,y,z,\left[D_{r,N_r-}, D_{r,N_r+}\right]\right)}{\int_{D_{r,N_r-}}^{D_{r,N_r+}} \left[\frac{d\hat{V}(D)}{dD}\right]_g dD} \Bigg/ \frac{\hat{f}_r\left(x,y,z,\left[D_{r,j_r-}, D_{r,j_r+}\right]\right)}{\int_{D_{r,j_r-}}^{D_{r,j_r+}} \left[\frac{d\hat{V}(D)}{dD}\right]_g dD} \qquad (4b)$$

Where $\left[D_{r,N_r-}, D_{r,N_r+}\right]$ is the bin in model $r$ with dust mass that overlaps in size with the new bin $\left[D_{k,N_k+}, D_{\max}\right]$ we

want to estimate for model $k$; and $\left[D_{r,j_r-}, D_{r,j_r+}\right]$ is the bin that similarly overlaps with $\left[D_{k,N_k-}, D_{k,N_k+}\right]$. To account for the bin-range mismatch between the model simulation that resolved dust up to $D_{\max}$ (with subscript $r$) and the model simulation being adjusted to the dust size range up to $D_{\max}$ (with subscript $k$), we normalize each bin mass fraction by its contribution to the globally-averaged size distribution. For cases where model $r$ is the same as model $k$ (i.e. for GISS and ARPEGE-Climat), $\beta_r$ reduces to one everywhere.

### 2.1.2    Estimating the sub-bin distribution of the dust size distribution

After setting the corrected dust size distribution from each model to a common diameter range, $[D_{\min}, D_{\max}]$, we next estimate the sub-bin distribution in order to combine estimates from different models into one dust size distribution

product. To do this, we fit a generalized theoretical function of the dust size distribution to the estimated bias-corrected dust size distribution from each model over each location and height level (Eqn. 2). We describe in this section this generalized function and the fitting procedure.

We define the generalized function for the atmospheric size distribution by considering the theoretical expressions

that characterize the processes affecting the dust size distribution. The degree of the impact of any of these processes on the dust size distribution will depend on the location. For example, the impact of emission processes on the shape of the dust size distribution is expected to be large close to major dust source regions but less farther from source regions. Furthermore, farther from dust source regions, deposition processes are expected to have more impact on the size distribution. We therefore assume that the atmospheric size distribution over any location is proportional to the

dust size distribution at emission, $\frac{dV_{emit}(D)}{dD}$, the size-resolved dust lifetime in the atmosphere, $T(D)$, and any other changes to the dust particle size distribution during transport, A(D) (e.g. Weinzierl et al., 2009; Schladitz et al., 2011; Kok et al., 2017). That is:





$$\frac{dV_{atm}(D)}{dD} \propto \frac{dV_{emit}(D)}{dD} \cdot T(D) \cdot A(D) \tag{5a}$$

For the dust size distribution at emission, Kok (2011) suggested that $\frac{dV_{emit}(D)}{dD}$ can be represented by a simple theoretical expression based on brittle fragmentation theory, which shows good agreement with measurements (e.g. Mahowald et al., 2014; Rosenberg et al., 2014). To better represent the variability in dust emission affecting the emitted size distribution in the different simulations, here we generalize this expression such that:

$$\frac{dV_{emit}(D)}{dD} = \frac{1}{C_v} \cdot \left[1 + \mathrm{erf}\left(\frac{\ln\left(\frac{D}{D_s}\right)}{\sqrt{2}\ln(\sigma_s)}\right)\right] \cdot e^{\left[-\left(\frac{D}{\omega}\right)^{\alpha}\right]} \tag{5b}$$

Where $D_s$ and $\sigma_s$ are respectively the geometric median diameter by volume and the geometric standard deviation of a typical desert soil, $\omega$ denotes the propagation distance of main cracks in dust aggregates during fragmentation, $\alpha$ is a tunable parameter primarily affecting the large dust particles, and $C_v$ is a normalization constant.

The second term in our generalized dust size distribution describes the size-resolved dust lifetime, which global model results compiled in Kok et al. (2017) suggest can analytically be approximated as an exponential function of particle diameter, such that:

$$T(D) \cong T_0 \cdot e^{-\left(\frac{D}{\kappa}\right)} \tag{5c}$$

Where $T_0$ is a constant associated with the lifetime for vanishingly small dust particles, which is determined by depositional processes, and $\kappa$ is a constant that scales the exponential decay of the dust lifetime with particle size. This exponential decay of dust lifetime with size is caused by the increase of the gravitational settling speed with particle size (e.g. van der Does et al., 2016, 2018).

Finally, we account for other changes to the dust size distribution during transport, by assuming that such changes are likely described by power law distribution (e.g. Seinfeld & Pandis, 2016). Maring et al. (2003) highlighted that between emission and deposition, changes in dust size distribution cannot be accounted for by simple preferential removal of dust particles by gravitational settling. Since such changes in the dust size distribution are difficult to account for, we represent them with a parameter that can affect the entire size range. In addition, $T(D)$ and $\frac{dV_{emit}(D)}{dD}$ represent expressions that describe the globally-averaged size distributions, and applying them to specific location requires additional parameter that captures the loss rate as a function of location. To represent all other changes to the dust size distribution between emission and deposition, we thus define:

$$A(D) \propto D^b \tag{5d}$$

Combining Eqns. 5b—5d, we obtain:

$$\frac{dV_{atm}(D)}{dD} = \frac{1}{C_v}\left[1 + \mathrm{erf}\left(\frac{\ln\left(\frac{D}{D_s}\right)}{\sqrt{2}\ln(\sigma_s)}\right)\right] e^{\left[-\left(\frac{D}{\omega}\right)^{\alpha}\right]} \cdot T_0 e^{-\left(\frac{D}{\kappa}\right)} \cdot D^b \tag{5e}$$

We combine the two exponential terms in Eqn. 5e in order to reduce the number of fitting parameters. It is worth noting that both parameters $\kappa$ and $\omega$ are sensitive to the larger particles, as they remain highly uncertain and poorly





constrained by observation (e.g. Mahowald et al., 2014). The parameter $\omega$ depends on the soil moisture, mineralogy and other processes (e.g. Mahowald et al., 2014; Rosenberg et al., 2014; Kok et al., 2017), while the parameter $\kappa$ depends on the dust wet and dry deposition rates, as the dust particles are transported away from the source (e.g. Han & Zender, 2010; van der Does et al., 2016). To combine them, we define $\Lambda$ to account for the uncertainty in the atmospheric large-size dust particles over every location. The generalized theoretical function for atmospheric size distribution therefore becomes:

$$\frac{dV_{atm}(D)}{dD} = \frac{1}{C_v^*} \cdot \left[ 1 + \mathrm{erf}\left( \frac{\ln\left(\frac{D}{D_s}\right)}{\sqrt{2}\ln(\sigma_s)} \right) \right] \cdot e^{\left[ -\left(\frac{D}{\Lambda}\right)^\alpha \right]} \cdot D^b \tag{6}$$

where $C_v^*$ is a new normalization constant that is obtained from requiring that the integral over Eqn. (6) from $D_{min}$ to $D_{max}$ yields unity.

For each height, horizontal location, season, and model simulation, we determine the parameters in Eqn. 6 by fitting the generalized size distribution to the corrected dust size distribution from Eqn. 2 above. We do so by minimizing the chi-squared ($\chi_k^2$) value for each height, location, and for each model $k$, such that:

$$\chi_k^2 = \sum_i^{N_k} \left[ \log\left( \int_{D_{k,i-}}^{D_{k,i+}} \frac{dV_{atm}}{dD} dD \right) - \log(\hat{f}_{k,i}) \right]^2 \tag{7}$$

In each case, we estimate the constrained dust size distribution, $\frac{d\hat{V}_{\mathrm{atm}}}{dD}$, based on the parameters we determine from Eqn. 7. In order to restrict the fitted function to physically realistic dust size distributions, we set the following bounds for the five parameters of Eqn. 6: $D_s = 0.25 - 6.0 \ \mu m$; $\sigma_s = 1.6 - 4.0$; $\Lambda = 1 - 30 \ \mu m$; $\alpha = 1 - 6$; and $b = -10 - 4$, consistent with previous studies (e.g. Kok, 2011; Kok et al., 2017; Ryder et al., 2013; Rosenberg et al., 2014; See supplementary Fig. S-2 for the probabilty distribution of each parameter).

## 2.2 Constraining the 3-D dust mass extinction efficiency

After obtaining the constrained atmospheric dust size distributions (section 2.1 above), we combine it with constraints on size-resolved single-particle extinction efficiency at 550 nm, to obtain constraints on the 3-D dust mass extinction efficiency ($\hat{\epsilon}_\tau$). That is (see also Kok et al., 2017):

$$\hat{\epsilon}_\tau(x, y, z) = \int_{D_{min}}^{D_{\max}} \frac{d\hat{V}_{\mathrm{atm}}(x, y, z, D)}{dD} \frac{3}{2\rho_d D} \hat{Q}_{\mathrm{ext}}(D) dD \tag{8}$$

where $\frac{d\hat{V}_{\mathrm{atm}}(x,y,z,D)}{dD}$ is the constrained atmospheric dust size distribution at a given location and height with sub-bin distribution (Eqn. 6); $\rho_d = 2.5 \pm 0.2$ g cm$^{-3}$ is the globally-averaged density of dust aerosols ( Fratini et al., 2007; Reid et al., 2008; Kaaden et al., 2009; Sow et al., 2009; Kok et al., 2017) and its error range is expected to account for the spatial and temporal variability of dust density (e.g. Tegen & Fung, 1994; Li et al., 2008); and $\hat{Q}_{\mathrm{ext}}(D)$ is the globally-averaged size-resolved single-particle extinction efficiency, with the extinction cross-section normalized by $\pi D^2/4$ – the projected area of a sphere with diameter $D$.



We obtain $\hat{Q}_{\text{ext}}$ from Kok et al. (2017), which constrained the dust extinction efficiency by combining measurements of the dust index of refraction and probability distribution of dust particle shape with the single-scattering database of Meng et al., (2010). Specifically, Kok et al. (2017) estimated the globally-averaged values of the real and imaginary dust index of refraction as $n = 1.53 \pm 0.03$ and $\log(-k) = -2.5 \pm 0.3$ (Sokolik et al., 1993; Patterson et al., 1977;

Dubovik et al., 2002; K. Kandler et al., 2009; Kim et al., 2011; Denjean et al., 2016), and both are assumed to be normally distributed. Dust particle shapes were represented by the dust aspect ratio – the ratio of the major and minor axes of an ellipsoid best fit to the irregularly-shaped 2-D image of a dust particle – and the height-to-width ratio. Kandler et al., (2007) showed that the deviation of measured dust aspect ratios from a sphere can be approximated by a log-normal distribution, with typical values ranging from 1 – a perfect sphere – to about 3, and median between

~1.5—1.9. Based on aggregates of measurements (Okada et al., 2001; Reid et al., 2003; Kandler et al., 2007; Chou et al., 2008; Kandler et al., 2009, 2011; Scheuvens et al., 2011; Scheuvens & Kandler, 2014), Kok et al. (2017) estimated the median and geometric standard deviation for the distribution of the dust aspect ratio as – $1.7 \pm 0.3$ and $0.6 \pm 0.2$, respectively. Similarly, based on limited available measurements of dust height-to-width ratio (Okada et al., 2001; Chou et al., 2008; Veghte & Freedman, 2014), Kok et al. (2017) used a mean value of 0.333 (see details in the

supplementary document of Kok et al., 2017). By combining these constraints on the optical properties and shape of the ensemble of dust particles in Earth's atmosphere with the single-scattering database of Meng et al. (2010), Kok et al. (2017) obtained a constraint on the globally-averaged size-resolved extinction efficiency $\hat{Q}_{\text{ext}}(D)$, which explicitly accounts for the enhancement of extinction by the asphericity of dust. Specifically, they found that accounting for dust asphericity enhances the extinction produced by a unit mass loading of dust by $29 \pm 5\%$ over the extinction calculated

from Mie theory for spherical dust particles, which is used in most climate models. We use this constrained globally-averaged $\hat{Q}_{\text{ext}}$ to constrain $\hat{\epsilon}_\tau$ (Eqn. 8) for every location because measurements of dust shapes and index of refraction are currently insufficient to constrain $\hat{\epsilon}_\tau$ on a regional basis.

### 2.3 Constraining the 2-D atmospheric dust loading

We now combine the above-estimated mass extinction efficiency at 550 nm (section 2.2) with dust aerosol optical depth at the same wavelength, to constrain the atmospheric dust loading ($\hat{L}$). That is:

$$\hat{L}(x, y) = \frac{\hat{\tau}_d(x, y)}{\hat{\epsilon}_m(x, y)} \tag{9}$$

where $\hat{\epsilon}_m(x, y)$ is the mass-weighted vertically-integrated 2-D mass extinction efficiency calculated from $\hat{\epsilon}_\tau$, and

$\hat{\tau}_d(x, y)$ is obtained from an ensemble of reanalysis dust aerosol optical depth products.

The ensemble dust aerosol optical depth (AOD) climatology is obtained from the average of four different reanalysis products (see section 3.2 for details). This individual reanalysis dataset assimilate several satellite and ground-based measurements from multiple platforms, including MODIS (*Terra* and *Aqua*), AVHRR and MISR satellites, as well as

from the ground-based AERONET stations (Lynch et al., 2016; Mccarty et al., 2016; Flemming et al., 2017; Yumimoto et al., 2017). As such the assimilation procedure takes advantage of the best features in both the





observations and model simulations, thus producing column-integrated dust AOD that is largely representative of what is observed, based on validation studies (e.g. Buchard et al., 2017).

Despite the advantage of assimilating observational datasets, estimating a realistic overall error in the dust AOD across
the reanalysis datasets is difficult, yet important. Here, we estimate the total error ($\sigma_d$) by considering both the systematic error ($\sigma_{sys}$) and random error ($\sigma_{rand}$) inherent in the reanalysis-derived dust AOD. As such, we estimate the uncertainty in dust AOD as: $\sigma_d(x,y) = \sqrt{\sigma_{sys}^2(x,y) + \sigma_{rand}^2(x,y)}$. We define the $\sigma_{rand}$ as the standard error between the four datasets, which represents that part of the total uncertainty that does not correlate across the four reanalysis dust AOD data sets. For instance, $\sigma_{rand}$ may be associated with differences in the assimilating systems for
the different reanalysis products. In contrast, the $\sigma_{sys}$ is expected to correlate between the four data sets since most of the reanalysis datasets use similar observational datasets. Hence, we assume that the $\sigma_{sys}$ will be proportional to the mean dust AOD, such that $\frac{\sigma_{sys}(x,y)}{\tau_d(x,y)} = C_d$. We estimate the proportionality constant, $C_d$, by requiring that the relative error is the same as the relative error obtained from annually-averaged climatology of dust AOD from Ridley et al. (2016), which leveraged observational datasets similar to those used for the reanalysis dataset, but propagated many of the relevant uncertainties. From that, we deduce that $C_d = \sqrt{\frac{\sigma_o^2}{\tau_o^2} - \frac{\sigma_{rand}^2}{\tau_d^2}}$, where $\tau_o$ and $\sigma_o$ are the mean and error
estimates of the observationally-constrained dust AOD from Ridley et al. (2016) respectively. We estimate $C_d = 0.26$ for annual climatology, averaged over regions that are constrained by Ridley et al. (2016) and account for about 95% of the global dust AOD. Similarly, we estimate 0.31, 0.22, 0.24, 0.28 for December-February, March-May, June-August, and September-November seasonal climatologies, respectively.

## 2.4 Quantifying the uncertainties in DustCOMM products

For each DustCOMM product above – the constrained dust size distribution, dust mass extinction efficiency and dust atmospheric loading - we describe here how we estimate the most likely value and quantify the uncertainty over each
location. Specifically, we use a non-parametric procedure based on the bootstrap method (Efron & Gong, 1983; Chernick, 2007). We use this method because the complexity of the equations (Eqns. 1-9) prevents a parametric quantification of error, and the bootstrap approach allows us to nonetheless propagate the uncertainty in the various physical variables used to estimate each product. Using this method, we further assume that the set of input variables in relevant equations above are independent, and are represented by defined probability distributions. Thus, we
estimate the probability distribution of the resulting products by randomly sampling (with replacement) the probability distribution of each of the input variables for a large number of times ($n \approx 1,500$).

In practice, the procedure uses the following steps to determine the dust size distribution, mass extinction efficiency and atmospheric loading, and their uncertainties:





1. We randomly-select a realization of the globally-averaged size distribution from Kok et al. (2017), which in turn was obtained in that study by randomly-selecting a realization of the emitted dust size distribution and the dust lifetime (Eqn. 1).

2. We use this randomly-selected constrained globally-averaged size distribution to correct a randomly-selected model simulation (Eqn. 2).

3. After this model simulation is corrected, we then scale the resulting 3-D dust size distribution between $D_{\min}$ and $D_{\max}$ following Eqns. 3 & 4.

4. We thereafter estimate the constrained dust size distribution, $\frac{d\hat{V}_{\text{atm}}}{dD}(x, y, z, D)$, and obtain the sub-bin distribution by fitting the generalized theoretical expression (Eqn. 6) and minimizing the chi-square over each location (Eqn. 7).

5. We randomly select a realization of the globally-averaged size-resolved single-particle extinction efficiency, $\hat{Q}_{\text{ext}}(D)$, from Kok et al. (2017). This realization is also similarly estimated by randomly-selecting from the distribution of the dust index of refraction and dust shape distribution parameters, as explained in section 2.2.

6. We then use the randomly-selected $\hat{Q}_{\text{ext}}(D)$ and $\frac{d\hat{V}_{\text{atm}}}{dD}(x, y, z, D)$ to estimate the dust mass extinction efficiency over each location, $\hat{\epsilon}_\tau(x, y, z, D)$, following Eqn. 8. This uses a randomly-selected dust density value ($\rho_d$) from its assumed normal distribution.

7. Similarly, assuming a normal distribution for the dust AOD, we randomly estimate the $\hat{\tau}_d(x, y)$ value within the range of its uncertainty, $\sigma_d(x, y)$.

8. We use this $\hat{\tau}_d(x, y)$ and the vertically-integrated value of dust mass extinction efficiency, $\hat{\epsilon}_m(x, y)$, to estimate the atmospheric dust loading, $\hat{L}(x, y)$, following Eqn. 9.

9. We repeated step 1-8 for $n$ = 1500 times, thereby producing a probability distribution for $\frac{d\hat{V}_{\text{atm}}}{dD}(x, y, z, D)$, $\hat{\epsilon}_\tau(x, y, z, D)$, and $\hat{L}(x, y)$ for each location and height. We report the mean, median, 1-sigma uncertainty range (68% of the distribution), and the 95% confidence interval (95% CI) of those distributions (Adebiyi et al., 2019a).

The above procedure propagates various uncertainties in the estimation of each product. These include the measurement uncertainties and the uncertainties in model simulations. First, the measurement uncertainties are associated with the globally-averaged size distribution and the globally-averaged extinction efficiency (Fig. 1), and these are propagated equally to every location. In addition, we estimated the correlated systematic error in the dust AOD (section 2.3), associated with the assimilated observational dataset, and this is also propagated. Second, the uncertainty in model simulations is associated with the spread of the model dust size distribution which is different for every location. This model uncertainty is, in turn, a result of many processes, such as dust emission, deposition, and transport processes in the models (Ginoux et al., 2001; Huneeus et al., 2011; Zhao et al., 2013). Our procedure constrains these model uncertainties (see supplementary Fig. S-1), while retaining the spatial distribution of the model ensemble.





To quantify the size-resolved discrepancies in the DustCOMM size distribution, we quantify the bias with respect to independent measurements as follows (e.g. Lee et al., 2009):

$$\psi_i^{bias} = \frac{1}{N_m} \sum_{m=1}^{N_m} log_{10}\left(\frac{M_{i,m}^f}{O_{i,m}^f}\right) \tag{10}$$

where $m$ sums over the $N_m$ *in-situ* measurements of the dust size distribution available in the literature (see Table 2), $O_{i,m}^f$ is the $mth$ measurement of the mass fraction contained in measurement bin $i$, and $M_{i,m}^f$ is the corresponding $mth$ DustCOMM dust mass fraction for the same diameter range as measured and collocated with the measurement – i.e. $M_{i,m}^f = \int_{D_{i-}}^{D_{i+}} \frac{d\hat{V}_{atm}}{dD} dD$. $\psi_i^{bias}$ is the log-mean normalized bias and it represents the average number of orders of magnitude bias for each bin $i$.

We also estimate the performance of DustCOMM mass extinction efficiency by quantifying the reduced chi-square ($\chi_\varepsilon^2$) defined as the chi-squared per degree of freedom (e.g. Bevington et al., 1993):

$$\chi_\varepsilon^2 = \frac{1}{v_\varepsilon} \sum_{m=1}^{N_m} \left(\frac{O_m^\varepsilon - \hat{\epsilon}_{\tau,m}}{\sigma_m^\varepsilon}\right) \tag{11}$$

where $O_m^\varepsilon$ is the $mth$ measurement of the dust mass extinction efficiency with error defined as $\sigma_m^\varepsilon$, $\hat{\epsilon}_{\tau,m}$ is the corresponding $mth$ DustCOMM dust extinction efficiency ($\hat{\epsilon}_\tau$) collocated with the measurement, and $v_\varepsilon$ is the number of degrees of freedom given as $N_m - 1$. A value of $\chi_\varepsilon^2 \approx 1$ in Eqn. 11 indicates there is agreement between DustCOMM and observations that is in accordance with the measurement errors, while $\chi_\varepsilon^2 \gg 1$ indicates that DustCOMM estimates do not fully capture the observations (e.g. Andrae et al., 2010).

To facilitate comparison between DustCOMM and model evaluations, Eqns. 10 & 11 are also used to evaluate the performance and calculate the discrepancies between the measurements and the model ensemble.

## 2.5 DustCOMM at other timescales

While we describe above the procedure that constrains the annually-averaged dust size distribution, dust mass extinction efficiency and atmospheric dust loading, a similar procedure as highlighted above can also be used to constrain the three products at any other timescale, such as at daily, monthly, or seasonal timescale. For this study, we only consider the seasonally-averaged and annually-averaged products.

First, to constrain the dust size distribution at any specific timescales, we correct an ensemble of model size distributions at that timescale in a way similar to Eqn. 2 above. However, unlike Eqn. 2 that uses the constrained globally-averaged size distribution, here we use the constrained annually-averaged dust size distribution over every location. That is:

$$\hat{f}^t{}_{k,i}(x,y,z,D_{k,i}) = \tilde{f}^t{}_{k,i}(x,y,z,D_{k,i}) \cdot \frac{\int_{D_{k,i-}}^{D_{k,i+}} \frac{d\hat{V}_{atm}}{dD}(x,y,z,D)\, dD}{\tilde{f}_{k,i}(x,y,z,D_{k,i})} \tag{12}$$





where $\frac{d\widehat{V}_{\text{atm}}}{dD}$ is the DustCOMM annually-averaged dust size distribution at a given 3-D location, obtained from the procedure described in Section 2.1, while $\tilde{f}$ and $\tilde{f}^t$ are the annually-averaged and specific time-averaged model simulations of the dust size distribution respectively. Using an ensemble of model simulations, as we do above, the

resulting corrected time-averaged dust size distributions, $\hat{f}^t$, are also taken through the steps highlighted in section 2.4 to calculate the mean and the uncertainty of the constrained dust size distribution ($\frac{dV_{atm}^t}{dD}$) at that particular timescale.

Second, to constrain the dust mass extinction at any specific timescale ($\hat{\epsilon}^t{}_\tau$), we combine the constrained dust size

distribution at that timescale, $\frac{dV_{atm}^t}{dD}$, with the globally-averaged extinction efficiency, $\widehat{Q}_{\text{ext}}$. This similarly follows Eqn. 8 above. We note here that the uncertainty range of $\widehat{Q}_{\text{ext}}$ also accommodates the location-dependent and time-dependent variability in the dust index of refraction and dust particle shape, consistent with previous studies (e.g. Dubovik et al., 2002). Hence, using $\widehat{Q}_{\text{ext}}$ propagates the uncertainty in the measurements that determine the dust mass extinction efficiency estimate at that timescale. Finally, we constrain the dust loading at any specific timescale ($\widehat{L}^t$)

using the constrained $\hat{\epsilon}^t{}_\tau$ and dust AOD at that same timescale, similarly following Eqn. 9.

## 3     Data and Models

The methodology described above highlighted measurements and model simulations, and reanalysis datasets used as

inputs for the DustCOMM products. Here we will describe these datasets, the model simulations, as well as other independent datasets we use to evaluate our products. To create the DustCOMM products (Fig. 1), we use three sets of input datasets: (1) the constrained globally-averaged data from Kok et al., (2017); (2) model simulation datasets of size-resolved dust mass concentrations, from which we estimate the modelled dust size distribution; and (3) reanalysis datasets of the dust aerosol optical depth. We focus here only on describing the model simulations (section 3.1) and

the reanalysis products (section 3.2), as details of the *in-situ* measurements used to constrain the globally-averaged datasets are described in Kok et al., (2017). In addition, we also describe the independent measurements we use to evaluate DustCOMM dust size distribution and the dust mass extinction efficiency in section 3.3.

### 3.1     Model Simulations

We use model outputs of dust aerosol properties from six leading atmospheric global models, namely: the Goddard Institute for Space Studies (GISS) ModelE atmospheric general circulation model (Miller et al., 2006); the Weather Research and Forecasting model coupled with Chemistry updated by the University of Science and Technology of China (USTC) suitable for quasi-global simulation (WRF-Chem; Zhao et al., 2010, 2013; Hu et al., 2016); the

Community Earth System Model (CESM; Hurrell et al., 2013); the Goddard Earth Observing System coupled with Chemistry (GEOS-Chem; See Kok et al., 2017); the ARPEGE-Climat model from the Centre National de Recherches Météorologiques Earth system model (Michou et al., 2015); and the Integrated Massively Parallel Atmospheric



Chemical Transport (IMPACT; Ito & Kok, 2017 and references therein) model. We use the different simulations from global climate and chemical transport models between 2004-2008 (except for WRF-Chem and IMPACT which are 2007-2016 and 2004 respectively) to capture the general model uncertainties that are associated with the dust emission, transport, and deposition processes. The GISS, CESM and GEOS-Chem model simulations are described in Kok et al. (2017) and the references therein (see section 5 of their supplementary document). Here, we supplement these simulations with three additional simulations from the WRF-Chem, ARPEGE-Climat and IMPACT models. The WRF-Chem model simulation represents an updated USTC version of that was used in Kok et al. (2017). Further details of these three additional model simulations are thus given in the supplementary document.

We obtain the spatially-varying dust size distribution from each of the six model simulations, which we use to define the spatial variability of the DustCOMM dust size distribution (see Section 2.1). We summarize the particle bin ranges, time periods, spatial resolutions, as well as the meteorology used for each model simulation of the dust size distribution in Table 1. All the models use discrete bins that represent the dust particles up to about 10 μm, except for the GISS, ARPEGE-Climat and IMPACT models, which extend beyond the 10μm diameter limit. Four of the models – WRF-Chem, CESM, ARPEGE-Climat, and IMPACT – have a lower diameter limit smaller than 0.2μm. For consistency, we set the lower diameter limits for all the model simulations to the common diameter of 0.2μm, and correct the upper diameter limit to 20μm, following the procedures described in section 2.1.1. In addition, since the time periods are different for the available model dataset (Table 1), we focus on annual and seasonal climatologies, which we obtain here from the monthly means of the model outputs.

In order to test our hypothesis that integrating experimental and observational constraints on dust size and shape distributions can constrain 3-D dust properties more accurately than possible from model simulations alone, we obtain a model ensemble of 3-D dust size distribution and mass extinction efficiency and 2-D dust column loading. To do so, we interpolated seasonal and annual climatologies of these dust properties to a common resolution of approximately 2.5° by 2.0° spatial resolution, with 35 levels from the surface to 100 hPa. In addition, we correct each modelled dust size distribution to a common particle bin spacing between 0.2-20μm by assuming a power-law distribution between nearby model particle bins. After putting all the model simulations on the same footing in this manner, we thus represent the ensemble of the model dust size distribution with the mean, standard deviation and range (minimum-maximum value), as a function of particle sizes, horizotal locations, heights, and seasons. Where neccesary, the 95% confidence interval of the model ensemble is estimated as 1.96 times the standard error (e.g. Altman & Bland, 2005). We also perform a similar aggregation and interpolation procedure on the modelled dust aerosol optical depth and column-integrated atmospheric dust loading, which are used to calculate the column-integrated dust mass extinction efficiency (MEE) for each model and thus for the model ensemble.

## 3.2 Reanalysis Dust Aerosol Optical Depth

We obtain the dust aerosol optical depth from four reanalysis products to constrain the atmospheric dust loading for DustCOMM (see section 2.3). These four reanalysis products are: the Modern-Era Retrospective analysis for Research and Applications, Version 2 (MERRA-2; Gelaro et al., 2017); the Navy Aerosol Analysis and Prediction System





(NAAPS; Lynch et al., 2016); the Japanese Reanalysis for Aerosol (JRAero; Yumimoto et al., 2017); and the Copernicus Atmosphere Monitoring Service (CAMS) interim Reanalysis (CAMSiRA; Flemming et al., 2017). While the description of each reanalysis product can be found in the supplementary documents, we give a general overview in this section.

A key advantage of these reanalysis products is that they assimilate data from several observing systems, and thus provide a complete spatial and temporal coverage of atmospheric composition that captures its variabilities and trends (Buchard et al., 2017). Most of these four reanalysis products assimilate similar satellite and ground-based observations of AOD, which includes data from at least one or all of the following observing systems: the *Terra* and

*Aqua* satellites of MODerate resolution Imaging SpectroRadiometer (MODIS), the Advanced Very High Resolution Radiometer (AVHRR), the Multi-angle Imaging SpectroRadiometer (MISR), as well as ground-based observation of AOD from several Aerosol Robotic Network (AERONET) stations (Lynch et al., 2016; Flemming et al., 2017; Gelaro et al., 2017; Yumimoto et al., 2017). In addition, some reanalysis products also assimilate other aerosol constituents and reactive gases, like carbon monoxide and ozone observations from the Measurements Of Pollution In The

Troposphere (MOPITT) instrument on the Terra Satellite, Solar Backscatter Ultraviolet (SBUV/2) instruments (from various National Oceanic and Atmospheric Administration (NOAA) platforms), and Microwave Limb Sounder (MLS) ozone profiles (e.g. Flemming et al., 2017). These observations are mostly bias corrected before they are assimilated through radiatively-coupled aerosol models, and used to constrain the different species that constitute the aerosol particles in the atmosphere.

Although the total AOD is constrained, errors in each reanalysis model's treatment of emission, transport, and deposition of mineral dust introduces uncertainties. Dust emission and deposition in the assimilation procedure are either modelled or sometimes constrained by observations. For example, the dust emission for NAAPS is constrained by using a regional source tuning that is, in turn, constrained by space-based and ground-based AOD observations

(Lynch et al., 2016). Other reanalysis products use dust emissions that are parameterized and model dependent (e.g. Yumimoto et al., 2017). In general, wet deposition is partially constrained by the assimilated global satellite-based precipitation information, such as from the NOAA Climate Prediction Center MORPHing technique data (CMORPH). Dry deposition is still mostly model dependent, but may also be adjusted based on assimilated AOD. For all the reanalysis products, aerosol transport in the atmosphere is constrained by the assimilation of several meteorological

observations of winds and temperature. Hence, in order to constrain the dust AOD, the assimilation procedure takes advantage of the best features in both the observations and model simulations.

Similar to our treatment of the model simulations described in section 3.1 above, we use annual and seasonal climatologies of dust AOD obtained from monthly averages of the reanalysis products. We use the reanalysis dust

AOD from 2004-2008 for each reanalysis product except for JRAero, for which we use 2011-2015. In order to combine the different reanalysis dust AOD products, we interpolate each product to approximately 2.5° by 2.0° spatial resolution and estimate the ensemble mean and standard error over each location (see section 2.3).

### 3.3 Description of measurements used for evaluation



We use several types of published measurements to evaluate the dust size distribution and dust mass extinction efficiency from both DustCOMM and the model ensemble. We select 20 studies that measured dust properties – 13 of these reported dust size distributions, and 11 of these reported dust mass extinction or scattering efficiencies (Table 2). These measurements were taken both near and far from dust-dominated regions (Table 2 and supplementary Fig. S-3). While some measurements were taken close to (or over) some of the northern hemisphere deserts – such as the Sahara, Middle East, and Asian deserts - no measurements were taken close to the southern hemisphere deserts. 12 of the 20 studies obtained measurements near the Sahara Desert, while one measurement each was taken near the Middle East (Sde Boker, Israel), and Asian (Qinghai Province, China) deserts. Other measurements represent dust properties at different distances of transport away from the dust sources.

Except for four measurements, most of the data are taken during airborne field campaigns that often occur over a wide geographical area, several altitude levels, and several days (Table 2). As such, studies often report measurements that represent the averages of the dust properties taken during the campaign. Details of the flight path, showing the locations where dust particles are encountered, are not always reported. To use these measurements, we therefore define a representative location and altitude for each measurement based on the area where the majority of dust was encountered. In addition, since the measurements often represent average of several days and sometimes multiple months, we also compare them against seasonal averages of the DustCOMM and model ensemble estimates.

Below, we give a broad overview of the measurements of the dust size distribution and mass extinction efficiency, and further information on each study, including the instruments used, can be found in the supplementary document.

### 3.3.1 Dust size distribution measurements

Dust size distribution measurements are taken using a variety of instruments with different sizing methodologies (e.g. Reid et al., 2003). These instruments generally fall within the categories of sample collectors (e.g. D'Almeida & Schutz, 1983; McConnell et al., 2008), cascade impactors (e.g. Chou et al., 2008; Kandler et al., 2009) and aerodynamic particle sizers (e.g. Otto et al., 2007), and optical particle counters or spectrometers (e.g. Chou et al., 2008; Clarke et al., 2004; Otto et al., 2007). The first category of instruments, sample collectors, are usually installed behind filters or thermal denuders to remove non-dust particles. The aerosol samples are then analysed using electron or light microscopy techniques, where they are counted and sized either manually or using an automated software. This type of measurement yields dust size distribution with respect to geometric diameters. For the second category of instruments, cascade impactors and particle sizers, aerosol particles are usually accelerated through a jet outlet, and sometimes collected on a substrate. Using these instruments, the aerosols are sized based on the mass-to-drag characteristics of the particles. Dust particle sizes measured using these types of instruments are associated with the aerodynamic diameter. Finally, the optical particle counters generally have separate channels for different particle sizes and determine particle sizes in optical diameters based on the amount of light they scatter. For many of the studies we use here, these instruments are sometimes combined to verify the accuracy of the measurements (e.g. Ryder


et al., 2013). For all dust size distribution measurements, the studies that used aerodynamic or optical sizing instruments eventually report the measured size distribution in geometric diameters.

An important consideration is the elevation at which the dust size distributions are measured. With the exception of two studies (D'Almeida and Schutz, 1983; Kandler et al., 2011) that took measurements at ground stations, most measurements were performed solely aboard aircrafts with in-cabin or wing-mounted instruments. Ground stations were equipped with stationary instruments to collect aerosol samples or stationary optical particle counters to measure size distributions directly. For aircraft measurements, size distributions are often measured during flight segments at constant altitude – also called horizontal legs.

Regardless of the instrument used and height of measurements, most dust size distribution measurements are subject to uncertainties associated with measurement type or presence of other aerosol species, such as biomass burning aerosols. The contamination by other aerosol specie is common for fine-mode dust particles, especially dust particles less than ~0.5µm (e.g. Dubovik et al., 2000; Clarke et al., 2004), since to the instruments these aerosols are indistinguishable from dust particles of the same size. This causes a high bias in the fine-mode of measured dust size distributions (e.g. Clarke et al., 2004). Another important measurement error arises from assumptions made about the non-sphericity of dust particles. For example, during the microscopy analysis, particle diameters are usually determined as the volume-equivalent geometric diameters based on 2-dimensional images. Since dust particles have a small height-to-width ratio (Okada et al., 2001), the resulting size distribution may overestimate dust particle diameters. In the case of cascade impactors and particle sizers, unusual dust particles shapes and the possibility of particle bouncing off the substrate may lead to significant bias, especially for coarse-mode particles. For dust measurements that used optical particle counters, irregularly-shaped dust particles are often assumed to be spherical in order to convert them to volume-equivalent geometric diameters, but light scattering between spherical and non-spherical particles are different. These assumptions often lead to biases, that many studies try to account for to various degrees (e.g. Ryder et al., 2013; Ryder et al., 2018).

### 3.3.2 Mass extinction efficiency measurements

In the literature, the term dust mass extinction efficiency (MEE) is sometimes used interchangeably with the mass scattering efficiency (MSE; e.g. Hand & Malm, 2007). This is because for typical solar wavelength at 550 nm, dust particles scatter more radiation than they absorb for $D \leq 10$ µm. Despite the strong scattering by these particles, larger particles ($D \geq 10$ µm) often exhibit substantial absorption relative to scattering in the visible wavelength (e.g. Ryder et al., 2018). In order to put all the measurements in the same equal footing, we convert the reported dust MSE in some of these studies to dust MEE, by using measured scattering albedo value of $0.95 \pm 0.02$ (Haywood et al., 2003; Ryder et al., 2013; Ryder et al., 2018).

Mass extinction efficiencies (MEE) that are reported in the literature are generally derived using two methods: regression and theoretical methods (e.g. Hand & Malm, 2007). The regression method calculates the dust MEE as the slope between the dust extinction coefficient ($m^{-1}$) and the dust mass concentration (g $m^{-3}$). For this case, the dust





samples are typically collected using filters, while aerosol extinction is measured using nephelometers. The difficulty however is that measured total aerosol extinction from the nephelometer may be influenced by several aerosol species other than dust particles. Some studies ignore the impact of other aerosol species, and derive the dust MEE using the total aerosol extinction and the collected dust mass concentration (e.g. Li et al., 1996). Others take advantage of the

linear relationship between the aerosol extinction and mass concentrations in order to separate the column MEE into constituents that correspond to each aerosol specie, using a multivariate linear regression method (e.g. Andreae et al., 2002; Maring et al., 2003). Such calculations therefore require that all the aerosol species contributing to the extinction are included. With this in mind, the regression-derived MEE is therefore subject to several systematic and random errors. These errors include errors due to the instrument measuring the extinction coefficient, meteorological influence

that may impact some aerosol species but not others, possible collinearity between several constituent aerosol species, and the assumption of internal or external mixing (Hand and Malm, 2007).

The theoretical method calculates the dust MEE using the measured size distributions of dust mass or number concentration (Seinfeld & Pandis, 2016). This may take the form of calculating the dust MEE directly using the dust

size distribution and the estimate of single-particle extinction efficiency, or indirectly by first calculating the size-resolved dust extinction coefficient, using dust size distribution, and then combining the result with dust mass concentration. In either case, the dust density, shape and index of refraction are needed. While assumptions of dust density and index of refraction are typically based on previously reported measurements, dust shapes are generally assumed to be spherical, which is contrary to observations (e.g., Okada et al., 2001; Kandler et al., 2007). This is a

major disadvantage that may result in an underestimation of the derived dust extinction efficiency (e.g. Kok et al., 2017). Another source of error is associated with the instrument used to measure the aerosol size distribution, which may assume certain mixing properties of the observed aerosols. For mobility measurements (differential mobility analyser, DMA), optical measurements (optical particle counter, OPC) or aerodynamic measurements (aerodynamic particle sizer, APS), aerosols are often assumed to be internally mixed (e.g. Quinn et al., 2002; Clarke et al., 2004).

In contrast, for an impactor, aerosols are often assumed to be externally mixed (e.g. Chiapello et al., 1999; Osborne et al., 2008).

Despite the differences between both methods used to derive dust MEE from observed quantities, previous studies have highlighted that they both produce similar values within measurement uncertainties (e.g. Maring et al., 2000;

Quinn et al., 2004). In addition, for measurements where only the mean dust MEE/MSEs are reported, but not the uncertainty estimates, we estimate here in this study what the measured uncertainty estimate could be by assuming that its relative uncertainty (that is the ratio of the presumed uncertainty to the reported mean) is proportional to the mean relative uncertainty that is calculated from other measurements. While this estimated uncertainty may likely not be representative of the specific field campaign to which the measurement was taken, they are likely representative of

the seasonal values over the region.

## 4.  Results





In this section, we present the DustCOMM products obtained using the methodology and data described above. We first present the dust particle size distribution (PSD; section 4.1) and then the dust mass extinction efficiency (MEE; section 4.2). In each case, we evaluate the DustCOMM and the model ensemble products against available *in-situ* measurements. We show that DustCOMM products generally reproduce observations better than model ensemble estimates. We then compare the spatial variability of the DustCOMM products against the model ensemble. In section 4.3, we compare the atmospheric dust loading obtained from both DustCOMM and the model ensemble, and we examine the spatial distribution of the uncertainty in all DustCOMM products in section 4.4.

### 4.1  Dust Size Distribution

#### 4.1.1    Evaluation of DustCOMM against measurements

We evaluate DustCOMM and the model ensemble PSD against available in-situ measurements taken during field campaigns (Figs. 2 & 3). We compare these location-based measurements against season-averaged DustCOMM and model ensemble estimates. The reason for using the seasonal averages is justified in section 3.3 above. An additional justification for the comparison between the individual measurements and the season-averaged DustCOMM and model ensemble estimates is that the variability of the normalized dust PSD within each season is relatively small, especially for dust with $D \leq 10 \ \mu m$ (e.g. McConnell et al., 2008; Mahowald et al., 2014).

Model simulations of dust PSD generally show substantial errors when compared against measurements. In each of the 12 studies used in Fig. 2, the model ensemble overestimates the observed fine-mode particles (defined here as $D \leq 2.5 \mu m$) and underestimates the coarse-mode particles (defined here as $D \geq 5 \ \mu m$). In some of the cases, the overestimation extends above $D = 2.5 \mu m$ and the underestimation below $D = 5 \mu m$. Nevertheless, these differences are apparent in all the comparisons, and consistent with previous studies indicating more coarse-mode dust particles are in the atmosphere than models account for (e.g. van der Does et al., 2016, 2018; Kok et al., 2017; Ryder et al., 2018).

In contrast, the DustCOMM dust PSD shows overall better agreement against measurement than the model ensemble (Fig. 2). This improved agreement includes a substantial reduction of the underestimation of coarse-mode dust, as well as a reduction of the overestimation of some fine-mode particle sizes. Although DustCOMM better reproduces the measurements for $D \geq 0.5 \mu m$, it shows poorer agreement for $D \leq 0.5 \ \mu m$ (e.g. Fig. 2e, h, i, j), underestimating the measurements by about one to two orders of magnitude. For example, during DARPO (Fig. 2e; Wagner et al., 2009) and BACEX (Fig. 2h; Jung et al., 2013), the differences between DustCOMM PSD and the measurements are about two orders of magnitude. The $D \leq 0.5 \ \mu m$ size range is also the size range in which measurements of dust PSD are potentially contaminated by the presence of other aerosol species (see section 3.3.1 and section 5.1). In addition to the disagreement for $D \leq 0.5 \ \mu m$, there is also some disagreement for $D \geq 10 \ \mu m$ (e.g. Fig. 2d, e, h), although for fewer cases. Overall, the DustCOMM dust PSDs significantly better represent the measurements in the $0.5 \leq D \leq 20 \ \mu m$ size range than the model ensemble.





DustCOMM also shows better agreement than the model ensemble against measurements of the dust PSD as a function of altitude (Fig. 3). We highlight here measurements taken from three campaigns: (1) the ACE-2 campaign (June/July, 1997) off the west coast of Western Sahara and Morocco (Otto et al., 2007); (2) the Fennec project (June 2011) between the Canary Islands and Mauritania/Mali (Ryder et al., 2013); and (3) the AER-D campaign in August 2015

near Cape Verde Island (Ryder et al., 2018). All three cases show that a significant fraction of coarse-mode dust particles, including with $D \geq 10 \ \mu m$, are transported off the coast of North Africa. We compare these measurements at selected altitude of 2500 m  (2700 m for ACE-2), 4000 m, 5500 m, and 6000 m (7000 m for ACE-2). Similar to Fig. 2 above, the DustCOMM dust PSD agrees better with the measurements than the model ensemble for these measurements at similar 2-D location but at different altitudes. For dust particles with $D \leq 0.5 \ \mu m$, the DustCOMM

size distributions also differ from the measurements by about an order of magnitude (similar to Fig. 2) for altitude at 2500 m. However, this difference increases to more than two orders of magnitude above ~4000 m altitude.

In summary, the overall differences between the in-situ measurements and DustCOMM are significantly smaller than the differences between the measurements and the model ensemble, especially for $D \geq 0.5 \ \mu m$. To quantify this, we

report the log-mean bias in each bin following Eqn. 10 and using all the measurements shown in Fig. 2 & 3 (see supplementary Fig S-4 for the error estimates). DustCOMM shows an overall reduction in the bias relative to the model ensemble, except for dust particles with $D \leq 0.5 \ \mu m$ (Fig. 4). For $D \leq 0.5 \ \mu m$, model shows an average (95% CI) positive log-mean bias of 0.26 (-0.08 — +0.6), while DustCOMM shows an average negative log-mean bias of -0.92 (-1.18 — -0.73). In contrast, DustCOMM shows a remarkable reduction in the average log-mean bias in the 0.5

$\leq D \ \leq 10 \ \mu m$ size range; for instance, the bias for the $5 - 10 \ \mu m$ bin is ~90% less than it is for the model ensemble. DustCOMM also shows a substantially reduced bias in the $10 \leq D \ \leq 20 \ \mu m$ size range, although the bias here remains substantially negative, indicating a persistent underestimation of these coarse particles. On average, DustCOMM reduces the log-mean bias for dust particles with $D \geq 0.5 \ \mu m$ by about 46%, relative to the model ensemble.

### 4.1.2  Global comparison between DustCOMM and the model ensemble

Considering that the DustCOMM dust PSD agrees better with in-situ measurements than the model ensemble, we now compare the differences between DustCOMM and model ensemble PSDs. Specifically, we first compare the

differences in the shape of the globally-averaged dust size distribution between DustCOMM and the model ensemble (section 4.1.2.1). Second, we examine the changes in the spatial variability of the DustCOMM and model dust mass fraction as a function of particle size range (section 4.1.2.2).

#### 4.1.2.1  Differences in dust size distribution

As we already concluded based on in-situ measurements, climate models globally overestimate fine-mode dust particles ($D \leq 2.5 \ \mu m$) and under-estimate coarse-mode dust particles ($D \geq 5 \ \mu m$), relative to globally-averaged DustCOMM dust PSD (compare black and coloured lines in Fig. 5a). On average, simulations in our model ensemble





overestimate the fine mode by ~14%, and underestimate the coarse mode by ~15%. The degree of this deviation from DustCOMM depends on the model, and can be as much as 50% in the fine mode or 37% in the coarse mode.

While the globally-averaged dust PSDs clearly show marked differences, it is also important to quantitatively examine the variability of the dust PSD for all locations. The variability of dust PSDs in the atmosphere is influenced by dust emission, transport, and deposition processes, and it can be assessed by considering metrics such as the volume median diameter (e.g. Maring et al., 2003; Formenti et al., 2011; Mahowald et al., 2014). Thus, the probability distributions of the volume median diameters (VMD) for the model simulations are generally biased towards smaller VMD values, with different peak diameters for each model. WRFChem and IMPACT show the lowest VMD at ~1.9 μm, and ARPEGE-Climat shows the highest VMD at ~5.5μm (Fig. 5a). In contrast, the DustCOMM VMD peaks around 5 μm. The probability distribution also shows that that the DustCOMM VMD lies between approximately 2.5 μm and 6.5 μm at most heights and locations (Fig. 5b). This range is consistent with the range of measured VMD (3-6 μm) for coarse-mode dust particles generally reported in the literature and compiled by Reid et al. (2003; see their Table 1). It also falls within the range of values measured at near-source regions and farther downwind. For instance, the VMD calculated from dust particle size distributions measured at Cape Verde, off the coast of North Africa (Ryder et al., 2018) is about 5.5μm. Farther downstream where dust particles are likely to deposit after long-range transport, the VMD values near Puerto Rico is approximately 4μm (Maring et al, 2003). It is noteworthy however, that some studies (e.g. Carlson & Caverly, 1977; Weinzierl et al., 2009) have reported measured VMD values that exceeds 13 μm, but these studies often include giant-mode dust particles with $D \geq 20\mu m$, whereas we limited our analysis to dust with $D \leq 20\mu m$ (see Section 2.1.1). Overall, DustCOMM shows better consistency with observations of VMD than model simulations.

### 4.1.2.2 Changes in spatial variability of dust mass fraction

Although coarse-mode particles dominate the dust mass fraction near source regions and fine-mode particles dominate the dust mass fraction in the far remote regions, there are considerable changes in the spatial variability of the dust mass fraction between DustCOMM and the model ensemble (left and middle panels of Fig. 6). As highlighted above in section 4.1.2.1, there is a general decrease of DustCOMM dust mass fraction for particles between $0.2 - 2.5\mu m$ and $2.5 - 5\mu m$, relative to the model ensemble (right panel of Fig. 6). In contrast, there is an overall increase of DustCOMM dust mass fraction for particles between $5 - 10\mu m$ and $10-20\mu m$. These changes cause DustCOMM to produce generally better agreement against in-situ measurements than the model ensemble, as shown in section 4.1.1 above. Overall, the most significant changes in DustCOMM dust mass fraction, relative to the model ensembles, are near dust-dominated regions, resulting in a decrease of up to 26% and an increase of up to 29% for dust particles between $2.5 - 5\mu m$ and $10 - 20\mu m$ respectively.

These changes in the dust mass fraction gradually decrease away from the dust-dominated regions. This is evident, for example, over the North Atlantic basin, where dust from the Sahara Desert is transported to the Caribbean and South America. Models generally simulate fewer large dust particles ($D \geq 5\mu m$), and thus transport only a small





fraction to the Caribbean. But observational evidence shown earlier in Figs. 2h & l indicates that dust in Barbados includes a significant fraction of coarse dust. Thus, the east-west gradient and the overall increase of the DustCOMM dust mass fraction over the North Atlantic helps resolve the underestimation of long-range transported coarse particles, such as near Barbados (Fig. 6; e.g. Weinzierl et al., 2017).

The vertical distribution of the DustCOMM dust mass fraction shows differences with the model ensemble that are consistent with the globally-averaged differences (Fig. 7) – that is, DustCOMM dust mass fractions are lower than for the model ensemble for particles between $0.2 - 2.5\mu m$ and $2.5 - 5\mu m$, and higher for particles between $5 - 10\mu m$ and $10 - 20\mu m$. It is noteworthy here that vertical changes in the dust PSD in DustCOMM are based on model simulations, causing a similarity in the shape of the vertical profile of the dust mass fraction between DustCOMM and the model ensemble.

### 4.2 Dust Mass Extinction Efficiency

### 4.2.1 Evaluation of DustCOMM against measurements

We evaluate the dust mass extinction efficiency (MEE – $m^2g^{-1}$) of DustCOMM and the model ensemble against measurement (Fig. 8). These measurements span from those taken near dust source regions such as the Saharan, Middle East and Asian deserts, to those taken farther downwind from source regions (Table 2). Higher values of dust MEE are expected where fine-mode dust particles dominate, because smaller dust particles scatter light more efficiently per unit mass at visible wavelengths. In contrast, dust MEE decreases as the coarse-mode fraction increases. Thus, observed dust MEE values generally range between ~0.3-0.8 $m^2g^{-1}$ at approximately 550 nm.

DustCOMM shows better agreement with measurements of dust MEE than the model ensemble, regardless of the season and location (Fig. 8). DustCOMM dust MEE estimates are within the measurement uncertainty range for most of the 11 studies used here. A notable exception is the comparison at Sde Boker, Israel (Andreae et al., 2002), where both the DustCOMM and the model ensemble underestimate the measured MEE. Nevertheless, the DustCOMM estimates better reproduce the lower values of dust MEE near dust sources, and the higher values farther downstream. For example, lower dust MEE values near the Sahara Desert, between Niamey and the Canary Islands (generally below 0.6 $m^2g^{-1}$), and higher values farther downstream, such as over Barbados, are better reproduced by DustCOMM. DustCOMM dust MEE also compares well against measurements at the same location but for different seasons. An example is the measurements over Cape Verde, off the coast of North Africa (Haywood et al., 2003; Ryder et al., 2018), taken in September 2000 and August 2015. For both cases, DustCOMM estimates compare better with the observed dust MEE, while the model ensemble over-estimates the values in both cases.

DustCOMM also reproduces the observed dust MEE values with strong spatial gradient, measured during the same campaign (INDOEX) over the Arabian Sea and Indian Ocean (Quinn et al., 2002). Dust particles emitted from Middle East deserts can get transported over the Arabia sea, and are deposited over the Indian Ocean where strong precipitation occurs year-round (e.g. Kulshrestha et al., 1996). Since dust MEE increases with distance from source regions due to deposition of larger dust particles, the measured dust MEE values increase from 0.5 $m^2g^{-1}$ measured in





the Arabian Sea to 0.75 $m^2g^{-1}$ in the Indian Ocean, south of the equator. DustCOMM captures much of this gradient, and is in better quantitative agreement than the model ensemble estimate (Fig. 8).

DustCOMM also shows better agreement than the model ensemble against the observed dust MEE averaged over all
measurements (see the last column of Fig. 8). DustCOMM shows a very small difference with the mean of the measurement estimates [0.007 $m^2g^{-1}$ (95% CI is -0.04—0.08)], whereas the model ensemble mean (95% CI) overestimates the measurements by 0.12 (-0.17 – 0.4) $m^2g^{-1}$ – that is about 94% reduction in the mean bias. We further assess DustCOMM performance by calculating the reduced chi-square ($\chi_\epsilon^2$; Eqn. 11); a value of $\chi_\epsilon^2 > 1$ highlights the degree that a model does not fit the observations within the uncertainty range (e.g. Andrae et al., 2010). DustCOMM
shows a $\chi_\epsilon^2$ value of 1.19 , in comparison to the model ensemble with $\chi_\epsilon^2$ value of 8.70 (Fig. 8), thereby showing a substantial improvement.

### 4.2.2    Global comparison between DustCOMM and model ensemble

After showing that DustCOMM better reproduces measurements of dust MEE than the model ensemble, we now compare the spatial variability of the DustCOMM and model ensemble dust MEE. To do so, we estimate the column-integrated dust MEE, weighted by the dust vertical distribution, for DustCOMM and model ensembles over each location (Fig. 9 a & b). Both DustCOMM and model estimates show smaller values of dust MEE over dust-dominated regions and higher values farther downwind – like over the Inter Tropical Convergence Zone (ITCZ), the eastern
Pacific Ocean and the polar regions. Although DustCOMM and model ensemble estimates are thus spatially similar, important differences exist. Near dust-dominated regions, DustCOMM dust MEE values are lower than model ensembles, but farther downstream, DustCOMM values are higher than model ensembles. This regional difference in dust MEE values corresponds to similar difference in dust mass fraction, with fractional increase in coarse-mode dust over dust-dominated regions than farther downstream (compare Fig. 9 & 6). In addition, there is also a gradual east-
to-west changes in the dust MEE values as coarser dust particles are deposited away from dust sources, consistent with similar changes in dust mass fraction shown earlier in Fig. 6. The globally-averaged DustCOMM dust MEE values are lower than predicted by the model ensemble. The global mean of dust MEE for DustCOMM and model ensembles are 0.68 (Min-Max: 0.22— 1.1) $m^2 g^{-1}$ and 0.95 (Min-Max: 0.30— 1.98) $m^2 g^{-1}$ respectively.

### 4.3    Global comparison of atmospheric dust load between DustCOMM and models

After obtaining the DustCOMM dust MEE as described in the previous section, we combine this with the reanalysis-derived dust AOD (Eqn. 9; see also section 2.3) to obtain the atmospheric dust loading. We find that the DustCOMM dust column loading is generally larger than the model ensemble estimate (Fig. 10a & b). DustCOMM shows
substantially larger dust column loading than the model ensemble over desert regions, such as the Middle East, and Asian deserts. The relative increase of dust load in DustCOMM over the Asian desert is more than twice the increases over the Middle East desert. DustCOMM also shows larger dust column loading over most parts of the North African desert, except some parts that includes the north-western section and the coastal regions which show smaller dust column loading than the model ensemble. Although reanalysis-derived mean dust AOD over North Africa is





substantially lower than the model ensemble, it is within the uncertainty estimates, which is higher over this region (see supplementary Fig. S-4; see also Ridley et al., 2016). In addition, DustCOMM estimates over the Australian deserts show a lower dust column loading than the model ensemble, similarly corresponding to lower reanalysis-derived dust AOD (Fig. 10c & supplementary Fig. S-4). Overall, globally-averaged DustCOMM dust column loading

is about 46% higher than the model ensemble.

### 4.4    Spatial distribution of DustCOMM relative uncertainty

We examine here the spatial distribution of the DustCOMM relative uncertainty– that is, the uncertainty characterizing

68% of the distribution of each variable over each location divided by the mean value of that variable at that location. We do this for the dust mass fraction for the particle bins shown in Fig. 6 & 7, the dust MEE, and the dust load (Fig. 11).

The relative uncertainties in the DustCOMM fine-mode fraction ($D = 0.2 − 2.5\mu m$) are higher mostly near emission

regions (Fig. 11a), while the relative uncertainties in the coarse-mode fractions are higher over remote regions, especially for $D = 10 − 20\mu m$ (Fig. 11d). These uncertainties are, in part, directly associated with the uncertainties in the measurement constraints. The globally-averaged constrained dust size distribution (Eqn. 1) has a higher relative uncertainty for the $D \leq 1\ \mu m$ and $D \geq 10\ \mu m$ diameter range than for the $1 \leq D \leq 10\ \mu m$ diameter range (see Fig. 2 in Kok et al., 2017), and we propagate these uncertainties over every location. In addition, the spatial distribution

for the relative uncertainties in the dust mass fraction is similar to that of the model ensembles (supplementary Fig. S-5), which is also propagated into the DustCOMM product.

The relative uncertainties in DustCOMM dust MEE are mostly higher over dust-dominated regions (Fig. 11e). The dust MEE is influenced by the uncertainty in the constrained globally-averaged extinction efficiency, which in-turn is

partially due to uncertainties in the measurements of index of refraction and dust particle shapes (see Fig. 1b in Kok et al., 2017), all of which are propagated into the DustCOMM dust MEE. In addition, the relative uncertainties in the dust MEE are also affected by the uncertainty in the dust size distribution. Thus, the spatial distribution of dust MEE relative uncertainty is particularly informed by the uncertainties in the fine-mode and coarse-mode dust particles (compare Fig. 11a & d with 11e). For the most part, uncertainties in the fine-mode dust fraction appears to dominate

the uncertainties in dust MEE, more than the uncertainties in the coarse-mode dust fractions.

The relative uncertainties in the DustCOMM dust column loading are mostly higher over remote regions, where the mean dust load is small (Fig. 11f). Though the dust column loading is influenced by the uncertainties in dust MEE, the spatial distribution of the relative uncertainties in dust load is largely informed by the uncertainties in the reanalysis

dust AOD (see supplementary Fig. S-4).

### 5     Discussion





We presented the DustCOMM products in the previous section, where we showed that both the dust particle size distribution (PSD) and the dust mass extinction efficiency (MEE) are reproduced more accurately than by an ensemble of model simulations. Despite the overall agreement with observations, there are some disagreements highlighting potential limitations of our methodology. In this section, we discuss these disagreements between DustCOMM and

measurements and provide possible insights into these discrepancies (section 5.1). We also discuss the impact of dust sizes and asphericity on DustCOMM dust mass extinction efficiency (section 5.2), and we highlight the limitations in using modelling constraints as part of DustCOMM estimates (section 5.3). We end by highlighting how our constrained DustCOMM products can be used by the research community to potentially improve estimates of dust impacts on the Earth system (section 5.4).

**5.1 Cause of discrepancy between DustCOMM and size distribution measurements**

The evaluation of the DustCOMM PSD shows an underestimation of dust with $D \leq 0.5\mu m$ and $D \geq 10\mu m$ (Figs. 2, 3 & 4). This is in contrast to the ensemble of model simulations overestimating the dust mass fractions for $D \leq 0.5\mu m$,

and underestimating the dust mass fraction substantially more than DustCOMM for $D \geq 10\mu m$. Although the comparison between date-specific individual measurements and season-averaged DustCOMM dust PSD is expected to induce errors, this difference cannot explain the apparently systematic difference between measurements and the DustCOMM dust PSD for both $D \leq 0.5\mu m$ and $D \geq 10\mu m$ (Fig. 4). We provide here possible reasons for this disagreement between DustCOMM and observations.

First, DustCOMM's underestimation of dust with $D \leq 0.5\mu m$ may be caused by contamination of the measured size distributions by other aerosol species for $D \leq 0.5\mu m$. Studies have shown that a substantial fraction of aerosols with $D \leq 0.5\mu m$ are not mineral dust, even in dust-dominated regions (Chou et al., 2008; Kandler et al., 2009; Weinzierl et al., 2009). For example, during the Saharan Mineral Dust Experiment (SAMUM) over southern Morocco, Kandler

et al. (2009) showed that more than 50% of the measured particles with $D \leq 0.5\mu m$ are ammonium sulphates or mixture of sulphate and dust. Even when strict measurement techniques are used to separate other non-mixing aerosol components, the aerosol mixing state for $D \leq 0.5\mu m$ often leads to outer coating of available dust particles, thus leading to a higher particle volume that overestimates the true dust size (Weinzierl et al., 2009). In addition, campaign logistics often require that some measurements of dust properties are taken close to major cities, where contaminations by other aerosol species, such as biomass-burning aerosols or urban pollutions are possible (e.g. McConnell et al.,

2008; Wagner et al., 2009). For example, Clarke et al. (2004) highlighted that the presence of biomass-burning aerosols (e.g. soot) led to a variability of about 2 orders of magnitude for measured size distributions with diameter less than $D \leq 0.6\mu m$ during the ACE-Asia campaign. This variability is consistent with the average difference between our estimates and the observations for $D \leq 0.5\mu m$. After separating out the contamination of the soot-mode

from the dust size distribution, their resulting dust PSD generally agrees with our estimate within the uncertainty range (Fig. 2f). Thus, the large variability of the measured size distribution is indicative of the potential problems with the representation of dust particles with $D \leq 0.5\mu m$.





Second, the constraint on the globally-averaged dust size distribution could also underestimate the contribution from dust with $D \leq 0.5\mu m$. A key input to this constraint is the emitted dust size distribution, but there is a dearth of measurements of the mass fraction of emitted dust with $D \leq 0.5\mu m$, leading to uncertainty in constraining the globally-averaged emitted dust size distribution with $D \leq 0.5\mu m$ (Kok et al, 2017). Moreover, the measurements of emitted dust size distribution with $D \leq 0.5\mu m$ that do exist (e.g. Fratini et al., 2007; Sow et al., 2009; see Fig. 1c in Kok et al., 2017) indeed show a larger dust mass fraction than represented in the constraint on the globally-averaged emitted dust size distribution. Therefore, more measurements of the size distribution of emitted dust particles extending to very fine sizes are needed.

Third, the underestimation of dust with $D \geq 10 \ \mu m$ by both DustCOMM and the model ensemble might be caused by biases in both global model simulations and the constraints on the global dust size distribution used by DustCOMM. Similar to $D \leq 0.5\mu m$, the experimental constraint on the emitted dust size distribution with $D \geq 10 \ \mu m$ also has a large uncertainty because of limited available measurements (Kok 2011). In addition, since spatial and temporal variability of large dust particles ($D \geq 10\mu m$) strongly depend on the model simulation of dust emission and deposition processes, uncertainties in these processes will influence the constraints on DustCOMM dust size distribution. For example, if the giant mineral dust particles are transported far away from the source regions as suggested by observations (e.g., van der Does et al., 2018), the lack of this mechanism would result in a negative bias of the simulated dust atmospheric lifetime (e.g. Huneeus et al., 2011). And since modelling constraints of globally-averaged dust lifetime are used to constrain the globally-averaged size distribution (Eqn. 1), such systematic negative bias may have contributed to the underestimation of dust particles with $D \geq 10\mu m$. Although our methodology partly constrains dust deposition globally, it does not constrain regional variability in dust deposition, and we expect that such uncertainties may increase as a function of distance away from dust-dominated regions. We note here that regional observational constraints on dust lifetime are currently not available, and stronger modelling constraints that may account for the underestimation of coarse dust particles in the atmosphere are a subject for future work.

**5.2 Impacts of dust sizes and asphericity on DustCOMM dust mass extinction efficiency**

The dust MEE is partially determined by the dust size distribution (Eqn. 8). Despite the good agreement between DustCOMM and the measurements of dust MEE (Fig. 8), the size discrepancies in the dust size distribution for particles with $D \leq 0.5 \ \mu m$ and $D \geq 10 \ \mu m$ (Figs. 2, 3 & 4) affect the estimation of dust MEE. Dust with $D \leq 0.5 \ \mu m$ has a large single-particle MEE (Fig. 7a), whereas dust with $D \geq 10 \ \mu m$ has a small single particle MEE (see supplementary Fig. 7). Consequently, errors due to the possible overestimation of both size fractions at least partially cancel each other.

In addition to the impact of dust sizes, dust asphericity also has a substantial impact on the dust MEE. The DustCOMM constraint on dust MEE leverages measurements of dust shape to represent dust particles as an ensemble of tri-axial ellipsoids (Meng et al., 2010; Kok et al., 2017). In contrast, most models use Mie theory, which approximates dust as spherical particles. Thus, the difference between single-particle dust MEE used in DustCOMM and calculated using Mie theory shows the impact of dust asphericity is substantial for both small and lager dust particles, increasing





extinction for particles with $D \geq 1\mu m$ (supplementary Fig. S-6). This implies that typical global model simulations, which contain too much fine-mode dust particles and approximate dust as spherical, the overestimation of the dust extinction due to the fine size bias could (partially) cancel out the underestimation of the dust extinction due to the treatment as dust spherical shapes, leading to nonetheless reasonable agreement with measured dust MEE. However,

for DustCOMM, both the size bias and dust asphericity are accounted for, thereby producing better agreement with measurements (Fig. 8). In addition, accounting for dust asphericity could allow dust particles to stay longer in the atmosphere because asphericity reduces dust settling speed (Ginoux, 2003), which may in turn lead to a more accurate estimation of dust deposition mass fluxes onto land and ocean ecosystems (e.g. van der Does et al., 2016; 2018).

**5.3 Limitations in using modelling constraints**

We used modelling constraints in DustCOMM where observational constraints were either not available or insufficient. In addition to the uncertainties associated with model simulations of dust emission and deposition processes that may influence the constraints on dust size distributions as highlighted in section 5.1, there are other

limitations in the modelling constraints that can influence DustCOMM estimates.

First, one such limitation is the uncertainty in the dust mass spatial distribution of the model ensemble, which directly determines the spatial distribution of dust mass for DustCOMM estimates. Variability in dust emission rates influence the distribution of simulated size-resolved atmospheric dust loading, and consequently the 3-D dust mass fractions. In

addition, ensemble model simulations of dust emission and transport are driven by different meteorological datasets (Table 1), which represent the actual historical meteorology with various degrees of accuracy (e.g. Evan, 2018). Dust transport is also influenced by model resolution and sub-grid parameterizations of wind and turbulence, which differ between models (e.g., Zender et al., 2003; Cakmur et al., 2004). Although averaging over multiple models and over long time periods reduces random errors, systematic errors that affect different models similarly would affect the

model ensemble (e.g. Ridley et al., 2012), and would impact the spatial distribution of dust mass (e.g. Johnson et al. 2012; Ridley et al., 2016). In addition, uncertainties in the vertical distribution of size-resolved dust mass fractions directly affect DustCOMM dust size distributions. Since we use the globally-averaged size-resolved extinction efficiency to constrain the dust MEE over every location (Eqn. 8), the spatial distribution of dust MEE is thus partially determined by the dust size distribution, effectively propagating any uncertainty in model simulations of the spatial

distribution.

Second, some errors may have been introduced while scaling and fitting the different model dust size distributions to a common diameter range (section 2.1). For the scaling procedure (section 2.1.1), the variance of the dust mass fraction in all the bins, including the newly-created ones, are of similar orders of magnitude, thus errors introduced through

this process are small relative to the magnitude of errors in the dust mass fraction. In addition, the resulting dust size distributions are dependent on the specific function and set of parameters used in the fitting procedure (section 2.1.2), which may also introduce some errors.



Third, our constraints on the dust atmospheric loading use ensemble estimates of reanalysis-derived dust AOD, which depends in part on the assimilated aerosol observations, in part on the numerical simulation of dust sources and sinks, and in part on the numerical simulation of other aerosol species. Although some of the reanalysis products try to constrain these dust processes using space-based observations (e.g. Lynch et al., 2016; see supplementary

information), the impact of the uncertainties associated with each process on the DustCOMM estimates of the atmospheric dust loading is beyond the scope of the study.

Finally, this study primarily uses climatologies of modelled dust size distribution between 2004-2008, except for WRF-Chem and IMPACT (see Table 1), and it also scales dust mass loading using the 2004-2008 reanalysis products

(see section 2.3 & 3.2). Thus, any application of our methodology to a different time periods is expected to have some errors. While these errors are expected to be small for the dust size distribution and dust mass extinction efficiency, they may have a substantial impact on the dust mass loading, depending on the inter-annual variability in the reanalysis-derived products and also on the assimilated observations

**5.4 Possible use of DustCOMM to improve estimates of dust impacts on the Earth system.**

Given that DustCOMM estimates of dust aerosol properties are in better agreement with measurements than the model ensemble, DustCOMM could be used to obtain improved constraints on dust impacts on the Earth system than is possible from current global models. Specifically, DustCOMM dust properties could be used as an alternative to

global model simulations in constraining dust impacts, such as the dust direct radiative impact or dust impacts on biogeochemistry and human health. For instance, dust radiative heating rates in the atmosphere strongly depend on the ability of dust particles to absorb solar and terrestrial radiation (e.g. Perlwitz and Miller, 2010). In turn, such absorption depends on the dust size distribution, which strongly influences the optical parameters like the dust absorption optical thickness (e.g. Tegen and Lacis, 1996). With improved constraints on dust size distribution and

therefore the dust optical properties, DustCOMM could be used to determine the dust heating rates in the atmosphere more accurately than possible with current global model simulations. In addition, DustCOMM's improved constraints on atmospheric dust loading and dust size distribution could contribute to better estimates of size-resolved dust concentration near the surface (e.g. Whicker et al., 2018). Over the ocean, such constraints on size-resolved dust concentration could potentially be used for constraints on dust deposition fluxes that are more accurate than possible

from global model simulations.

In addition to being used as an alternative to global model simulations, DustCOMM could also be used to improve the simulation of dust aerosol properties in global models. Incorporating DustCOMM products in the simulation process can potentially be achieved when the aerosol module is coupled with the global model in either the so-called

online or offline modes (e.g. Tegen, 2003; Pérez et al., 2011; Han et al., 2012). In the online mode, the simulated dust size distributions could be adjusted ("nudged") to match the DustCOMM constraints on dust size distribution, similar to what is often done with meteorological fields (e.g. Kooperman et al., 2012). Alternatively, the 3-D dust size distribution could also be corrected offline after the simulated size distribution is obtained but before dust impacts such as on radiation are estimated (e.g. Weaver et al., 2002). Specifically, the modelled dust size distribution can be



corrected by minimizing the differences between the DustCOMM and the modelled size distributions for a specific timescale (see section 2.5). Whether simulated dust properties are corrected in the online or offline modes, using DustCOMM to bias correct global model simulations could produce better estimation of dust impacts, such as dust impacts on radiation, clouds and precipitation, biogeochemistry, and human health.

An example of dust impacts that can be substantially improved by DustCOMM product are dust radiative effects. These radiative effects are sensitive to dust particle sizes and shapes, which are both constrained substantially more accurately in DustCOMM than in models (Fig. 2-6). Smaller dust particles ($D \leq 2.5 \mu m$) scatter more shortwave radiation and cool the climate while larger dust particles ($D \geq 5 \mu m$) absorb more longwave radiation and warm the

climate. Thus, correcting both biases of too much fine dust and not enough coarse dust in models (Figs. 4 & 5), as we do here in DustCOMM, decreases the shortwave cooling and increases the longwave warming (e.g. Otto et al., 2011; Kok et al., 2017). Using the 3-D DustCOMM size distribution to correct modelled dust properties could yield more accurate estimates of dust radiative effects.

In addition, simulated dust impacts on clouds and precipitation can also be improved using DustCOMM dust aerosol properties. For the interactions of dust particles with clouds, it is important to know the number of particles that are activated above a given particle size as cloud condensation nuclei or ice nuclei (e.g. Andreae & Rosenfeld, 2008; DeMott et al., 2015). Therefore, in regions with significant dust loading, accurate estimates of dust size distribution can be key to accurate simulations of precipitation initiation and aerosol-cloud interactions, including dust aerosol

indirect effects (e.g. Sassen, 2003; Doherty and Evan, 2014). Since DustCOMM represent the dust size distribution more accurately than model simulations, it could be used to improve the simulated dust impacts on clouds, precipitation and aerosol-cloud interactions.

Another key advantage of DustCOMM over global model simulations is that it propagates many observational,

experimental, and modelling uncertainties of dust properties, which can be propagated into the calculation of dust impacts on the Earth system. For instance, experimental uncertainties associated with the emitted dust size distributions are propagated into the DustCOMM 3D dust size distribution, and experimental uncertainties in the dust index of refraction and dust particle shapes are propagated into the DustCOMM mass extinction efficiency. In addition, our methodology propagates the uncertainty due to the spread in model predictions of the dust spatial

distribution, although substantial biases in the model ensemble might exist (see section 5.3 for example).

Finally, it is worth noting that DustCOMM can be readily updated as more accurate constraints on dust properties and abundance become available. Current constraints in DustCOMM can also be expanded to include more information about dust properties. For instance, a next step could be to include constraints on the dust vertical concentration profile

over every location, in order to more accurately estimate dust deposition, and dust concentration at the surface and in 3D. Another addition could be constraining the relative contribution of each dust source region to the 3D dust load, which can be combined with constraints on optical properties of dust emitted from each region (Di Biagio et al., 2017, 2019; Green et al., 2018) to obtain more accurate quantifications of dust radiative impacts.



## 6       Summary and Conclusions

In this study, we presented a new dataset of atmospheric dust aerosol properties called the 'Dust Constraints from Joint Observational-Modelling-experiMental Analysis' – DustCOMM. DustCOMM combines observational and experimental constraints on dust properties and abundance with an ensemble of global model simulations of dust spatial distribution to obtain more accurate 3-D annual and seasonal climatologies of dust properties and abundance than possible with global model simulations alone. Here, we presented three DustCOMM products: the three-dimensional (3-D) dust size distribution, 3-D dust mass extinction efficiency, and two-dimensional dust loading. First, we obtained constraints on the 3-D dust size distribution by combining constraints on the globally-averaged dust size distribution with an ensemble of model simulations of the spatial variability of the dust size distribution. Second, we combined the resulting 3-D dust size distribution with constraints on the size-resolved globally-averaged dust extinction efficiency, which accounts for the substantial asphericity of dust aerosols, to constrain the 3-D dust mass extinction efficiency. Finally, we used the resulting column-integrated dust mass extinction efficiency with an ensemble of reanalysis-derived dust aerosol optical depth to constrain the atmospheric dust column loading.

By comparing DustCOMM estimates of dust size distribution and dust mass extinction efficiency against independent *in-situ* measurements, we showed that DustCOMM reproduces observations substantially better than an ensemble of model simulations (Figs. 2-4, & 8). Models generally overestimate the contribution of fine-mode dust ($D \leq 2.5 \mu m$) and underestimate the contribution of coarse-mode dust ($D \geq 5 \mu m$), consistent with previous studies (e.g. Mahowald et al., 2014; Kok et al., 2017). In contrast, the DustCOMM size distribution is in substantially better agreement with measurements for different locations, heights and seasons over the $0.5 \leq D \leq 20$ $\mu m$ size range. However, there remain some discrepancies between DustCOMM and measurements, notably an underestimation of dust with $D \leq 0.5 \mu m$. Potential reasons for these discrepancies include contamination of measured dust size distribution by other aerosol species for $D \leq 0.5 \mu m$, and biases in observational and modelling constraints for $D \leq 0.5 \mu m$ (section 5.1). Overall, DustCOMM shows a bias against measured size distributions that is significantly less (about 46% less for $D \geq 0.5 \mu m$) than for an ensemble of global model simulations.

DustCOMM similarly shows better agreement against measurements of the dust mass extinction efficiency (MEE) than an ensemble of model estimates. Because DustCOMM predicts a coarser dust size distribution, as supported by the comparison against *in situ* size distribution measurements, it yields a global-mean dust MEE that is about 28% lower than that from the model ensemble, driven by large reductions in MEE over dust-dominated regions, where coarse particles dominate. For specific locations and seasons, DustCOMM estimates consistently show smaller errors relative to dust MEE measurements than an ensemble of model results, including in regions with strong spatial gradients in dust loading. On average, there is a negligible difference (~1%) between DustCOMM and measurements of MEE, while the model ensemble overestimates MEE by about 23% relative to measurements.

DustCOMM estimates of spatially-varying dust properties and abundance can be used to constrain various dust impacts on the Earth system in a manner that is more robust than possible with current global models. This is because





DustCOMM reproduces dust properties more accurately than global model simulations, and also because DustCOMM explicitly propagates uncertainties in experimental, observational, and modelling constraints used in obtaining the DustCOMM products, and these uncertainties can be propagated in calculations of dust impacts on global climate, biogeochemistry, and human health.

**List of some acronyms.**

| | |
|---|---|
| GISS | Goddard Institute for Space Studies (GISS) ModelE atmospheric general circulation model |
| WRF-Chem | Weather Research and Forecasting model coupled with Chemistry |
| CESM | Community Earth System Model |
| GEOS-Chem | Goddard Earth Observing System coupled with Chemistry |
| IMPACT | Integrated Massively Parallel Atmospheric Chemical Transport |
| INDOEX | Indian Ocean Experiment Intensive Field Phase |
| SHADE | Saharan Dust Experiment |
| ACE-Asia | Asian Pacific Regional Aerosol Characterization Experiment |
| TRACE-P | Transport and Chemical Evolution over the Pacific |
| ACE-2 | Aerosol Characterisation Experiment |
| DABEX | Dust and Biomass-burning Experiment |
| AMMA | African Monsoon Multidisciplinary Analysis |
| DODO | Dust Outflow and Deposition to the Ocean project |
| SAMUM | Saharan Mineral Dust Experiment |
| DARPO | Desert Aerosols over Portugal |
| BACEX | Barbados Aerosol Cloud Experiment |
| SALTRACE | Saharan Aerosol Long-Range Transport And Aerosol–Cloud-Interaction Experiment |
| AER-D | AERosol Properties – Dust |
| MERRA-2 | Modern-Era Retrospective analysis for Research and Applications, Version 2 |
| NAAPS | Navy Aerosol Analysis and Prediction System |
| JRAero | Japanese Reanalysis for Aerosol |
| CAMSiRA | Copernicus Atmosphere Monitoring Service interim Reanalysis |

(Line numbers in left margin: 10, 15, 20, 25)

30 **Data availability**

Annually-averaged and seasonally-averaged DustCOMM dust size distribution, dust mass extinction efficiency and dust loading are publicly available at http://doi.org/10.5281/zenodo.2620475. The model dust mass concentration, Kok et al. 2017 datasets, as well as other input dataset used to generate the DustCOMM dataset presented here can be found at  http://doi.org/10.5281/zenodo.2620547 (Adebiyi et al., 2019b).

**Code availability**

The codes used to generate DustCOMM size distribution, dust mass extinction efficiency and the atmospheric loading are available at http://doi.org/10.5281/zenodo.2620556.

40 **Author contribution.**





A.A.A and J.F.K designed the project. A.A.A performed the analysis and wrote the paper with contribution from J.F.K. Y.W helped with gathering the literature that reported dust size distribution, and A.I, P.N and C.Z contributed global model simulations outputs. D.A.R provided observationally-constrained dust aerosol optical depth data used in the analysis. All authors discussed the results and commented on the manuscript.

**Competing interests.**

The authors declare that they have no conflict of interest.

**Acknowledgements.**

We acknowledge support from National Science Foundation (NSF) grant 1552519 (J.F.K.). Chun Zhao is supported by the National Natural Science Foundation of China NSFC (Grant No. 41775146), and Akinori Ito is supported by the Japan Society for the Promotion of Science KAKENHI Grant Number JP16K00530 and Integrated Research Program for Advancing Climate Models (MEXT). The authors thank Claire L. Ryder for providing access to the Fennec and AER-D data which are both available at http://www.met.reading.ac.uk/~jp902366/research/. We also

thank Bruce Albrecht and Eunsil Jung for providing access to the size distribution collected  during the Barbados Aerosol Cloud Experiment (BACEX). The MERRA-2 reanalysis datasets are acquired from the NASA Goddard Earth Sciences (GES) Data and Information Services Center (DISC) and available at https://gmao.gsfc.nasa.gov/reanalysis/MERRA-2/. NAAPS data was obtained through the USGODAE Data Catalog at https://usgodae.org/cgi-bin/datalist.pl?generate=summary and CAMSiRA data was obtained through the ECMWF's

Meteorological Archiving and Retrieval System at http://apps.ecmwf.int/data-catalogues/cams-reanalysis. Finally, we thank Yumimoto, K. for providing access to the Japanese Reanalysis dataset through their website at https://www.riam.kyushu-u.ac.jp/taikai/JRAero/.

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



| Model | Particle size bins (diameter – μm) | Time Period | Spatial resolution | Meteorology used for simulation | Relevant reference |
|---|---|---|---|---|---|
| GISS | 0.2-0.36, 0.36-0.6, 0.6-1.2, 1.2-2.0, 2.0-4.0, 4.0-8.0, 8.0-16.0, 16.0-32.0 | 2004-2008 | 5° by 4° deg with 20 levels up to 0.1 hPa | Internal model meteorology | Miller et al. (2006) |
| WRF-Chem | 0.039-0.312, 0.312-0.625, 0.625-1.25, 1.25-2.5, 2.5-5.0, 5.0-10.0 | 2007-2016 | 1° by 1° deg with 35 levels up to 50 hPa. | NCEP/FNL reanalysis | Zhao et al. (2013) |
| CESM | 0.1-1.0, 1.0-2.5, 2.5-5.0, 5.0-10.0 | 2004-2008 | 2.5° by 1.89° deg with 56 levels up to 1.8 hPa. | ERA-Interim reanalysis | Hurrell et al. (2013) |
| GEOS-Chem | 0.2-0.36, 0.36-0.6, 0.6-1.2, 1.2-2.0, 2.0-3.6, 3.6-6.0, 6.0-12.0 | 2004-2008 | 2.5° by 2° deg with 47 levels up to 0.1 hPa. | MERRA reanalysis | See Kok et al., (2017) |
| ARPEGE-Climat | 0.1-0.2, 0.2-0.5, 0.5-1.0, 1.0-2.5, 2.5-10.0, 10.0-100.0 | 2004-2008 | 1.4° by 1.4° deg with 91 levels up to 10 hPa. | Internal model meteorology | Michou et al., 2015 |
| IMPACT | 0.1-1.26, 1.26-2.5, 2.5-5.0, 5.0-20.0 | 2004 | 2.0°×2.5° deg with 59 levels up to 0.02 hPa. | Meteorology from GEOS-5 model | Ito & Kok, 2017 |

**Table 1: Details of model simulations used in this study. Shown are the particle bin ranges, time periods of simulations, the spatial resolutions, the meteorology used, and the relevant model reference. We interpolate all model simulations to 2.5° by 1.9° to facilitate comparison and consistency with other datasets. See section 3 for details.**



| Measurement Reference | Project Name | Time Period | Representative Location | Comments |
|---|---|---|---|---|
| D'Almeida & Schutz, (1983) | N/A (Ground station) | Feb-Mar 1979; Jan-Feb1982 | Niamey (Niger): 14.21N,2.5E | PSD only; Z = 0-100m |
| Li et al., 1996 | N/A (Ground Station) | April-May 1994 | Barbados: 13.19N, 59.54W | MEE only; λ = 530 nm |
| Li et al., 2000 | N/A (Ground Station) | Oct-Nov, 1997 and Jan,1998 | Qinghai Province (China): 33.16N, 96.25E | MSE only; λ = 550 nm |
| Maring et al., 2000 | N/A (Ground Station) | July 1995 | Tenerife (Canary Island): 28.29N, 16.63W | MSE only; λ = 532 nm |
| Andreae et al., 2002 | ARACHNE | Dec, 1995 -Oct, 1997 | Sde Boker (Israel): 30N,34.79E | MSE only; λ = 550 nm |
| Quinn et al., 2002 | INDOEX | Feb- Mar, 1999 | Arabia Sea: 15N, 69E<br>Arabia Sea – Indian Ocean: 8N, 72E<br>Indian Ocean: 8S, 74E | MEE only; λ = 550 nm |
| Haywood et al., 2003 | SHADE | September 2000 | Cape Verde: 18N, 21W | MEE only; λ = 550 nm |
| Clarke et al. 2004 | ACE-Asia/TRACE-P | Feb.-Apr., 2001 | Sea of Japan: 38.85N, 130E | PSD and MSE; λ = 550 nm, Z = 0-6000m |
| Otto et al., 2007 | ACE-2 | Jun-Jul, 1997 | Canary Islands: 27.65N, 14.25W | PSD only; Z=2700m, 4000m, 5500m, 7000m |
| Osborne et al., 2008 | DABEX | Jan-Feb, 2006 | Niamey (Niger): 15.5N, 5.0E | MEE only; λ = 550 nm |
| Chou et al., 2008 | AMMA/DABEX | Jan-Feb, 2006 | Niamey (Niger): 15.5N, 5.0E | PSD only; Z = 0 – 1,500m |
| McConnell et al., 2008 | DODO-1 | Feb., 2006 | Dakar (Senegal): 14.76N, 17.38W | MEE only; λ = 550 nm |
| McConnell et al., 2008 | DODO-2 | August 2006 | Dakar (Senegal): 19.89N, 12.5W | PSD only; Z = 0 - 1000m |
| Weinzierl et al., 2009 | SAMUM-1 | May-Jun, 2006 | Morocco: 31.26N 7.5W | PSD only; Z = 3700-4900m |
| Wagner et al., 2009 | DARPO | May 2006 | Évora (Portugal): 38.57N 7.91 W | PSD only; Z = 2300-5000m |
| Kandler et al., 2009 | SAMUM-1 | May 2006 | Morocco: 31.26N 7.5W | PSD only; Z = 0-700 m |
| Kandler et al., 2011 | SAMUM-2 | Jan-Feb, 2008 | Praia (Cape Verde): 14.21N, 22.5W | PSD only; Z = 0-110m |
| Jung et al., 2013 | BACEX | Mar–Apr, 2010 | Barbados: 12.32N, 60W | PSD only; Z = 1250-2700m |



| Ryder et al., 2013 | Fennec 2011 | June, 2011 | Mauritania-Mali.: 24N, 6W | PSD and MEE; λ = 550 nm; Z = 0 – 6000m; |
|---|---|---|---|---|
| Weinzierl et al., 2017 | SALTRACE | June, 2013 | Cape Verde: 14.21N, 22.5W<br>Barbados: 13.19N, 59.54W | PSD only; Z=0-2600m |
| Ryder et al., 2018 | AER-D | August, 2015 | Cape Verde: 18N, 21W | PSD and MEE; λ = 550 nm; Z = 0-6000m |

1 **Table 2: Overview of in-situ measurements used to evaluate DustCOMM and model ensemble estimates of the dust size distribution and dust mass extinction efficiency (see**
2 **section 3.3 for details). The label PSD in the last column indicates that we take dust size distribution values from the study. MSE and MEE similarly indicates that we take**
3 **dust mass scattering or extinction efficiency values.**

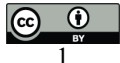



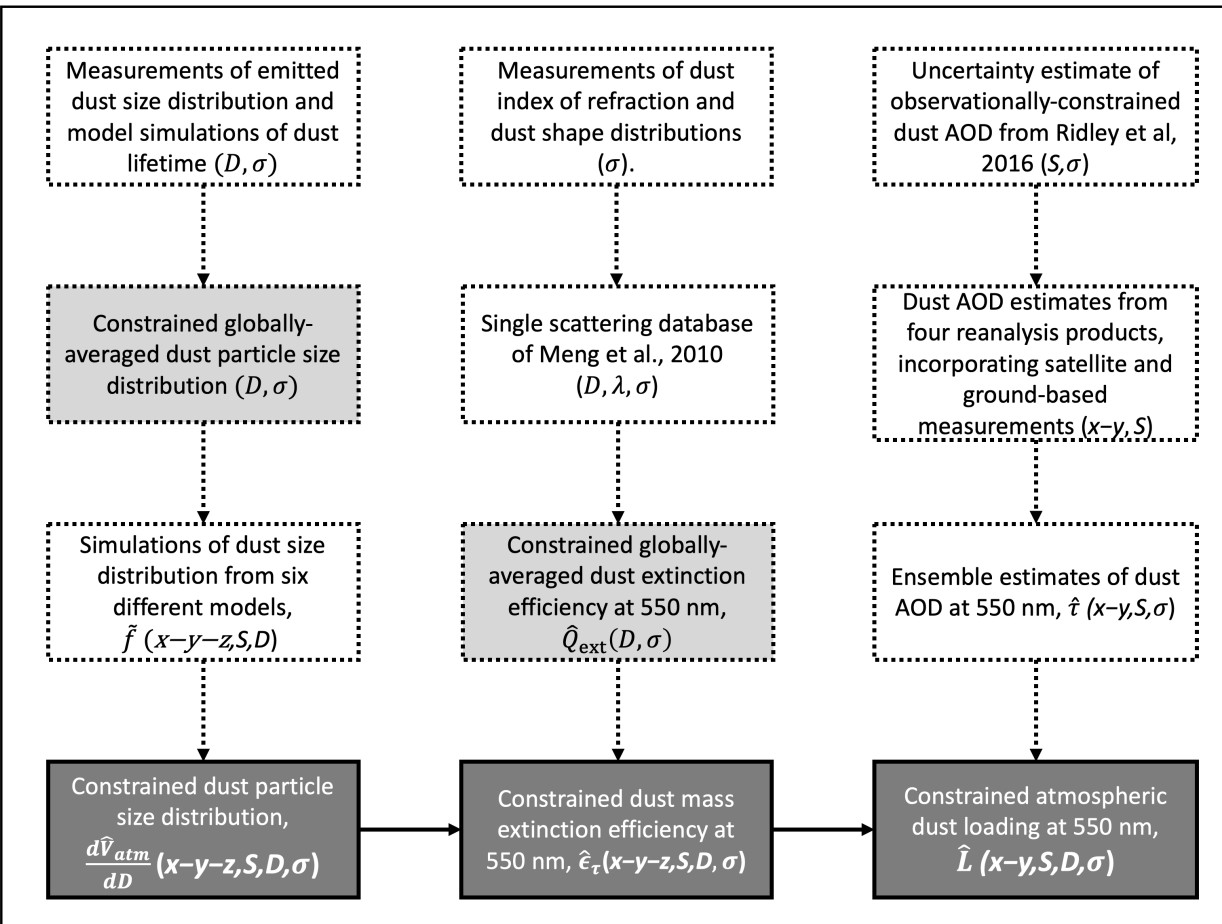

**Figure 1: Schematic of the key steps to obtain the DustCOMM products (dark shaded boxes): constraints on the 3-D dust size distribution, 3-D mass extinction efficiency and the 2-D atmospheric loading. The variables are a function of the following: x-y-z (three-dimensional field), x-y (two-dimensional field), S (seasonally-resolved), D (size-resolved), $\sigma$ (includes uncertainties). The variables in the light grey boxes are obtained from Kok et al (2017). See section 2 for details.**

3 **Figure 2: Comparison of normalized dust size distributions between published in-situ measurements (blue and purple dots; see**
4 **Table 2) and season-averaged DustCOMM (black lines) and model ensemble (red lines) estimates. The grey shading shows the 95%**
5 **confidence interval for the DustCOMM dust size distributions, whereas the pink shading shows the range of the model ensemble**
6 **size distributions. The size distributions are normalized between 2.5-10 µm. The comparisons are made at the nearest model grid-**
7 **points to the representative location and height level of the measurements.**



**Figure 3: Same as Figure 2 above, but as a function of height, which increases from bottom to top. The measurements plotted on the left panels are from Otto et al., (2007) taken during the ACE-2 campaign (June/July, 1997) off the west coast of Western Sahara and Morocco; the measurements plotted on the middle panels are from Ryder et al., (2013) taken during Fennec project (June 2011) between Canary Island and Mauritania/Mali; and the measurements plotted on the right panels are from Ryder et al., (2018) taken during the ACE-2 campaign in August 2015 near Cape Verde Island.**



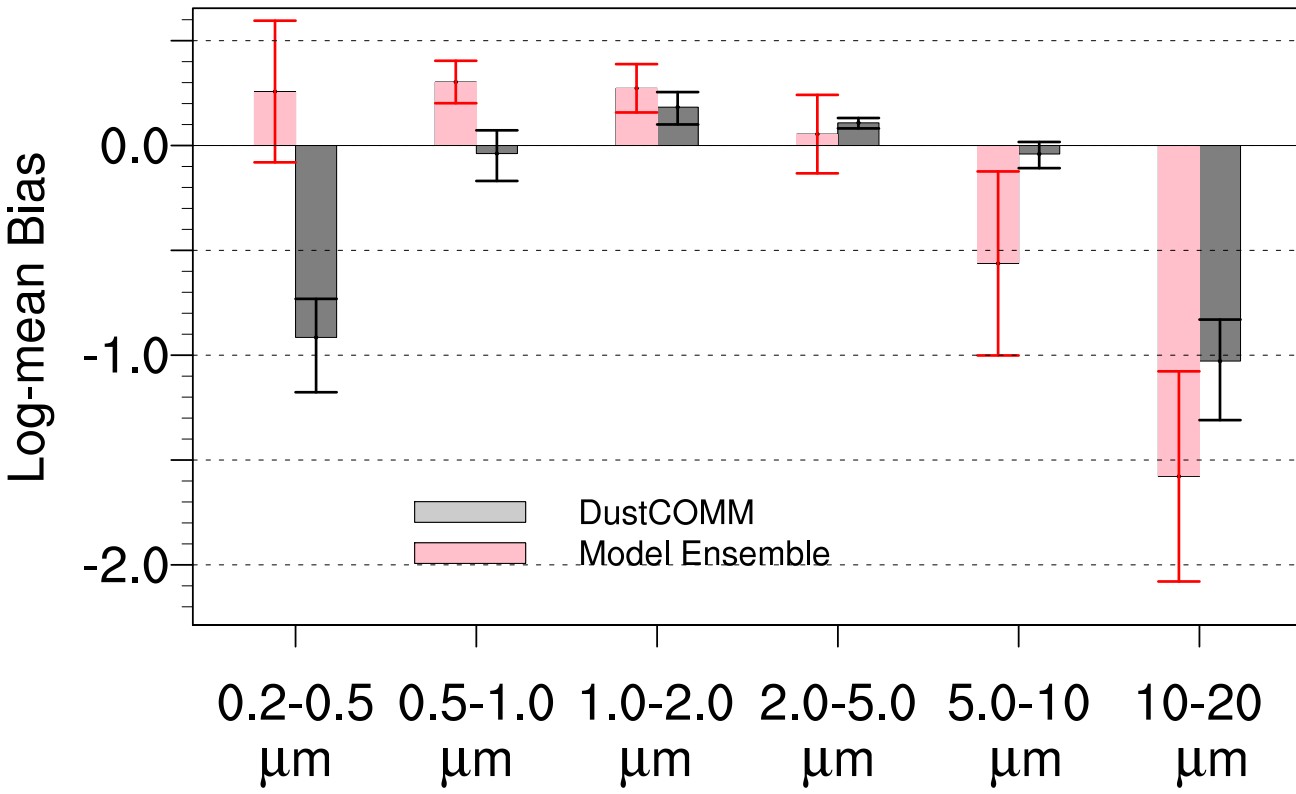

**Figure 4: Average log-mean bias between measurements and DustCOMM (grey) or model ensemble (pink) estimates of dust size distributions (shown in Figures 2 and 3), for different particle bins. The vertical bars represent the 95% confidence interval.**

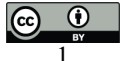

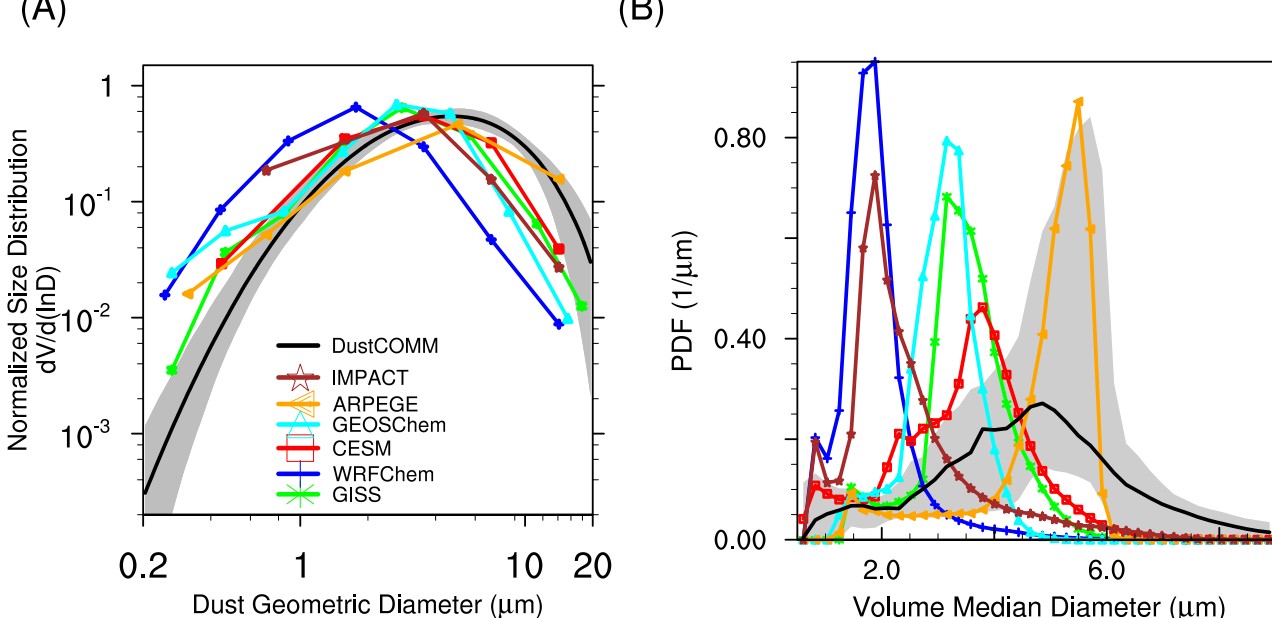

**Figure 5: (A) Comparison between DustCOMM (black line) and model simulations (colored lines) of the globally-averaged dust particle size distribution (PSD). The grey shading denotes the 95% confidence interval for the DustCOMM product. (B) The probability distribution of the volume median diameter (µm) of the PSD for DustCOMM (black line) and the individual model simulations (colored lines) over all locations and height levels.**



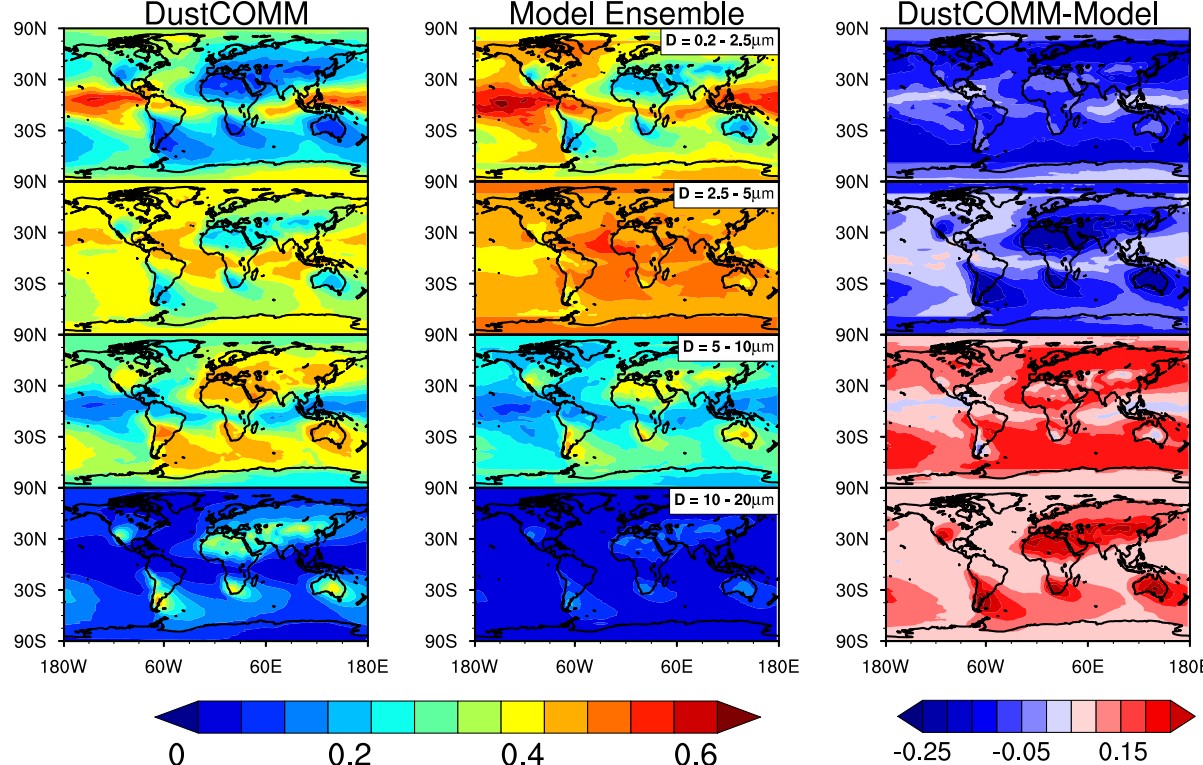

**Figure 6: Differences in the spatial variability of the dust mass fraction between DustCOMM and the model ensemble. Shown are the spatial distributions of the vertically-integrated dust mass fractions for different particle bins for DustCOMM (left panel), the model ensemble (middle panel), and the difference between the two (right panel; DustCOMM -Model Ensemble).**



**Figure 7: Vertical distribution of dust mass fractions as a function of particle size for the model ensemble (red lines) and DustCOMM (black lines). The grey and pink shading shows the 95% uncertainty confidence interval for DustCOMM and model ensemble, respectively.**

**Figure 8: Comparison of measurements (blue dots) of dust mass extinction efficiencies (MEE – m²g⁻¹) against column-integrated**
**DustCOMM (black bars) and model ensemble (red bars) estimates. Vertical bars on the measurements represent reported**
**uncertainty. For the DustCOMM and model ensemble estimates, the black and red boxes show one standard error, whereas the**
**vertical dotted lines show the 95% confidence interval; the middle horizontal bar and star shows the median and mean values,**
**respectively. The DustCOMM and model ensemble values are season-averaged values corresponding to the observation time period**
**(see Table 2 for details). These seasons are labelled DJF— Dec-Feb., MAM—Mar-May, JJA – Jun.-Jul., SON – Sep.-Nov; ANN**
**represents an annually-averaged value. The model ensemble MEE is calculated from the ratio between individual model dust aerosol**
**optical depth and the dust mass loading, while the DustCOMM MEE is calculated using the constrained dust size distributions and**
**single-particle extinction efficiency that takes into account the asphericity of dust aerosols. $\chi^2_\epsilon$ is the reduced chi-squared (Eqn. 10b),**
**and quantifies the performance of a model in representing observations (e.g. Andrae et al., 2010).**



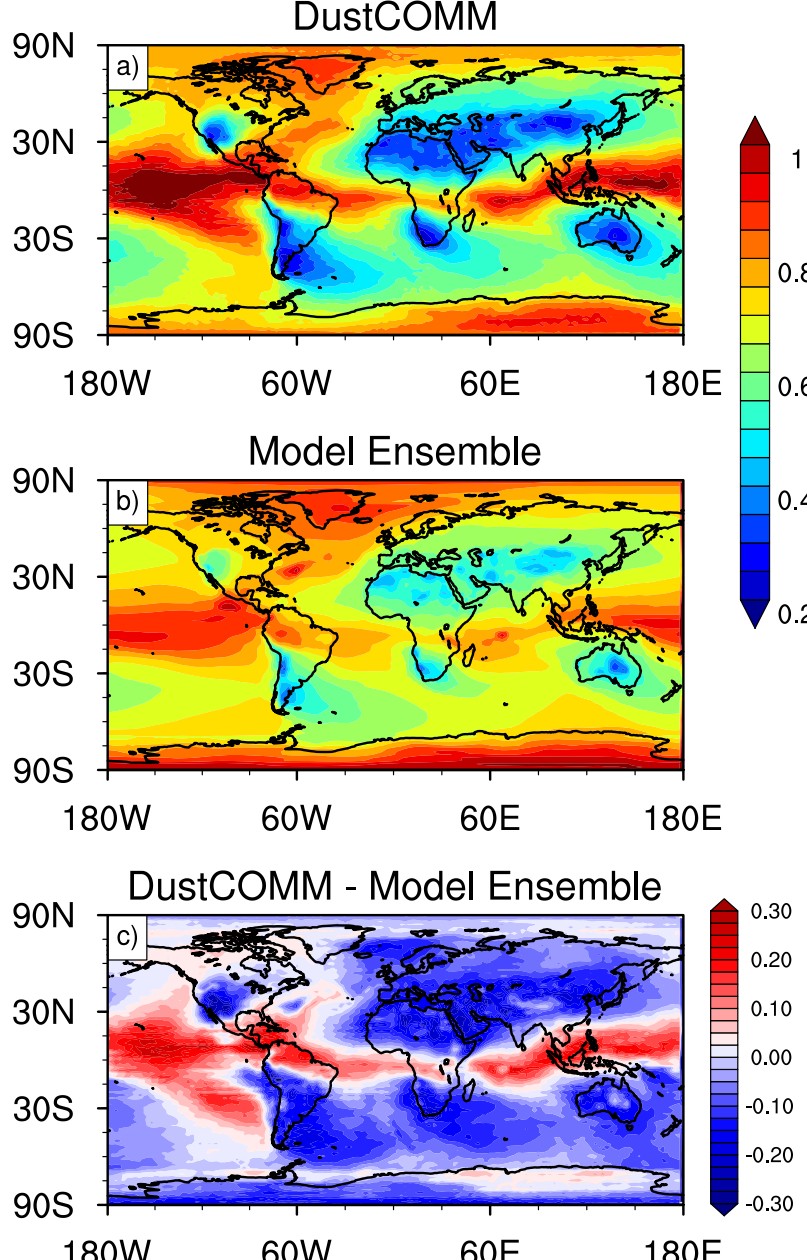

**Figure 9: Spatial distributions of column-integrated dust mass extinction efficiency (MEE – m²g⁻¹), weighted by the dust vertical distribution, for (a) DustCOMM, (b) the model ensemble, and (c) the difference between the two (DustCOMM -Model Ensemble).**

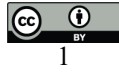



**Figure 10: Spatial distributions of the atmospheric dust column loading (g m⁻²) for the (a) DustCOMM and (b) model ensemble estimates, and (c) the difference between the two (DustCOMM -Model Ensemble).**





**Figure 11: Spatial distributions of DustCOMM relative uncertainties for (a-d) the dust mass fraction in the diameter ranges of $0.2 - 2.5\mu m$, $2.5 - 5\mu m$, $5 - 10\mu m$, and $10 - 20\mu m$; (e) the dust mass extinction efficiency (MEE); and (f) dust load. The relative uncertainties are calculated as the ratio of the uncertainty characterizing 68% of the distribution of each variable, divided by the mean value.**

