# Peer review of "Dust Constraints from joint Observational-ModellingexperiMental analysis (DustCOMM): Comparison with measurements and model simulations"

_Atmospheric Chemistry and Physics, 2019_

## Referee Comment (RC1) · Anonymous Referee #1 · 26 Jul 2019

**Review of "Dust Constraints from joint Observational-Modelling-experiMental analysis (DustCOMM): Comparison with measurements and model simulations" by Adebiyi et al.**

**General Comments**

A new dataset is presented and analysed to provide a global 3-D seasonally resolved dataset of dust properties with constrained size distribution, shape and refractive index properties. The dataset (DustCOMM) provides annual and seasonal mean 3-D dust size distributions, 3-D mass extinction efficiency at 550 nm wavelength and 2-D dust column mass loading. Since models are mostly unable to reproduce these properties in faith with measurements, this is a welcome and valuable step forwards in the dust field. The authors show that the DustCOMM results perform significantly better than global model simulations when evaluated against independent measurements. This is a notable achievement considering the challenges involved in representing global dust fields. The authors outline the benefits and potential future uses of the DustCOMM dataset to the community, which are likely to be substantial.

This paper covers a considerable amount of work and therefore methodology in order to deliver the final results in the DustCOMM dataset. Although the authors do a credible job of explaining the long and complicated methodology, in places it requires further explanation and clarity, potentially with re-ordering of some sections. The results impeccably described and presented. The description of measurements used requires some corrections and clarifications. Although the authors provide the datasets online, one of the links is broken. The article fits the scope of ACP and I recommend publication subject to the corrections detailed below.

**General Scientific Comments**

1. Overall, it is not clear why there is a need for the AOD reanalyses in this study – i.e. column 3 in Figure 1. Given the volume size distribution, and an assumed dust density, it should be possible to calculate column mass loading directly from size distribution without the requirement for AODs, and without any uncertainty involved in the mass extinction efficiency (MEE) values. Currently this is not explained, and therefore the AOD reanalysis section (fig 1 column 3 and sections 2.3) seems superfluous. More details are given below.

2. The authors do not include any dust properties in the longwave (LW) spectrum. The impact of the coarse particles on the radiative budget is one of the motivators of this study, and indeed highlighted in the discussions and conclusion as one of the benefits of the new dataset. The LW radiative effect makes up a large part of the total change in radiative effect due to a better representation of coarse particles (Kok et al., 2017). Thus the omission of LW dust properties here detracts slightly from the novelty of this work. There may be valid reasons for excluding the LW here, such as scope of material and lack of MEE observations in the LW spectrum. However, this should be discussed, since the total radiative impact of the constrained size distribution cannot be calculated without dust LW properties. Additionally, there may be some limitations in radiative studies which could be done with the DustCOMM data due to MEE being required spectrally (at least on some spectral resolution), rather than just at 550 nm as provided.

3. Section 2.1 (Constraining the 3-D atmospheric size distribution) is a crucial part of the paper, but difficult to follow. This section needs further explanation and clarity – see specific comments below.

**Specific Comments**

1. Links to datasets – the first asset link to the DustCOMM dataset v1 (https://doi.org/10.5281/zenodo.2620475) is broken.
2. Abstract – l17-21 – the use of model simulations should be mentioned here as this is a crucial part of the work.
3. It is not entirely clear what the benefits of model-constrained data presented here are over the data used in Kok et al. (2017). It would be useful to the reader to make this crystal clear, probably at the end of the introduction.
4. All the way through the article, but particularly in the method, the authors should be absolutely clear which size distribution they are referring to  - the Kok 2017 globally constrained size distribution, or the model-simulated size distribution(s) – when they state 'globally averaged size distribution,' or similar generalisations.
5. Section 5.4 – would you expect DustCOMM to show improved inter-annual variability in dust loading compared to conventional models, if more than one year of data were constructed? Or would DustCOMM's ability in this context be limited by the underlying global models' limitations? This is another significant challenge for models, e.g. Evan et al. (2014).
6. Section 2.1 – p4 l21-35 – These lines cover methodology and results from Kok et al. (2017). While necessary and useful to repeat here, I suggest making it very clear that up to l35 this is a repeat of method and size distribution from Kok et al. (2017).
7. Section 2, p5, l1-17 – this section forms a crucial part of the method – the main correction/constraining of the model size distributions – yet it is difficult to follow. The authors should explain this section much more clearly and in detail, perhaps with additional figures in the supplement and/or outlining a specific example, to clearly illustrate how the PSDs are forced away from the model simulations. Some specific comments are given in the points below, but I suggest they review and rephrase these lines.
8. It is not very clear at which point annual averages vs spatially varying size distributions are used from the simulations. E.g. p5 l5 – "annually-averaged, globally-averaged" size distributions are forced here – how/at which point are the spatially varying size distributions corrected? Better signposting of global averages vs spatial variations throughout the paper would be extremely useful in understanding the methodology.
9. P5 l9-10 "to the global dust loading" – which global dust loading – simulated or from Kok 2017?
10. P5 l1-17 – the description of equation 2 is not clear enough. E.g. what is the numerator in the equation for alpha?
11. P5 l16-17 – "where the discrete sum over each location and height equals unity, that is:…" – Why is the sum unity?  Is this because the size distributions are normalized?
12. Fig S1 – Please include the Kok 2017 globally averaged constrained size distribution on each panel. This would enable the reader to see what the size distribution is being constrained by.
13. Figure S1 – please also include (either in this plot, or as a separate one), how the size distributions changed due to the re-binning/extension/curtailing, as described in Section 2.1.1.
14. Section 2.1.1 – Is this diameter range correction performed after the size distribution correction (section 2.1)? Figure S1 suggests that first the diameter range correction is performed, and then the size distribution correction. But the ordering of the text (2.1 – size distribution correction, 2.1.1 – diameter range correction) suggests the opposite. Please clarify and order the text appropriately to follow the steps in the method.

15. P6 l1-3 – Ryder et al. (2019) show that over the Sahara D>20µm contribute to at least 18% of SW extinction and 26% of LW extinction – these values are not negligible and represent aged Saharan dust.

16. P6 l4, "dust particles with D>Dmax generally stay only for a short period in the atmosphere before they are deposited" – this is not the case in van der Does et al. (2018), as stated elsewhere in this manuscript.

17. P5 l1-6 – the authors should revisit the impact of particles d>dmax in the discussion (e.g. Section 5.4). For example, if better global constraints/observations on this size range became available, could such observations be incorporated into DustCOMM?

18. P6 l30-32 – the authors essentially extend 4 models' size distributions towards a larger size range based on the other 2, which cover a wider diameter range. Does this implicitly assume that all the models behave the same way in terms of the coarse end of the size distribution? This seems unlikely. This should be discussed more, particularly since many of the results are most sensitive to the size changes above d=10 microns

19. Equation 5 – please state how/if this equation is different to that from Kok et al. (2011), and if so why.

20. Equation 5 & p8 l6 – why choose $D_s$ for the geometric median diameter by volume? S subscript typically implies with respect to surface area. $D_v$ would be more appropriate.

21. P8 l20-27 – This seems a great generalization. It's not clear how b is applied to specific locations as implied.

22. Section 2.1.2 – What is the reason for choosing this method of fit (eqn 6) as opposed to fitting a series of lognormal modes, as is typically done for size distribution measurements? Presumably given the simulated size distributions have been corrected based on the same function, the fitting of the corrected size distribution is more naturally aligned with eqn 6?

23. P10, section 2.3, l26-30 – units for all quantities would be helpful. What do the authors mean by "mass-weighted" in "mass-weighted vertically-integrated 2-D mass extinction efficiency" and what are the units of epsilon_tau and epsilon_m?

24. P10, section 2.3, l26-30 – I believe this calculation is the same as first used by Kaufman et al. (2005), which should be cited.

25. P10 l31-32 – please list the reanalysis products (MERRA-2 etc) here to avoid confusion. Also see later comment about section ordering of 3.2. "Dataset" should be 'datasets.'

26. P10 l32-p11 l2 – "This individual reanalysis dataset…." - I suggest removing this (and adding to section 3.2 if necessary). It is confusing here given that the AOD reanalyses have not yet been described.

27. P11 l10 "the four data sets…" – this is also confusing given that the AOD datasets have not been properly introduced at this stage in the paper. See later comment about relocating section 3.2.

28. Ordering and section 3 – I suggest the authors move sections 3.1 and 3.2 to before section 2. This would be easier to follow and understand. Section 3.3 should remain after section 2 since it follows on logically.

29. P16 l35 – 2011-2015 is presumably limited by available years? Is there any impact of this difference in years used?

30. P17 l32 – and is also a 2-D diameter project of a 3-D shape, which may introduce bias (e.g. Chou et al., 2008).

31. P17 l 36 "separate channels for different particle sizes" - this is not really relevant and could introduce confusion.

32. P17 l25-40 – there is a 4th category, which covers imaging probes, as used in the AER-D field campaign (section 4.18 of supplement) – which are beneficial since they do not suffer from uncertainties in converting scattered light to size as OPCs do.

33. P18 l22-25 – OPCs have other sources of uncertainty – such as refractive index applied in the inversion of scattered light to size and the non-monotonic relationship between scattered light and particle size. These should also be mentioned.

34. P18 l11-25 and section 3.3.2 – in-cabin measurements are also subject to uncertainties and size-bias in sampling due to aircraft inlets. As such, the MEE values from studies in table 2 are likely biased high in some cases.

35. P18 l14 – although the size distribution measured does not allow aerosol type to be distinguished, various chemical composition measurements made in parallel are now mostly a matter of routine during airborne campaigns. Individual studies often use these to infer size distributions or ranges dominated by different aerosol types.

36. Section 3 – there is a huge variety of measurement data available, and I do not suggest the authors attempt to significantly widen their coverage. The authors should describe how and why the studies in Table 2 were selected. There also appears to be a geographical gap of sampling Arabian dust (Fig S1). Additionally, I suggest the studies of dust sampled during the AMMA airborne missions (Formenti et al., 2011) as being a very useful addition, since they provide summertime sampling in the Niger region, which is currently not covered by the studies in Table 2.

37. Table 2 – please indicate which studies relate to which numbers on the map on Figure S1.

38. Section 3.3.2 and Supplement section 4 – The descriptions of data taken from each measurement campaign are too vague, and occasionally in error. Often it is not enough just to reference a paper as within the measurement papers observations are collected/averaged in different ways (time periods, meteorological regimes, altitudes, etc.) and it is not clear which are being used here. The authors should state specifically which data are taken from each paper, and what the values of MEE or MSE are, preferably listing them in a table in the supplement. Specific comments about data described in the supplement are given in the Supplement section.

39. P18 l34-35 – Ryder et al. (2013b) SSA values fall well outside this SSA range. This is a fairly narrow SSA range selected. The authors should note that measured SSA is sensitive to the size range sampled in the observations, which is likely to exclude the coarse mode and often d>~2-3 microns in many cases due to the effects of inlets. Only 3 studies are cited, while there are a huge variety of studies in existence which have measured dust SSA.

40. P19 l9-11 – "These errors include errors due to the instrument measuring the extinction coefficient" – change to 'instrumental uncertainties.' "meteorological influence" – such as? "the assumption of internal or external mixing" – how is this important?

41. P20 l13-17 – Field campaigns additionally often sample a variety of cases which are representative of the within-season variability, and also often include uncertainties/ranges to cover the variability encountered.

42. P21 l2-3 "(1) the ACE-2 campaign (June/July, 1997) off the west coast of Western Sahara and Morocco (Otto et al., 2007)" – would be better referred to as in the vicinity of the Canary islands. Same for caption of Figure 3.

43. P21 l6 "(2) the Fennec project (June 2011) between the Canary Islands and Mauritania/Mali (Ryder et al., 2013)" – If the authors refer to measurements between the Canary Islands and Morocco/Western Sahara (not Mauritania/Mali which are inland) the citation should be Ryder et al. (2013a – GRL) and the geographic references corrected. The same applies to the caption of Figure 3.

44. Figure 3 – what is the reason for the selection of altitude choice? It seems biased very high – presence of dust at z>6km is unlikely and concentrations will be very low at 5.5km – therefore the value of such high altitude comparisons is questionable. What is the reason for the selection of these 3 studies for Figure 3? The geographic spacing is very close, with all sampling JJA SAL dust. "ACE-2" in line 7 of the caption should read "AER-D."

45. P22 l1 – 14% and 15% - in terms of which variable?

46. P23 l25 – and also Qinghai Province China?

47. P24 l17 – "weighted by the dust vertical distribution" – why is this necessary?

48. P24 l31-32 – as stated earlier, it is not clear why the MEE and AODs need to be used to calculate the column mass loading, given that this is typical model output, and the size distributions are already available. It should be a direct step to calculate column mass loading from size distributions, given a dust density.

49. Section 5.1 – are there any impacts of uncertainties in wet deposition on the size distribution biases?

50. P27 l30 – Fig 7a does not show MEE.

51. P27 l31 – there is no figure S7 in the supplement.

52. Section 5.4 – the implications of dust LW properties should be reflected on here, considering the points about the LW radiative impacts of dust being crucial to the total impact on the radiation balance described above.

53. P30 l20 – should 'indirect effects' be 'semi-direct effects'?

54. P30 l28 – some reference to the SW spectrum and 550 nm should be included, since refractive index and MEE are only considered at this wavelength.

55. Section 5 – There is a general focus on in-situ observations for validation of DustCOMM. However, remote sensing observations are developing rapidly and it would be useful for the authors to consider whether lidar retrievals, for example, would be usable within the DustCOMM framework.

56. P31 – l25-26 – the bias across the full size range should also be stated.

57. Are there any important dust altitude or seasonal changes in DustCOMM vs the models?

58. Discussion/Conclusion - It would be interesting if the authors could comment on bias of models vs measurements in previous studies (e.g. Hunneus et al., 2011; Evan et al., 2014), and similarities/improvements seen in those studies vs DustCOMM and the model simulations in this study.

59. AOD reanalyses – do the authors combine these into one single reanalysis dataset themselves? This is not really clear.

**Technical Comments**

P3 l12 – "The resulting product constrains the climatology of 3-D global atmospheric dust properties on seasonal and annual timescales" – change to "The resulting product constrains the climatology of 3-D global atmospheric dust properties **and is provided** on seasonal and annual timescales" – to avoid confusion that the authors are constraining the temporal variability of dust properties.

P3 l34-35 – "After correcting…" – unclear – do you mean you combine all models into one multi-model representation?

P6 l28 – "globally-averaged size distribution" – Kok 2017 or the simulated one?

P7 l13 - "globally-averaged size distribution" – Kok 2017 or the simulated one?

P9 l18 – typo – should be -10 to -4?

Eqn 8 – please provide units for epsilon_tau

P10 l27 – "atmospheric "column" dust loading"?

P14 l25 – change to "…of the in-situ emission measurements.."

**Supplement Comments**

1. The supplement contains two Figure S1s. The second should be S3 (?).
2. Section 3.1 – l7-8 – mention that it is the AOD which is assimilated.
3. Section 3.3 - "1.1ox1.1o" typo
4. Section 4 – To make this easier to navigate, relate each observational subheading to the numbers on fig S3 (map). Also include the campaign name in the heading for each section. Take care to state for each subheading whether the campaign was ground-based or airborne. Also explain the choice of altitude selection defined in table to where relevant. Please also be aware, and state where necessary, that although a large size range may have been measured, inlet-size effects may have prevented coarser particles from being measured for some campaigns.
5. Section 4 – a subsection on Kandler et al. (2009), as listed in table 2, is missing.
6. Section 4.1 – please make the locations listed consistent with those listed in table 2.
7. Section 4.7 – note that these size distributions were not corrected for refractive index. The FSSP was *not* used as it did not operate correctly. Instead the size distribution larger than d=3 microns was taken from a sunphotometer retrieval.
8. Section 4.10 – Please note that these studies operated instruments behind significant pipework and suffered loss of the majority of coarse particles (e.g. Ryder et al., 2018, Table 1).
9. Section 4.11 – Why is MEE only taken from DODO1 (winter time?). It appears that the MEE for DODO1 is taken from table 4 of Osborne et al. (2008), for the 'AM+CM' case (a value of 0.41). No coarse mode was measured during DODO1 (see McConnell et al., 2008). The AM+CM DODO1 case in Osborne et al. (2008) was calculated using the coarse mode size distribution from DABEX since none was available from DODO1. This should be stated, or preferably the value from DODO2 used, where coarse mode was measured. Why is only z<1km used for the DODO2 size distributions?
10. Section 4.12 – is the campaign average size distribution used?
11. Section 4.13 – and also same aircraft as SAMUM1? SAMUM1 also used a high spectral resolution lidar.
12. Section 4.14 – 'used the same instrumentation…' – as which paper/campaign? Presumably the same as Kandler et al. (2009) which is missing? It is not clear which instruments the size distribution comes from – but probably because the 2009 section is missing.
13. Section 4.15 – Data from Ryder et al. (2013a – GRL, Canary Islands) is also used in the paper (Figure 3) and should be described here. Please take care to specify whether Ryder et al. (2013a or 2013b) is being cited – both are given in the references as 2013. Comparisons of both can be found in Ryder et al. (2019). "For this study, mean distribution from PCASP and CDP were selected because they were the most credible based on the authors' analysis." – change to …"based on the authors' analysis over the size range we use here." MEE is not given in Ryder et al. (2013b - ACP), presumably this is taken from Ryder et al. (2013a) (please state). Mean values in Ryder et al. (2013a) are 0.15 for fresh dust or 0.23 for aged dust – these appear much lower than the value plotted in Figure 8 (around 0.3).

14. Section 4.16 "above the SAL" – I would expect a dust measurement to be taken 'in' the SAL – is this a typo?
15. Section 4.18 – "The AER-D campaign uses similar instrument as the Fennec 2011 campaign. They use wing-mounted optical particle counters and shadow probes to measure dust sizes between 0.1 and 100 µm diameter." – but additionally this AER-D used cloud imaging probes (CIP15 and 2DS) for size distributions at d>10 microns (which were used in Fennec but were not mentioned in Section 4.15 as the authors did not use the shadow probe data (d>18.5 microns) in this study).

**References**

Evan, A. T., C. Flamant, S. Fiedler, and O. Doherty (2014), An analysis of Aeolian dust in climate models, Geophys. Res. Lett., 41, doi:10.1002/2014GL060545.

Formenti, P., Rajot, J. L., Desboeufs, K., Saïd, F., Grand, N., Chevaillier, S., and Schmechtig, C.: Airborne observations of mineral dust over western Africa in the summer Monsoon season: spatial and vertical variability of physico-chemical and optical properties, Atmos. Chem. Phys., 11, 6387-6410, https://doi.org/10.5194/acp-11-6387-2011, 2011.

Huneeus, N., Schulz, M., Balkanski, Y., Griesfeller, J., Prospero, J., Kinne, S., Bauer, S., Boucher, O., Chin, M., Dentener, F., Diehl, T., Easter, R., Fillmore, D., Ghan, S., Ginoux, P., Grini, A., Horowitz, L., Koch, D., Krol, M. C., Landing, W., Liu, X., Mahowald, N., Miller, R., Morcrette, J.-J., Myhre, G., Penner, J., Perlwitz, J., Stier, P., Takemura, T., and Zender, C. S.: Global dust model intercomparison in AeroCom phase I, Atmos. Chem. Phys., 11, 7781-7816, https://doi.org/10.5194/acp-11-7781-2011, 2011.

Kaufman Y J, Koren I, Remer L A, Tanr´e D, Ginoux P and Fan S 2005 Dust transport and deposition bserved from the terra-moderate resolution imaging spectroradiometer (MODIS) spacecraft over the Atlantic Ocean J. Geophys. Res. 110 D10S12

Ryder, C. L., Highwood, E. J., Walser, A., Seibert, P., Philipp, A., and Weinzierl, B.: Coarse and Giant Particles are Ubiquitous in Saharan Dust Export Regions and are Radiatively Significant over the Sahara, Atmos. Chem. Phys. Discuss., https://doi.org/10.5194/acp-2019-421, in review, 2019.

---

## Referee Comment (RC2) · Anonymous Referee #2 · 9 Oct 2019

This study presents a new dataset, the Dust Constraints from joint Observational-Modelling-experiMental analysis (DustCOMM), which combines in-situ measurements, reanalysis products, and an ensemble of six global model simulations. Particularly, globally-averaged dust size distribution and extinction efficiency from observational and experimental data are used to constrain the DustCOMM products. The annual and seasonal mean products of 3-dimensional (3D) dust size distribution, 3D dust mass extinction efficiency, and 2D dust loading are provided for the time period from 2004 to 2008. It is found the dataset shows a better agreement with measurements than the

six-model ensemble in terms of dust size distribution and mass extinction efficiency. This dataset may be used to constrain dust simulation in global models and to study dust impacts on the earth system. The paper is generally well written. The methodology to develop the datasets is thoroughly introduced and related uncertainties are also discussed in detail. I have a few comments would like the authors to address.

Major comments:

1. Here globally-averaged dust size distribution is used to obtain 3D dust size distribution. Is it possible to demonstrate that the regional differences in dust size distribution are small? Or have you considered using different dust size distribution for different regions, e.g., by applying regional averaged values to areas where individual measurements are available and the globally-averaged value to areas where measurements are not available? This might provide better spatial constraints on the dataset. Similarly, globally-averaged dust extinction efficiency at 550 nm is used. How large are the spatial differences? Is it possible to give a rough estimation based on available data?

2. As discussed in the paper, dust aerosol optical depth from the reanalyses largely depends on the models' treatment of the dust cycle, and this adds uncertainties to the DustCOMM. I wonder if you considered using satellite products of dust optical depth, such as level 3 dust optical depth from the Cloud-Aerosol Lidar with Orthogonal Polarization (CALIOP).

3. Sections 4-5 show that the new dataset has a better agreement with in-situ measurements than the multi-model mean. I think it is better to add some discussion to emphasize why this dataset is a good complement to the currently available observational data, especially individual measurements. For instance, the global coverage and vertical distribution of the dust size distribution and mass extinction efficiency of the dataset make it easier to be adapted to global models to constrain simulations or to study global dust impacts.

Minor comments:

1. Lines 21-23, page 2, this can be a bit misleading since both small are large dust particles absorb and scatter shortwave and longwave radiation.

2. Line 6, page 3, "To address this problem", not sure the dataset would be able to address the "numerous important biases". You may want to point out a few detailed problems.

3. You may want to add the horizontal and vertical resolutions of the DustCOMM product at someplace in Section 2.

4. Line 17, page 11, what time period does the "climatology" refer to?

5. Section 3.1, are all the model results interpolated to the same horizontal and vertical grids? And what's the resolution?

6. Line 35, page 16, why the JRAero in a different time period is used? It's not available from 2004 to 2008?

7. Line 16, page 21, do you refer to Fig. 4 instead of Fig. S4?

8. Line 16-18, page 21, can you please add some discussion about why the Dust-COMM has a larger bias than model ensemble for $D \leq 0.5$ $\mu$m?

9. Line 19-24, page 23, "...regardless of the season and location", except Sde Boker, Israel.

10. Table 1, please remove "deg" in column four, since you already added a degree symbol there.

11. Figs. 2-3, can you please add latitude, longitude, and location of the measurements on the top of each plot? Or you may number the measurements listed in Table 2 and then simply list the corresponding numbers in the figure.

12. Fig. 5, is it possible to add a globally averaged PSD and its PDF to the plot?

13. Fig. 6, why is dust mass fraction for D= 0.2-2.5 $\mu$m high over the ITCZ? Is this

consistent with observations?

14. Fig. 7, it would be more interesting to show individual model results (as in Fig. 5) instead of multi-model results.

15. Fig. 8, why do some blue dots have a light blue outline?

---

## Author Response (AR1)

We thank both reviewers and the editor for their constructive comments, which has greatly helped us to improve the paper. Below we include a point-by point response to the referee comments, and describe the corresponding changes we have made to the manuscript

Comments from reviewer #1

**General Comments**

A new dataset is presented and analysed to provide a global 3-D seasonally resolved dataset of dust properties with constrained size distribution, shape and refractive index properties. The dataset (DustCOMM) provides annual and seasonal mean 3-D dust size distributions, 3-D mass extinction efficiency at 550 nm wavelength and 2-D dust column mass loading. Since models are mostly unable to reproduce these properties in faith with measurements, this is a welcome and valuable step forwards in the dust field. The authors show that the DustCOMM results perform significantly better than global model simulations when evaluated against independent measurements. This is a notable achievement considering the challenges involved in representing global dust fields. The authors outline the benefits and potential future uses of the DustCOMM dataset to the community, which are likely to be substantial.

This paper covers a considerable amount of work and therefore methodology in order to deliver the final results in the DustCOMM dataset. Although the authors do a credible job of explaining the long and complicated methodology, in places it requires further explanation and clarity, potentially with re-ordering of some sections. The results impeccably described and presented. The description of measurements used requires some corrections and clarifications. Although the authors provide the datasets online, one of the links is broken. The article fits the scope of ACP and I recommend publication subject to the corrections detailed below.

We thank the reviewer for the constructive and helpful comments that helped us to further improve the paper. We address these comments below.

**General Scientific Comments**

1. Overall, it is not clear why there is a need for the AOD reanalyses in this study – i.e. column 3 in Figure 1. Given the volume size distribution, and an assumed dust density, it should be possible to calculate column mass loading directly from size distribution without the requirement for AODs, and without any uncertainty involved in the mass extinction efficiency (MEE) values. Currently this is not explained, and therefore the AOD reanalysis section (fig 1 column 3 and sections 2.3) seems superfluous. More details are given below.

   We thank the reviewer for this comment. We note that our dust size distributions are normalized such that the integral over the diameter range is unity, and with no information about the size-resolved volume of the particles. Therefore, the normalized

dust size distribution cannot be used directly with dust density to calculate the column dust loading. We have added the following sentence in section 2.3.3 (formerly section 2.3) to clarify this point: "Since our constraints on dust size distributions are normalized to unity, and also to ensure that our estimates of dust loading produce the same extinction as those from reanalysis dataset or satellite measurements, we use this approach to estimate the atmospheric dust loading…"

2.  The authors do not include any dust properties in the longwave (LW) spectrum. The impact of the coarse particles on the radiative budget is one of the motivators of this study, and indeed highlighted in the discussions and conclusion as one of the benefits of the new dataset. The LW radiative effect makes up a large part of the total change in radiative effect due to a better representation of coarse particles (Kok et al., 2017). Thus the omission of LW dust properties here detracts slightly from the novelty of this work. There may be valid reasons for excluding the LW here, such as scope of material and lack of MEE observations in the LW spectrum. However, this should be discussed, since the total radiative impact of the constrained size distribution cannot be calculated without dust LW properties. Additionally, there may be some limitations in radiative studies which could be done with the DustCOMM data due to MEE being required spectrally (at least on some spectral resolution), rather than just at 550 nm as provided.

The reviewer raises an excellent point here, as indeed a large portion of the uncertainty in dust radiative effects is due to uncertainty in LW interactions. That being said, we provide dust optical properties at the mid-visible wavelength (550 nm), rather than in the longwave spectrum, for several reasons. First, as the reviewer pointed out, measurements of dust mass extinction efficiency (MEE) in longwave are scarce and therefore it will be difficult to validate our constraints on dust MEE. Second, the estimation of dust mass loading in Eqn. 9 requires MEE value at 550 nm which is the same wavelength as the observational constraints on dust optical depth (e.g. Ridley et al., 2016). Third, the measurements of dust refractive index needed to constrain the single-particle extinction efficiency at the longwave spectrum are also scarce, potentially leading to large uncertainties in constraining the dust MEE. Despite these reasons, our future studies will focus on incorporating the available measurements of dust refraction index – such as those from Di Biagio et al. (2017, 2019) – to constrain the spectral dependence of dust optical properties for both longwave and shortwave spectra as part of DustCOMM dataset.

To clarify these reasons for not including dust properties in the LW spectrum in this paper, we have included a paragraph in section 2.3.2. "We use this constrained globally-averaged $\widehat{Q}_{ext}$ to constrain $\hat{\epsilon}_\tau$ (Eqn. 8) for every location. We thus neglect any regional variation in $\widehat{Q}_{ext}$ because measurements of dust shapes and index of refraction are currently insufficient to constrain $\hat{\epsilon}_\tau$ on a regional basis. In addition, since measurements of dust refractive index needed to constrain $\hat{\epsilon}_\tau$ at other wavelengths are also scarce, we

limit our estimate here only to the 550 nm wavelength. We use 550 nm as the wavelength of choice because measurements to validate our estimate of $\hat{\epsilon}_\tau$ and the observational constraints to estimate the dust atmospheric loading are mostly available at mid-visible wavelength."

3. Section 2.1 (Constraining the 3-D atmospheric size distribution) is a crucial part of the paper, but difficult to follow. This section needs further explanation and clarity – see specific comments below.

   Thank you for pointing this out. Based on the specific comments by the reviewer, we have reordered both sections 2 and 3 to make it easier for the reader to understand. For example, in addition to other specific changes below, we now have the description of the model simulation and reanalysis datasets before the methodology that constrains the dust size distribution, mass extinction efficiency and the atmospheric loading.

**Specific Comments**

1. Links to datasets – the first asset link to the DustCOMM dataset v1 (https://doi.org/10.5281/zenodo.2620475) is broken.

   Thank you for pointing this out. The link has been fixed.

2. Abstract – l 17-21 – the use of model simulations should be mentioned here as this is a crucial part of the work.

   Thank you, that's a good point. We have re-written this part to reflect the use of global model simulations in constraining the dust size distribution. The new sentence now reads as:

   "This dataset leverages an ensemble of global model simulations with observational and experimental constraints on dust size distribution and shape to obtain more accurate constraints on three-dimensional (3-D) atmospheric dust properties than is possible from global model simulations alone."

3. It is not entirely clear what the benefits of model-constrained data presented here are over the data used in Kok et al. (2017). It would be useful to the reader to make this crystal clear, probably at the end of the introduction.

   We have included a sentence at the end of the introduction to clarify the difference between the Kok et al. (2017) results and the results in the study. We added the following to the last paragraph: "DustCOMM builds on the results from Kok et al. (2017), however, unlike the globally-averaged results obtained in Kok et al. (2017), our product constrains

the climatology of 3-D global atmospheric dust properties on seasonal and annual timescales".

4. All the way through the article, but particularly in the method, the authors should be absolutely clear which size distribution they are referring to - the Kok 2017 globally constrained size distribution, or the model-simulated size distribution(s) – when they state 'globally averaged size distribution,' or similar generalizations.

Thank you for the comments. We have clarified this confusion where necessary.

5. Section 5.4 – would you expect DustCOMM to show improved inter-annual variability in dust loading compared to conventional models, if more than one year of data were constructed? Or would DustCOMM's ability in this context be limited by the underlying global models' limitations? This is another significant challenge for models, e.g. Evan et al. (2014).

Thank you for the question. While we have not looked at this topic yet, we expect that the inter-annual variability in DustCOMM dust size distributions largely depends on the ensemble of model simulations used. However, the interannual variability of the dust mass loading will include not only the variability from the ensemble of model simulations but also from the observations used to constrain the dust aerosol optical depth and dust extinction. Although this study only present seasonal variability, DustCOMM inter-annual variability is expected to show some improvements when compared to individual model simulation.

6. Section 2.1 – p4 l21-35 – These lines cover methodology and results from Kok et al. (2017). While necessary and useful to repeat here, I suggest making it very clear that up to l 35 this is a repeat of method and size distribution from Kok et al. (2017).

To improve clarity for this section, we have added an extra sentence before Eqn. 1, and re-write the paragraph following the same equation.

"While details can be found in Kok et al. (2017), a summary of their globally-averaged size distribution is given here as:"

"As reported in Kok et al. (2017), the constrained globally-averaged size distribution of emitted dust particles, $\left[\frac{d\hat{V}_{emit}(D)}{dD}\right]_g$, is based on an analysis of different measurements of the emitted dust size distribution, while the size-resolved globally-averaged dust lifetime, $\left[\tilde{T}(D)\right]_g$, is based on an ensemble of global model simulations"

7. Section 2, p5, l1-17 – this section forms a crucial part of the method – the main correction/constraining of the model size distributions – yet it is difficult to follow. The authors should explain this section much more clearly and in detail, perhaps with additional figures in the supplement and/or outlining a specific example, to clearly illustrate how the PSDs are forced away from the model simulations. Some specific comments are given in the points below, but I suggest they review and rephrase these lines.

   We have re-written the paragraph before Eqn. 2 to better clarify the procedure. After Equation 2, we have also made additional clarifications to the sentences, including clearly explaining the dimensions x, y, and z. For this part, we added "…;x is the dimension for longitude, y is for latitude and z is for height."

8. It is not very clear at which point annual averages vs spatially varying size distributions are used from the simulations. E.g. p5 l5 – "annually-averaged, globally-averaged" size distributions are forced here – how/at which point are the spatially varying size distributions corrected? Better signposting of global averages vs spatial variations throughout the paper would be extremely useful in understanding the methodology.

   We made this distinction by specifying in each formula the independent variable of each parameter. For example, $\hat{f}_{k,i}(x, y, z, D_{k,i})$ in Eqn. 2 is a spatially-varying parameter defined for x-longitude, y-latitude and z-height, while the $\left[\bar{\bar{\hat{f}}}_{k,i}(D_{k,i})\right]_g$ is the globally-averaged counterpart denoting that the parameter is averaged over all space (x-y-z). To make this clearer, we have explicitly defined the independent variable following its first use in Eqn. 2., adding "x is the dimension for longitude, y is for latitude and z is for height"

9. P5 l9-10 "to the global dust loading" – which global dust loading – simulated or from Kok 2017?

   The sentence has been re-written for clarity. It now reads "This correction factor ($\alpha$) is defined by the ratio of the Kok et al. (2017) constraint on the fractional contribution of the particle bin to the simulated fractional contribution of the particle bin per unit global dust loading."

10. P5 l1-17 – the description of equation 2 is not clear enough. E.g. what is the numerator in the equation for alpha?

    The numerator is the fractional contribution of the Kok et al. 2017 globally-averaged size distribution defined between diameter $D_{k,i+}$ and $D_{k,i+}$. In addition to the description of $\alpha$ before Eqn. 2, we have added the following statement to improve the clarity after the

equation.: "the numerator, $\int_{D_{k,i-}}^{D_{k,i+}} \left[\frac{d\hat{V}(D)}{dD}\right]_g dD$, is the constraint obtained from Kok et al. (2017)"

11. P5 l16-17 – "where the discrete sum over each location and height equals unity, that is:..." – Why is the sum unity? Is this because the size distributions are normalized?

Yes, that's indeed correct. This is a key point, so we have also rewritten this for clarity.

12. Fig S1 – Please include the Kok 2017 globally averaged constrained size distribution on each panel. This would enable the reader to see what the size distribution is being constrained by.

That's a good idea. Since DustCOMM is forced to the Kok et al 2017 size distribution, both DustCOMM and Kok et al 2017 are the same and thus overlap in the Figure S1. Although not always visible, we have nonetheless included the Kok et al 2017 size distribution in Fig. S1 for completeness (see below).

[Figure]

13. Figure S1 – please also include (either in this plot, or as a separate one), how the size distributions changed due to the re-binning/extension/curtailing, as described in Section 2.1.1.

We have included the figure below in the supplementary document

[Figure]

14. Section 2.1.1 – Is this diameter range correction performed after the size distribution correction (section 2.1)? Figure S1 suggests that first the diameter range correction is performed, and then the size distribution correction. But the ordering of the text (2.1 – size distribution correction, 2.1.1 – diameter range correction) suggests the opposite. Please clarify and order the text appropriately to follow the steps in the method.

This is indeed confusing. The order described in section 2.3.1 (formerly 2.1) does follow the steps taken in the methodology. That is, the dust mass fractions were corrected and then the size ranges were set to the common diameter limits. To make this clearer, we have added a statement that the dust mass fractions were re-normalized after the correction, such that the discrete sum still equals unity over each location. This statement at the end of section 2.1.1 reads: "After the dust mass fractions are corrected, they are re-normalized such that the discrete sum between $D_{min}$ and $D_{max}$ equals unity over each location and height."

For the 'uncorrected' model dust mass fractions, we highlight in section 2.1 (formerly 3.1) that, for consistency, they are also set to the same diameter limits between 0.2 and 20μm following the same procedure described in section 2.3.1.1 (formerly 2.1.1).

15. P6 l1-3 – Ryder et al. (2019) show that over the Sahara D>20μm contribute to at least 18% of SW extinction and 26% of LW extinction – these values are not negligible and represent aged Saharan dust.

   P6 l4, "dust particles with D>Dmax generally stay only for a short period in the atmosphere before they are deposited" – this is not the case in van der Does et al. (2018), as stated elsewhere in this manuscript.

   That's a good point – those two recent papers indicate that the community has greatly underestimated the effect of dust with D > 20 um. In this study, we need to limit the maximum diameter to 20 µm because results from Kok et al. 2017 used to constrain our dust size distribution is limited to this diameter. But to better acknowledge that particles greater than 20 µm might be important even farther from source regions (e.g. Ryder et al. (2019)), we have re-written this part of the paragraph:

   "Further, we set the upper diameter limit to $D_{max} = 20$ µm, because most global models generally do not incorporate dust particles beyond 20 µm and also because the observational constraints on the size distribution from Kok et al. (2017) is limited to this maximum diameter. Although advances in airborne observations in recent years have led to measurements of larger dust particles with $D > D_{max}$ in the atmosphere which has shown that the contribution of $D > 20$ µm to shortwave and longwave extinction are non-negligible (e.g. Ryder et al., 2013b, 2019; Weinzierl et al., 2009, 2017), there is still a scarcity of these measurements, such that an observational constraint on dust particles with $D > D_{max}$ would be very uncertain (e.g. Mahowald et al., 2014)."

16. P5 l1-6 – the authors should revisit the impact of particles d>dmax in the discussion (e.g. Section 5.4). For example, if better global constraints/observations on this size range became available, could such observations be incorporated into DustCOMM? \

   Yes, that's a good point. We have added the sentence below in section 4.4 (formerly 5.4) to clarify this point.

   "Given that dust particles with $D \geq 20$ µm can contribute substantially to dust extinction both in the shortwave and longwave spectrum (Ryder et al., 2019), future versions of DustCOMM could be extended to a diameter range beyond 20 µm as more measurements of dust size distribution with $D \geq 20$ µm become available."

17. P6 l30-32 – the authors essentially extend 4 models' size distributions towards a larger size range based on the other 2, which cover a wider diameter range. Does this implicitly assume that all the models behave the same way in terms of the coarse end of the size distribution? This seems unlikely. This should be discussed more, particularly since many of the results are most sensitive to the size changes above d=10 microns

No, our constraint does not implicitly assume that the models behave the same way. Since dry depositions in these models are controlled largely by gravitational settling, the rates of deposition of the dust particles are often different. Our constraints thus account for these differences by taking into account the differences in the spatial variability of the bin that overlaps between the two models. That is, in correcting for model simulation k in Equation 4a (for example), we account for the difference between the bins $[D_{k,N_k-}, D_{k,N_k+}]$ and $[D_{r,j_r-}, D_{r,j_r+}]$ in model r, which partially overlap with each other (Eqn. 4b). Although the correction factor $\beta_r(x, y, z)$ is expected to take into account the differences between model k and model r, we however still assume that the distribution largely follow the same form controlled by the rate of dust deposition in each model. We have added the sentence below to clarify this point.

"It should be noted that the correction of Eqn. 4 takes into account the potential difference in the dust deposition between models k and r, by considering the differences in the spatial variability of dust loading between similar bins of $[D_{k,N_k-}, D_{k,N_k+}]$ and $[D_{r,j_r-}, D_{r,j_r+}]$."

18. Equation 5 – please state how/if this equation is different to that from Kok et al. (2011), and if so why.

Despite the similarity in Equation 5/6 and that from Kok 2011, an important difference is that Kok 2011 describes the size distributions at emission while Eqn. 5/6 describes the size distribution in the atmosphere. Because of that, our formulation in Eqn. 5/6 builds on the brittle fragmentation theory of Kok 2011, but adds analytical expressions of dust deposition and dust changes during transport. In addition, the generalizations of parameters in our equation also allow us to better fit different shapes of dust mass fractions over different locations, and thus able to place a better constraints on dust size distribution.

To clarify this point, we have added the following section at the end of section 2.3.1.2 (formerly 2.1.2): "Finally, we note here that although our generalized theoretical function of Eqn. 6 builds on the brittle fragmentation theory of Kok 2011, it adds analytical expressions of dust deposition and dust changes during transport that allow us to better fit different shapes of dust size distribution over different locations."

19. Equation 5 & p8 l6 – why choose D_s for the geometric median diameter by volume? S subscript typically implies with respect to surface area. D_v would be more appropriate.

Thank you for this comment. We have changed the subscript from s to v

20. P8 l20-27 – This seems a great generalization. It's not clear how b is applied to specific locations as implied.

The application of all the parameters in Equation 6, including the parameter b, was done by fitting the expression using Equation 7 for each location. As a result, values for b could be different from one location to the other, based on the shape of the corrected dust mass fraction from Eqn. 2. The distributions of these parameter, including parameter b, are included in supplementary Fig. S-2. To make this clearer, we have reworded the paragraph before Eqn. 7, and also added additional sentence to the paragraph after Eqn. 7.

"To determine the parameters in Eqn. 6 for each height, horizontal location, season, and model simulation, we fit the generalized size distribution of Eqn. 6 to the corresponding corrected dust size distribution from Eqn. 2 above. To do this, we minimize the chi-squared ($\chi_k^2$) value for each height, location, and for each model k, such that:"

"The probabilty distribution of these parameters for all heights, horizontal locations, and model simulations of the annually-averaged dust size distribution is shown in the supplementary Fig. S-2"

21. Section 2.1.2 – What is the reason for choosing this method of fit (eqn 6) as opposed to fitting a series of lognormal modes, as is typically done for size distribution measurements? Presumably given the simulated size distributions have been corrected based on the same function, the fitting of the corrected size distribution is more naturally aligned with eqn 6?

That's a good point. We use Eq. (6) because it includes some mechanistic understanding of what determines the functional form of dust size distributions. Furthermore, although lognormal modes are appropriate for several other aerosol species, dust size distributions generally do not follow lognormal distributions very well, in part because the emitted dust size distribution is distinctly lognormal, as detailed in Mahowald et al. (2014). We have added a sentence to clarify this point in the first paragraph of section 2.3.1.2: "Although fitting lognormal modes are appropriate for several other aerosol species, Mahowald et al., (2014) highlighted that dust size distributions are usually not lognormal and are thus better characterized by a generalized function based on mechanistic understanding of dust emission and deposition processes. "

22. P10, section 2.3, l26-30 – units for all quantities would be helpful. What do the authors mean by "mass-weighted" in "mass-weighted vertically-integrated 2-D mass extinction efficiency" and what are the units of epsilon_tau and epsilon_m?

The units of each variable have now been included. And this phrase is indeed confusing as we meant "vertically-integrated 2-D mass extinction efficiency", thus the "mass-weighted" has been removed.

23. P10, section 2.3, l26-30 – I believe this calculation is the same as first used by Kaufman et al. (2005), which should be cited.

Good point. Kaufman et al. (2005) has now been cited.

24. P10 l31-32 – please list the reanalysis products (MERRA-2 etc) here to avoid confusion. Also see later comment about section ordering of 3.2. "Dataset" should be 'datasets.'

Thank you. This comment has been addressed in the paper.

25. P10 l32-p11 l2 – "This individual reanalysis dataset...." - I suggest removing this (and adding to section 3.2 if necessary). It is confusing here given that the AOD reanalyses have not yet been described.

P11 l10 "the four data sets..." – this is also confusing given that the AOD datasets have not been properly introduced at this stage in the paper. See later comment about relocating section 3.2. Ordering and section 3 – I suggest the authors move sections 3.1 and 3.2 to before section 2. This would be easier to follow and understand. Section 3.3 should remain after section 2 since it follows on logically.

This is a great idea to improve the paper's clarity, thank you. As suggested by the reviewer, we have combined section 2 and 3 and reorder the subsections, such that the description of model simulations (section 2.1) and the reanalysis products (section 2.2) comes before the description of the DustCOMM products (section 2.3)..

26. P16 l35 – 2011-2015 is presumably limited by available years? Is there any impact of this difference in years used?

Yes indeed, the dust AOD for JRAero is only available between 2011 and 2015. Although we did not analyze the impact of the difference in DAOD climatology between JRAero and the others, we expect that the relative difference will be smaller over the dust dominated region.

27. P17 l32 – and is also a 2-D diameter project of a 3-D shape, which may introduce bias (e.g. Chou et al., 2008).

Thank you. We have reworded the relevant sentence of the 3$^{rd}$ paragraph of the section to better reflect this.

"during the microscopy analysis, particle diameters are usually determined as the volume-equivalent geometric diameters based on 2-dimensional images (Chou et al., 2008). Because of the asphericity of dust aerosols, this could introduce some biases (e.g., Okada et al., 2001; Huang et al., in prep.)."

28. P17 l 36 "separate channels for different particle sizes" - this is not really relevant and could introduce confusion.

Thank you. This part of the sentence has been removed.

32. P17 l25-40 – there is a 4[th] category, which covers imaging probes, as used in the AER-D field campaign (section 4.18 of supplement) – which are beneficial since they do not suffer from uncertainties in converting scattered light to size as OPCs do.

That's a good point. We have included two sentences to highlight this point. First sentence is added in the first paragraph stating: "Another category is the imaging probe whereby the particle image is detected by linear photodiode array providing a two-dimensional projection of the particle (Baumgardner et al., 2017; Ryder et al., 2018)."

The other is added to second paragraph of section 2.4.1 stating: "Unlike the optical particle counters that require assumption regarding dust refractive index and shape to convert scattered light intensity to particle size, the imaging probes are not subject to these uncertainties (Baumgardner et al., 2017; Ryder et al., 2018)"

33. P18 l22-25 – OPCs have other sources of uncertainty – such as refractive index applied in the inversion of scattered light to size and the non-monotonic relationship between scattered light and particle size. These should also be mentioned.

We have included a sentence to clarify these points.

"In addition, optical particle counters also make assumptions about the refractive index to derive the dust size distribution, and are affected by the non-monotonic increase in the intensity of scattered light with particle size (Ryder et al., 2018; Weinzierl et al., 2011)."

34. P18 l11-25 and section 3.3.2 – in-cabin measurements are also subject to uncertainties and size-bias in sampling due to aircraft inlets. As such, the MEE values from studies in table 2 are likely biased high in some cases.

Thank you for this comment. We have added a sentence in this paragraph to highlight this point. "For in-cabin measurements, studies have shown that the loss rate of coarse dust particles can be substantial due to the aircraft's instrument inlet, therefore leading to lower sampling rate and size bias (e.g. von der Weiden et al., 2009)."

35. P18 l14 – although the size distribution measured does not allow aerosol type to be distinguished, various chemical composition measurements made in parallel are now mostly a matter of routine during airborne campaigns. Individual studies often use these to infer size distributions or ranges dominated by different aerosol types.

We thank the reviewer for this comments. While measurements of chemical composition helps to isolate dust from non-dust particles, there are still potential for mis-identification. We further address the discrepancy between measurements and our results in section 4.1

36. Section 3 – there is a huge variety of measurement data available, and I do not suggest the authors attempt to significantly widen their coverage. The authors should describe how and why the studies in Table 2 were selected. There also appears to be a geographical gap of sampling Arabian dust (Fig S1). Additionally, I suggest the studies of dust sampled during the AMMA airborne missions (Formenti et al., 2011) as being a very useful addition, since they provide summertime sampling in the Niger region, which is currently not covered by the studies in Table 2.

We thank the reviewer for this comment. For the most part, the studies we selected for dust size distribution are those that reported actual measurements of coarse dust particles and not log-normal fit or parameterized distributions. The dust size distribution reported in Fig. 10 of Formenti et al. (2011) is a log-normal fit to measured data, such that we cannot use this data. We have included additional sentences in the second paragraph of section 2.4.1 that clarify how and why some of the studies in Table 2 are selected.

"For the dust size distributions, our criteria for selection of studies are as follows: (1) the measured size range of the data should extend into the coarse dust (D > 5 um) size range; (2) the study should report the original in-situ measurements, instead of (lognormal) fits to the actual measurements; and (3) each study's measurements should be taken with commonly-used instrumentation in order to ensure some consistency with measurements taken by other studies. "

37. Table 2 – please indicate which studies relate to which numbers on the map on Figure S3.

We now clarify this in the caption of Fig. S-3.

38. Section 3.3.2 and Supplement section 4 – The descriptions of data taken from each measurement campaign are too vague, and occasionally in error. Often it is not enough just to reference a paper as within the measurement papers observations are collected/averaged in different ways (time periods, meteorological regimes, altitudes, etc.) and it is not clear which are being used here. The authors should state specifically which data are taken from each paper, and what the values of MEE or MSE are, preferably listing them in a table in the supplement. Specific comments about data described in the supplement are given in the Supplement section.

Thank you for the comment. We have made several additions to the supplementary section 4 to address this point. In addition, we also state in Table 2 where the data are taken from, by referencing the specific figure or table where applicable.

39. P18 l34-35 – Ryder et al. (2013b) SSA values fall well outside this SSA range. This is a fairly narrow SSA range selected. The authors should note that measured SSA is sensitive to the size range sampled in the observations, which is likely to exclude the coarse mode and often d>~2- 3 microns in many cases due to the effects of inlets. Only 3 studies are

cited, while there are a huge variety of studies in existence which have measured dust SSA.

We thank the reviewer for pointing this out. We have increased the uncertainty range to 0.03 and have also included citations to other studies.

40. P19 l9-11 – "These errors include errors due to the instrument measuring the extinction coefficient" – change to 'instrumental uncertainties.' "meteorological influence" – such as? "the assumption of internal or external mixing" – how is this important?

For clarity, we have removed this sentence, and added "including instrument uncertainties" to the preceding sentence.

41. P20 l13-17 – Field campaigns additionally often sample a variety of cases which are representative of the within-season variability, and also often include uncertainties/ranges to cover the variability encountered.

That is right. Thank you for pointing this out.

We have added a sentence to highlight this point: "Furthermore, most of these measurements are campaign averages often over a variety of cases that could be representative of the season-averaged size distribution."

42. P21 l2-3 "(1) the ACE-2 campaign (June/July, 1997) off the west coast of Western Sahara and Morocco (Otto et al., 2007)" – would be better referred to as in the vicinity of the Canary islands. Same for caption of Figure 3.

Thank you for pointing this out. We've corrected this accordingly.

43. P21 l6 "(2) the Fennec project (June 2011) between the Canary Islands and Mauritania/Mali (Ryder et al., 2013)" – If the authors refer to measurements between the Canary Islands and Morocco/Western Sahara (not Mauritania/Mali which are inland) the citation should be Ryder et al. (2013a – GRL) and the geographic references corrected. The same applies to the caption of Figure 3.

Thank you for this comment. We realize the confusion and we have clarified it in the paper. Data from the two Ryder 2013 papers are used in our study. Here, we do in fact mean the Ryder et al. (2013 – GRL) which is represented in the paper as Ryder et al. (2013a). We have adjusted the text, the geographical reference and images accordingly. The representative location is now placed at 27.65N, 14.25W.

44. Figure 3 – what is the reason for the selection of altitude choice? It seems biased very high – presence of dust at z>6km is unlikely and concentrations will be very low at

5.5km – therefore the value of such high altitude comparisons is questionable. What is the reason for the selection of these 3 studies for Figure 3? The geographic spacing is very close, with all sampling JJA SAL dust. "ACE-2" in line 7 of the caption should read "AER-D."

Thanks for the correction. ACE-2 has been changed to AER-D.

We identify these 3 studies to show that DustCOMM performs better than model simulations for a range of heights. We recognize that dust concentrations are lower for z>6km, but we follow Fig. 1 of Ryder et al. (2013 – GRL) which shows that there are still some coarse dust particles at ~6km. To address this comment, we have added additional comments in Table 2, and a justification for the height selection in the supplementary document.

45. P22 l1 – 14% and 15% - in terms of which variable?

This is in terms of the dust mass fraction – i.e. the fraction of dust per unit mass of dust loading. We have clarified this in the paper. The sentence now reads "On average, simulations in our model ensemble overestimate the dust mass fraction of the fine mode by ~14%, and underestimate that of the coarse mode by ~15%."

46. P23 l25 – and also Qinghai Province China?

Yes, good point. We have added this to the text: "… and Qinghai Province, China (Li et al., 2000)"

47. P24 l17 – "weighted by the dust vertical distribution" – why is this necessary?

We agree with the reviewer that "column-integrated dust MEE" explains our point in that sentence, and "weighted by the dust vertical distribution" is indeed not necessary. As a result, we have removed it.

48. P24 l31-32 – as stated earlier, it is not clear why the MEE and AODs need to be used to calculate the column mass loading, given that this is typical model output, and the size distributions are already available. It should be a direct step to calculate column mass loading from size distributions, given a dust density.

See our response to similar comments above (General Scientific Comments #1)

49. Section 5.1 – are there any impacts of uncertainties in wet deposition on the size distribution biases?

Uncertainties in dust deposition broadly affect our estimates of dust size distribution, as we discuss in the fourth paragraph of the new section 4.1 (formerly 5.1).

50. P27 l30 – Fig 7a does not show MEE.

Thank you. This is noted and has been corrected.

51. P27 l31 – there is no figure S7 in the supplement.

Thank you. This is noted and has been corrected. We meant to say supplementary Fig. S-6

52. Section 5.4 – the implications of dust LW properties should be reflected on here, considering the points about the LW radiative impacts of dust being crucial to the total impact on the radiation balance described above.

We thank the reviewer for this comment. We have now added a sentence in section 4.4 (formerly 5.4) that emphasizes the use of our dust size distribution for longwave radiative impacts. "With improved constraints on the dust size distribution and therefore the dust optical properties, DustCOMM could be used to determine the dust (shortwave and longwave) heating rates in the atmosphere more accurately than possible with current global model simulations. As a result, our constraints on dust size distribution could be used to better quantify radiative effects of dust, especially in the longwave spectrum which have remained very uncertain (Di Biagio et al., 2017; Dufresne et al., 2002; Kok et al., 2017; Song et al., 2018)"

53. P30 l20 – should 'indirect effects' be 'semi-direct effects'?

Good point. We have included both indirect and semi-direct effects.

54. P30 l28 – some reference to the SW spectrum and 550 nm should be included, since refractive index and MEE are only considered at this wavelength.

We have clarified that the DustCOMM mass extinction efficiency is at 550 nm. Since our constraints is taken from Kok et al. 2017, we have also included that reference in the sentence.

55. Section 5 – There is a general focus on in-situ observations for validation of DustCOMM. However, remote sensing observations are developing rapidly and it would be useful for the authors to consider whether lidar retrievals, for example, would be usable within the DustCOMM framework.

Remote sensing observations are certainly useful as we continue to develop DustCOMM. As we stated in section 4.4, we hope to incorporate more observational constraints within DustCOMM framework. For example, to constrain the vertical distribution of the atmospheric dust loading, lidar-based retrieval of dust extinction, such as from CALIPSO, will be very useful, and it is part of future work.

We have added a sentence in section 4.4 to further highlight this point. "For instance, a next step could be to include constraints on the dust vertical concentration profile over every location, in order to more accurately estimate dust deposition, and dust concentration at the surface and in 3D. For this, lidar-based retrieval of vertical dust extinction profiles from Cloud-Aerosol Lidar and Infrared Pathfinder Satellite Observations (CALIPSO) can be combined with the corresponding constraints on dust mass extinction efficiency from this study to obtain constraints on the dust vertical concentration profile."

56. P31 – l25-26 – the bias across the full size range should also be stated.

We have included an additional sentence to address this point.

"Because DustCOMM underestimates the measurements for $D \leq 0.5\mu m$, it shows a more negative bias (~50% more) over the full size range (between $D = 0.2 - 20\mu m$), although the error is markedly lower (~15 %), when compared to the ensemble of model simulations. Overall for $D \geq 0.5\mu m$, DustCOMM shows a bias against measured size distributions that is significantly less (about 46% less) than for an ensemble of global model simulations."

57. Are there any important dust altitude or seasonal changes in DustCOMM vs the models?

As Fig. 7 suggests, DustCOMM vertical profiles follow the form of the ensemble of global model simulations, but the fraction of dust mass in each bin is different from that of the model ensemble since the constraints adjust every location and height by the same factor. In addition, similar adjustment to the annually-average dust mass fraction between DustCOMM and model ensemble is also apparent at the seasonally-averaged timescale.

We have added a sentence in section 3.1.2.2 to clarify this: "Finally, similar changes in the spatial variability of the annually-averaged dust mass fraction are apparent in the seasonally-averaged values."

58. Discussion/Conclusion - It would be interesting if the authors could comment on bias of models vs measurements in previous studies (e.g. Hunneus et al., 2011; Evan et al., 2014), and similarities/improvements seen in those studies vs DustCOMM and the model simulations in this study.

Since models used in those studies (e.g. AeroCom in Hunneus et al., 2011) are similar to those used in our study here, they suffer from similar biases and shortcomings. That is the biases in dust properties are associated with biases in dust size distribution, it therefore suggest that better constraints on size distribution as done with DustCOMM should provide a better estimates of these dust properties. Here in section 4.4, we have added the sentences below to highlight this point.

"Furthermore, since recent studies associate much of the biases in dust properties, such as the dust aerosol optical depth, deposition fluxes and surface dust concentration, to model biases in dust size distribution (Evan et al., 2014; Huneeus et al., 2011), DustCOMM estimates can therefore serve as a better alternative. For example, DustCOMM's improved constraints on atmospheric dust loading and dust size distribution could contribute to better estimates of size-resolved dust concentration near the surface (e.g. Whicker et al., 2018). Over the ocean, such constraints on size-resolved dust concentration could potentially be used for constraints on dust deposition fluxes that are more accurate than possible from global model simulations."

59. AOD reanalyses – do the authors combine these into one single reanalysis dataset themselves? This is not really clear.

Yes, they are combined into one single data and the details of this is given in the (new) section 2.2.

**Technical Comments**

1. P3 l12 – "The resulting product constrains the climatology of 3-D global atmospheric dust properties on seasonal and annual timescales" – change to "The resulting product constrains the climatology of 3-D global atmospheric dust properties and is provided on seasonal and annual timescales" – to avoid confusion that the authors are constraining the temporal variability of dust properties.

Thanks for this comment. We have changed the sentence accordingly.

2. P3 l34-35 – "After correcting..." – unclear – do you mean you combine all models into one multi-model representation?

Yes. We have added the word "multi-model" to make it clear

3. P6 l28 – "globally-averaged size distribution" – Kok 2017 or the simulated one? P7 l13 - "globally-averaged size distribution" – Kok 2017 or the simulated one?

We meant the Kok et al 2017 or the constrained globally-averaged dust size distribution here. We have changed them to ""constrained globally-averaged dust size distribution". We have also clarified other places where globally-averaged size distribution are mentioned.

4. P9 l18 – typo – should be -10 to -4?

    Thank you. It is in fact between -10 and 4, but because we realize this can be confusing, we have changed this (and others) to be $-10$ to 4.

5. Eqn 8 – please provide units for epsilon_tau

    We have provided unit for this parameter, and others alike.

6. P10 l27 – "atmospheric "column" dust loading"?

    Yes. We have included "column".

7. P14 l25 – change to "...of the in-situ emission measurements.."

    Thank you for the comment. The sentence now read "The dust MEE is influenced by the uncertainty in the constrained globally-averaged extinction efficiency, which in-turn is partially due to uncertainties in the in-situ emission measurements of index of refraction and dust particle shapes"

**Supplement Comments**

1. The supplement contains two Figure S1s. The second should be S3 (?).

    This is corrected. Thank you.

2. Section 3.1 – l7-8 – mention that it is the AOD which is assimilated.

    Yes. We have done that. The sentence now reads.

    "For the first time, meteorological and aerosol observations (which include bias-corrected **aerosol optical depth** from MODIS, AVHRR, MISR – over desserts, and ground-based AERONET instruments) are jointly assimilated into MERRA-2…"

3. Section 3.3 - "1.1ox1.1o" typo
   This has been corrected. Now written as 1.1°x1.1°

4. Section 4 – To make this easier to navigate, relate each observational subheading to the numbers on fig S3 (map). Also include the campaign name in the heading for each section. Take care to state for each subheading whether the campaign was ground-based or airborne. Also explain the choice of altitude selection defined in table to where relevant. Please also be aware, and state where necessary, that although a large size range may have been measured, inlet-size effects may have prevented coarser particles from being measured for some campaigns.

   Thank you for the comment. We have included in each observational heading the campaign name as well as whether only PSD or MEE is taken or both. For cases where PSD are taken, we have also included sentences explaining the choice of our representative altitude.

5. Section 4 – a subsection on Kandler et al. (2009), as listed in table 2, is missing.

   Thank you for pointing this out. We have included a brief description of Kandler et al (2009).

6. Section 4.1 – please make the locations listed consistent with those listed in table 2.

   Thank you for the comments. We have corrected where discrepancies occur.

7. Section 4.7 – note that these size distributions were not corrected for refractive index. The FSSP was *not* used as it did not operate correctly. Instead the size distribution larger than d=3 microns was taken from a sunphotometer retrieval.

   Thank you for pointing this out. We have removed the mention of FSSP, and include a sentence mentioning that the they did not correct for refractive index.

   "The size distributions were not corrected for refractive index because they assumed that the refractive index of latex is approximately similar to that of dust."

8. Section 4.10 – Please note that these studies operated instruments behind significant pipework and suffered loss of the majority of coarse particles (e.g. Ryder et al., 2018, Table 1).

   Thank you for pointing this out. We have included a sentence in the section stating that: "Because of the aerosol inlet configuration on the aircraft, the measurement of coarse dust were particularly problematic."

9. Section 4.11 – Why is MEE only taken from DODO1 (winter time?). It appears that the MEE for DODO1 is taken from table 4 of Osborne et al. (2008), for the 'AM+CM' case (a value of 0.41). No coarse mode was measured during DODO1 (see McConnell et al., 2008).

The AM+CM DODO1 case in Osborne et al. (2008) was calculated using the coarse mode size distribution from DABEX since none was available from DODO1. This should be stated, or preferably the value from DODO2 used, where coarse mode was measured. Why is only z<1km used for the DODO2 size distributions?

*Thank you for this helpful comment. We have clarified this point in the section.*

10. Section 4.12 – is the campaign average size distribution used?

*These data are taken from their Fig. 8 which represent the composite size distribution for L02 on flight #060519a and L07 on flight #060604a. We have included an additional sentence in this section to clarify this point.*

11. Section 4.13 – and also same aircraft as SAMUM1? SAMUM1 also used a high spectral resolution lidar.

*They used the Falcon aircraft, which was also used in SAMUM-1. We have added this information to the section.*

12. Section 4.14 – 'used the same instrumentation...' – as which paper/campaign? Presumably the same as Kandler et al. (2009) which is missing? It is not clear which instruments the size distribution comes from – but probably because the 2009 section is missing.

*Thank you for pointing this out. Yes, the instrumentations are similar to that from SAMUM-1 (Kandler et al., 2009). This detail has been clarified in the section.*

13. Section 4.15 – Data from Ryder et al. (2013a – GRL, Canary Islands) is also used in the paper (Figure 3) and should be described here. Please take care to specify whether Ryder et al. (2013a or 2013b) is being cited – both are given in the references as 2013. Comparisons of both can be found in Ryder et al. (2019). "For this study, mean distribution from PCASP and CDP were selected because they were the most credible based on the authors' analysis." – change to ..."based on the authors' analysis over the size range we use here." MEE is not given in Ryder et al. (2013b - ACP), presumably this is taken from Ryder et al. (2013a) (please state). Mean values in Ryder et al. (2013a) are 0.15 for fresh dust or 0.23 for aged dust – these appear much lower than the value plotted in Figure 8 (around 0.3).

*Thank you for these very helpful comments. We have re-written this section to include the descriptions of both Ryder et al. (2013a – GRL) and Ryder et al. (2013b – ACP). Indeed, data from both studies were used in this paper. For Ryder et al. (2013a), we obtained the dust size distributions as a function of heights, which were generously given to us by the first author. For Ryder et al. (2013b), we obtained the campaign averaged dust size*

distribution already published. Appropriate geographical references have also been noted both in the supplementary document as well as in the main text and figures. For the MEE, we used the averaged values of 0.31+/-0.08 between the reported values for aged dust (0.23) and the SAL categories (0.39).

14. Section 4.16 "above the SAL" – I would expect a dust measurement to be taken 'in' the SAL – is this a typo?

Thank you for pointing this out. It is indeed within the SAL layer. We have made the correction accordingly.

15. Section 4.18 – "The AER-D campaign uses similar instrument as the Fennec 2011 campaign. They use wing-mounted optical particle counters and shadow probes to measure dust sizes between 0.1 and 100 μm diameter." – but additionally this AER-D used cloud imaging probes (CIP15 and 2DS) for size distributions at d>10 microns (which were used in Fennec but were not mentioned in Section 4.15 as the authors did not use the shadow probe data (d>18.5 microns) in this study).

Thank you for the comments. We have included in this section that the imaging probes are also used.

This study presents a new dataset, the Dust Constraints from joint Observational- Modelling-experiMental analysis (DustCOMM), which combines in-situ measurements, reanalysis products, and an ensemble of six global model simulations. Particularly, globally-averaged dust size distribution and extinction efficiency from observational and experimental data are used to constrain the DustCOMM products. The annual and seasonal mean products of 3-dimensional (3D) dust size distribution, 3D dust mass extinction efficiency, and 2D dust loading are provided for the time period from 2004 to 2008. It is found the dataset shows a better agreement with measurements than the six-model ensemble in terms of dust size distribution and mass extinction efficiency. This dataset may be used to constrain dust simulation in global models and to study dust impacts on the earth system. The paper is generally well written. The methodology to develop the datasets is thoroughly introduced and related uncertainties are also discussed in detail. I have a few comments would like the authors to address.

We thank the reviewer for the constructive and helpful comments that helped us to further improve the paper.

Major comments:
1. Here globally-averaged dust size distribution is used to obtain 3D dust size distribution. Is it possible to demonstrate that the regional differences in dust size distribution are small? Or have you considered using different dust size distribution for different regions, e.g., by applying regional averaged values to areas where individual measurements are available and the globally-averaged value to areas where measurements are not available? This might provide better spatial constraints on the dataset. Similarly, globally-averaged dust extinction efficiency at 550 nm is used. How large are the spatial differences? Is it possible to give a rough estimation based on available data?

We thank the reviewer for this insightful comment. Assuming globally consistent size distributions and extinction efficiency is indeed one of the main assumptions in this paper.

We used globally-averaged dust size distribution and dust extinction efficiency because the measurements to constrain these parameters on a regional basis across the different dust-source regions are currently insufficient. Since North African dust dominates most of the global dust emission and many of the measurements used to constrain the globally-averaged values are associated with the North African dust, constraining the regional dust properties with insufficient measurements will likely result in larger uncertainties than estimated in this study. Moreover, since our constraints are applied globally, regional differences in dust size distribution, for example, are assumed to follow the ensemble of the six model simulations.

To better clarify this point in the manuscript, we have added additional sentences to the first paragraph of section 4.3: "We used modelling constraints in DustCOMM where observational constraints were either not available or insufficient. For example, modelling constraints are used for the regional differences in dust size distribution and extinction efficiency because the measurements to constrain these parameters on a regional basis across the different dust-source regions are currently insufficient. To further reduce the uncertainty associated with using modelling constraints, we used an ensemble of six model simulations."

2. As discussed in the paper, dust aerosol optical depth from the reanalyses largely depends on the models' treatment of the dust cycle, and this adds uncertainties to the DustCOMM. I wonder if you considered using satellite products of dust optical depth, such as level 3 dust optical depth from the Cloud-Aerosol Lidar with Orthogonal Polarization (CALIOP).

That's a good suggestion. Although our analysis includes both random and systematic errors in dust AOD by incorporating the satellite-based study of Ridley et al. 2016 with the ensemble of reanalysis datasets, we agree that the estimate of the dust AOD likely incurs additional uncertainties associated with model treatments of dust cycle. However, dust extinction retrieval from CALIOP also suffer from several uncertainties, such as weak signal-to-noise ratio during daytime versus nighttime retrievals (e.g. Kacenelenbogen et al., 2011; Winker et al., 2013) and erroneous assumption of aerosol extinction-to-backscatter ratio used in the extinction retrieval (e.g. Omar et al., 2009; Mamouri et al., 2013; Nisantzi et al., 2015). In addition, it is unclear how the limited spatial coverage of CALIOP AOD retrieval affects the climatological estimates. Nonetheless, our plan is that future versions of DustCOMM will incorporate extinction profiles from CALIOP in estimating the vertical distribution of dust concentration.

We have added a sentence in section 4.4 to further highlight this point. "For instance, a next step could be to include constraints on the dust vertical concentration profile over every location, in order to more accurately estimate dust deposition, and dust concentration at the surface and in 3D. For this, lidar-based retrieval of vertical dust extinction profiles from Cloud-Aerosol Lidar and Infrared Pathfinder Satellite Observations (CALIPSO) can be combined with the corresponding constraints on dust mass extinction efficiency from this study to obtain constraints on dust vertical concentration profile."

3. Sections 4-5 show that the new dataset has a better agreement with in-situ measurements than the multi-model mean. I think it is better to add some discussion to emphasize why this dataset is a good complement to the currently available observational data, especially individual measurements. For instance, the global coverage and vertical distribution of the

dust size distribution and mass extinction efficiency of the dataset make it easier to be adapted to global models to constrain simulations or to study global dust impacts.

We discuss these possible uses of DustCOMM in section 4.4 (formerly section 5.4). There we discuss how DustCOMM can be used to constrain dust impacts in models, and how it can also serve as alternative to global model simulations.

Minor comments:

1. Lines 21-23, page 2, this can be a bit misleading since both small are large dust particles absorb and scatter shortwave and longwave radiation.

   Yes, the result of the combined absorption and scattering (i.e. extinction) is cooling for fine dust and warming for coarse dust. For clarity, we now specify that "fine dust **predominantly** cools the climate system by extinguishing shortwave (SW) radiation …"

2. Line 6, page 3, "To address this problem", not sure the dataset would be able to address the "numerous important biases". You may want to point out a few detailed problems.

   Thank you for the comment. Our dataset does address the problem showing significant improvement over model simulation, although it does not completely eliminate the biases. We have re-written this part as: "To address the problem of size and shape biases in models"

3. You may want to add the horizontal and vertical resolutions of the DustCOMM product at someplace in Section 2.

   Thank you. We have added a sentence in section 2.3 stating "We estimate all DustCOMM products at 2.5º X 2.0º horizontal grid with 35 levels that is up to 100 hPa."

4. Line 17, page 11, what time period does the "climatology" refer to?
   The climatology is between 2004 and 2008. We have added this to the sentence

5. Section 3.1, are all the model results interpolated to the same horizontal and vertical grids? And what's the resolution?

   Yes. We stated in the last paragraph of section 2.1 (formerly 3.1) that "…we interpolated seasonal and annual climatologies of these dust properties to a common resolution of approximately 2.5º by 2.0º spatial resolution, with 35 levels from the surface to 100 hPa"

6. Line 35, page 16, why the JRAero in a different time period is used? It's not available from 2004 to 2008?
Yes. JRAero is only available between 2011-2015.

7. Line 16, page 21, do you refer to Fig. 4 instead of Fig. S4?
Thanks for the comment. We have deleted the statement in parenthesis because it is no longer available in the supplementary document.

8. Line 16-18, page 21, can you please add some discussion about why the DustCOMM has a larger bias than model ensemble for D ≤ 0.5 μm?

Thank you for the comment. We discussed this in section 4.1 (formerly 5.1). First, we highlighted that "DustCOMM's underestimation of dust with $D \leq 0.5 \mu m$ may be caused by the contamination of the measured size distributions by other aerosol species for $D \leq 0.5 \mu m$." Second, we discussed that "the constraint on the globally-averaged dust size distribution could also underestimate the contribution from dust with $D \leq 0.5 \mu m$."

9. Line 19-24, page 23, "...regardless of the season and location", except Sde Boker, Israel.

Although the statement is true overall, we have removed this part of the sentence.

10. Table 1, please remove "deg" in column four, since you already added a degree symbol there.
Thank you. We have done just that.

11. Figs. 2-3, can you please add latitude, longitude, and location of the measurements on the top of each plot? Or you may number the measurements listed in Table 2 and then simply list the corresponding numbers in the figure.

Thank you for the comment. We have instead added the names of the campaign listed in Table 2 on each plot.

12. Fig. 5, is it possible to add a globally averaged PSD and its PDF to the plot?
Since DustCOMM globally-averaged values are forced to the globally-averaged values from Kok et al. 2017, the globally-averaged PSD the reviewer requested is also represented by the black lines.

13. Fig. 6, why is dust mass fraction for D= 0.2-2.5 μm high over the ITCZ? Is this consistent with observations?

Because dust concentration is usually low over the ITCZ region, they are dominated by the fine particles.

14. Fig. 7, it would be more interesting to show individual model results (as in Fig. 5) instead of multi-model results.

Thank you for the comment. We have included the individual model results in Figure 7.

15. Fig. 8, why do some blue dots have a light blue outline?

Those are cases with different measurement type as explained in section 2.4 (formerly 3.3). To avoid confusion, we have removed this from the figure.

[revised manuscript text omitted]

**1. Supplementary Figures**

[Figure]

*Figure S-1: The globally-averaged size distributions for each model normalized between 0.2 and 20 μm. It shows the model uncorrected dust size distribution (red lines), corrected dust size distribution (blue lines), Kok et al. 2017, and the final constrained DustCOMM dust size distribution with the sub-bins (black lines). Note that since the constrained DustCOMM dust size distribution forced to that of Kok et al., 2017, the lines overlap.*

[Figure]

*Figure S- 2: Probability distribution of the parameters for the generalized analytical function describing the atmospheric dust size distribution. See section 2.1.2 for details. The shaded regions denote the 95% confidence intervals of each distribution.*

[Figure]

Figure S- 3: Map showing the locations of measurements for evaluation used in this study (Table 2). The measurements in Table 2 that corresponds to the numbers are as follows:

**1 – D'Almeida & Schutz, (1983), Osborne et al., 2008, Chou et al., 2008;**
**2 – Li et al., 1996, Jung  et al., 2013, Weinzierl et al., 2017;**
**3 – Li et al., 2000;**
**4 – Maring et al., 2000, Otto et al., 2007; Ryder et al., 2013a**
**5 – Andreae et al., 2002;**
**6 – Quinn et al., 2002;**
**7 – Quinn et al., 2002;**
**8 – Quinn et al., 2002;**
**9 – Haywood et al., 2003, Kandler et al., 2011, Weinzierl et al., 2017, Ryder et al., 2018;**
**10 – Clarke et al. 2004;**
**11 – McConnell et al., 2008;**
**12 – Weinzierl et al., 2009, Kandler et al., 2009;**
**13 – Wagner et al., 2009;**
**14 – Ryder et al., 2013b**

[Figure]

*Figure S- 4: Annually-averaged ensemble mean and relative uncertainty of reanalysis dust aerosol optical depth (left panel). See section 3.2 for details Right panel shows the difference between the reanalysis dataset and the model ensemble dust aerosol optical depth.*

**Relative Uncertainties in Model Ensemble**

[Figure]

*Figure S- 5: Spatial distributions of model ensemble relative uncertainties for (a-d) the dust mass fraction in the diameter range between $0.2 − 2.5\mu m$, $2.5 − 5\mu m$, $5 − 10\mu m$, and $10 − 20\mu m$; (e) the dust mass extinction efficiency (MEE), and (f) dust load.*

[Figure]

*Figure S- 6: (a) Globally-averaged Single-particle DustCOMM dust mass extinction efficiency (MEE; Black line) and one calculated from Mie theory (blue line); (b) the effect of dust asphericity shown as the percentage differences between the dust MEE from DustCOMM and the one from Mie theory. All the black lines present the median of the distribution for each diameter, while the grey shade is the 95% confidence interval. The DustCOMM dust MEE leverages observational constraints on dust shape and dust size distribution (see section 2 in text). In contrast, the blue dashed line denotes the dust MEE calculated from Mie theory, which uses the assumption that dust particles are spherical.*

[Figure]

*Figure S-7: The original and modified globally-averaged dust size disttribution for each model simulation.*

**2.        Global model simulations**

We describe here the model simulations used in this study. The GISS, CESM and GEOS-Chem models are described in detail in Kok et al. (2017) and the references therein (see section 5 of their supplementary document), while the simulations with the WRF-Chem, ARPEGE-Climat and IMPACT models are described below.

2.1.1    WRF-Chem

We use the version of WRF-Chem model (Grell et al., 2005) that is improved by the University of Science and Technology of China (Zhao et al., 2013). This version uses the quasi-global channel configuration with the periodic boundary conditions in the zonal direction and $360 \times 145$ grid cells ($180°$ W-$180°$ E, $67.5°$ S-$77.5°$ N) to perform the simulations at $1°$ horizontal resolution, 35 vertical layers up to 50 hPa, and for the period of 2007-2016. The meteorological initial and lateral meridional boundary conditions are derived from the National Center for Environmental Prediction final analysis (NCEP/FNL) data. In addition, the model simulated winds and atmospheric temperature are nudged towards the NCEP/FNL reanalysis data with a nudging timescale of 6 hr (Stauffer & Seaman, 1990). Furthermore, the simulation uses MOSAIC (Model for Simulation Aerosol Interactions and Chemistry) aerosol module (Zaveri et al., 2008) coupled with the CBM-Z (carbon bond mechanism) photochemical mechanism (Zaveri and Peters, 1999). This aerosol model uses the bin approach with eight discrete size bins to represent aerosol size distributions (Fast et al., 2006). All major aerosol compositions are simulated in the model, including the including sulfate, nitrate, ammonium, black carbon, organic matter, sea-salt, and mineral dust. The MOSAIC aerosol scheme also includes physical and chemical processes of nucleation, condensation, coagulation, aqueous phase chemistry, and water uptake by aerosols. More details, including the model physics scheme used, can be found in Zhao et al. (2013).

Vertical dust emission fluxes are calculated as described in Zhao et al. (2010) based on the GOCART dust emission scheme (Ginoux et al., 2001). The emitted dust particles are distributed into the MOSAIC aerosol size bins following a theoretical expression that is based on the physics of scale-invariant fragmentation of brittle materials derived by Kok (2011). For MOSAIC 8-bin,

dust particles are emitted into eight size bins with mass fractions of $10^{-6}$ %, $10^{-4}$ %, 0.02%, 0.2%, 1.5%, 6%, 26%, and 45%, respectively. The dry deposition of aerosol mass and number is simulated following the approach of Binkowski & Shankar (1995), which includes both turbulent diffusion and gravitational settling. Wet removal of aerosols by grid-resolved stratiform clouds and precipitation includes in-cloud removal (rainout) and below-cloud removal (washout) by impaction and interception, following Easter et al. (2004) & Chapman et al. (2009). Cloud-ice-borne aerosols are not explicitly treated in the model, but the removal of aerosols by the droplet freezing process is considered. Convective transport and wet removal of aerosols by cumulus clouds follow Zhao et al. (2010, 2013).

The AOD is computed as a function of wavelength for each model grid box. Aerosols are assumed internally mixed in each bin (i.e., a complex refractive index is calculated by volume averaging for each bin for each chemical constituent of aerosols). The Optical Properties of Aerosols and Clouds (OPAC) data set (Hess et al., 1998) is used for the shortwave and longwave refractive indices of aerosols, except that a constant value of 1.53+0.003i is used for the SW refractive index of dust following Zhao et al. (2010, 2011). A detailed description of the computation of aerosol optical properties in WRF-Chem can be found in Fast et al. (2006) & Barnard et al. (2010).

**2.1.2  IMPACT**

The global chemical transport model used in this study is a coupled gas-phase (Ito et al., 2007) and aerosol chemistry version (Liu et al., 2005) of the Integrated Massively Parallel Atmospheric Chemical Transport (IMPACT) model (Rotman et al., 2004). A detailed description can be found in Ito & Kok (2017) and references therein. The IMPACT model is driven by assimilated meteorological fields from the Goddard Earth Observation System (GEOS) of the NASA Global Modeling and Assimilation Office (GMAO) with a horizontal resolution of 2.0° × 2.5° and 59 vertical layers up to 0.01 hPa. The model simulates the emissions, chemistry, transport, and deposition of major aerosol species (Liu et al., 2005) and their precursor gases (Ito et al., 2007). IMPACT takes into account emissions of primary aerosols and precursor gases of secondary aerosols such as sulfate, nitrate, ammonium and oxalate. Mineral dust aerosols are distributed among 4 bins in the model. A total dust source is dynamically calculated by a physically-based dust emission scheme (Kok et al., 2014a, 2014b) in conjunction with satellite products of

vegetation cover and soil moisture in the model (Ito & Kok, 2017). The chemical composition of mineral dust aerosols may change dynamically from that in the originally emitted aerosols due to reactions with gaseous species.

Dry deposition of aerosol particles uses a resistance-in-series parameterization (Zhang et al., 2001). Gravitational settling is also taken into account (Rotman et al., 2004; Seinfeld & Pandis, 2016). Aerosols and soluble gases can be incorporated into cloud drops and ice crystals within cloud (rainout), collected by falling rain and snow (washout), and be entrained into wet convective updrafts (Liu et al., 2001; Rotman et al., 2004; Ito et al., 2007; Ito & Kok, 2017). The aging of dust and combustion aerosols from hydrophobic to hydrophilic enhances their dry and wet deposition. Hygroscopic growth of mineral dust and combustion aerosols in gravitational settling uses the Gerber (1991) scheme, including the particle growth due to sulfate, ammonium, and nitrate associated with the particles (Liu et al., 2005; Xu & Penner, 2012). Scavenging efficiencies for mineral dust and combustion aerosols in wet deposition are calculated based on the amount of sulfate, ammonium and nitrate coated on the particles (Liu et al., 2005; Xu & Penner, 2012).

The AOD at 550 nm is calculated online using a look-up table as a function of wavelength and size parameter, following Xu & Penner (2012). Five types of aerosols (i.e., carbonaceous aerosols from anthropogenic combustion, carbonaceous aerosols from open biomass burning, dust, sulfate, and sea salt) were assumed to be externally mixed in each size bin, while sulfate, ammonium, and nitrate coated on each aerosol was internally mixed within each aerosol type and size bin. The refractive index for internally mixed aerosols is calculated based on the volume weighted mixture for each aerosol type and size bin.

**2.1.3   ARPEGE-Climat**

This study uses the global climate model from CNRM, namely ARPEGE-Climat, in its version 6 used in the CMIP6 exercise, with a horizontal resolution of ~1.4° and 91 vertical levels (Michou et al., 2015). ARPEGE-Climat includes an interactive tropospheric aerosol scheme, named TACTIC (Tropospheric Aerosols for ClimaTe In CNRM), able to represent the main anthropogenic and natural aerosol types in the troposphere. Originally developed in the GEMS/MACC project (Morcrette et al., 2009), this scheme has been adapted to the

ARPEGE/ALADIN-climate code (Michou et al., 2015; Nabat et al., 2015). Aerosols are included through sectional bins, separating desert dust (6 size bins whose limits are 0.1, 0.2, 0.5, 1.0, 2.5, 10.0 and 100 µm), sea-salt (3 bins whose limits are 0.03, 0.5, 5.0 and 20.0 µm), sulfate (1 bin, as well as 1 additional variable for sulfate precursors considered as SO2), organic matter (2 bins: hydrophobic and hydrophilic particles) and black carbon (2 bins: hydrophobic and hydrophilic particles) particles. All these 15 species are prognostic variables in the model, submitted to transport (semi-lagrangian advection, and convective transport), dry deposition, in-cloud and below-cloud scavenging. The interaction with shortwave and longwave radiation, is also taken into account through optical properties (extinction coefficient, single scattering albedo and asymmetry parameter) calculated using the Mie theory. Sulfate, organic matter and sea salt concentrations are used to determine the cloud droplet number concentration following Menon et al. (2002), thus representing the cloud-albedo effect (1st indirect aerosol effect).

Focusing more on dust aerosols, emissions are fully interactive, based on the parameterization of Marticorena & Bergametti (1995) which provides the saltation flux depending on surface wind and soil characteristics. The latter consist in the roughness length and the sand/clay/silt fractions, which are based on the ECOCLIMAP database (Masson et al., 2003). The distribution of the resulting emitted dust vertical flux follows then the study of Kok (2011), assuming an analogy with the fragmentation of brittle materials. The six dust size bins have the following effective diameters: 0.09, 0.18, 0.4, 0.9, 3.7 and 13.2 µm. Dry deposition (for the 6 dust bins) and sedimentation (only applied to the two coarsest size bins) are calculated from fixed vertical speeds (respectively Wisely and Hicks, 2000, and Thompson, 2005). Wet deposition includes below-cloud and in-cloud scavenging. The latter relies on the parameterization of Giorgi & Chameides (1986), assuming a fraction of dust aerosols included in droplets equal to 0.1 for the two finest bins and 0.2 for the 4 other bins.

In the present study, a five-year simulation (2004-2008) has been carried out using the ARPEGE-Climat model and its interactive aerosol scheme.

**3. **Description of the reanalysis datasets**

**3.1. MERRA-2 Aerosol Reanalysis**

The MERRA-2 is the second version of the MERRA atmospheric reanalysis product from the NASA Global Modeling and Assimilation Office (Gelaro et al., 2017), with updates on the reanalysis system to include addition of more observational platforms and correction of known limitations from previous MERRA version (Mccarty et al., 2016), as well as improvement to the Goddard Earth Observing System -5 (GEOS-5) atmospheric general circulation model, used as the base model for the global assimilation system (Mccarty et al., 2016). For the first time, meteorological and aerosol observations (which include bias-corrected aerosol optical depth from MODIS, AVHRR, MISR – over desserts, and ground-based AERONET instruments) are jointly assimilated into MERRA-2, with the aerosol fields simulated with radiatively-coupled version of Goddard Chemistry, Aerosol, Radiation and Transport model (GOCART) (Colarco et al., 2010). GOCART treats aerosol particles as externally mixed, with dust particles provided in five non-interacting bins (Randles et al., 2017). The dust emission in GOCART is based on Ginoux et al. (2001), which depend on wind speed, following the parameterization of Marticorena & Bergametti (1995). Aerosol loss processes include dry deposition, large-scale wet removal, and convective scavenging. While the dry deposition is mostly model dependent, the precipitation-induced aerosol deposition however, depends largely on the assimilated global precipitation information in MERRA-2 (Reichle et al., 2014, 2017). MERRA-2 aerosol properties are available from 1980 onward, but the number of observations assimilated is more than doubled after the year 2003 (Fig. 3 in Randles et al., 2017). MERRA-2 is available for 3-hourly temporal resolution, and 1.5º X 1.5º horizontal fixed spatial resolution.

We use the monthly averages (calculated from daily means) of MERRA-2 DAOD to construct the seasonal and climatological DAOD values between 2003 and 2012. Aerosol products from MERRA-2 have been validated against independent observation (Buchard et al., 2017; Randles et al., 2017), especially for the aerosol optical depth. It is worth noting here also that only AOD is directly constrained by the assimilation in MERRA-2, while other non-analyzed, non-constrained aerosol properties, like the vertical distribution and aerosol speciation are mostly model-dependent, thereby providing a possible source of uncertainty in the MERRA-2 DOAD reanalysis.

3.2.    NAAPS

The Navy Aerosol Analysis and Prediction System (NAAPS) is an offline aerosol transport model (Lynch et al., 2016) driven by the Navy Operational Global Analysis and Prediction System (NOGAPS; Hogan & Rosmond, 1991; Hogan & Brody, 1993). The quality-assured and quality-controlled MODIS and MISR aerosol optical depth are assimilated through the Navy Atmospheric Variational Data Assimilation System (NAVDAS; Zhang et al., 2008), that became operational in 2010. Details on the aerosol model dynamics, emission and sink processes can be found in Lynch et al. (2016). NAAPS contains dust, sea salt, smoke, SO2, and other anthropogenic and biogenic fine particles, all of which are treated as externally mixed. The dust emission in NAAPS is based on Ginoux et al. (2001) erodibility map, with regional source tuning constrained by space-based and ground-based AOT observations (Lynch et al., 2016). While dust removal processes include dry deposition and wet removal, the dry deposition over ocean is adjusted based on assimilated AOT, and the wet deposition is constrained by satellite-based precipitation information retrieved from NOAA Climate Prediction Center MORPHing technique data (CMORPH; Joyce et al., 2004). NAAPS aerosol optical depth are available at 6 h temporal resolution, and 1º X 1º spatial resolution. For consistency with other reanalysis data, seasonal and climatological averages of AOT is also calculated for 2003 to 2012, using monthly averages. Reanalyzed NAAPS coarse and fine-mode AOT have good agreement with ground-based AOT from AERONET stations (Lynch et al., 2016). Similar to MERRA-2 reanalysis, NAAPS does not assimilate aerosol vertical information or speciation, hence the relative dust vertical profiles are uniformly varied, along with other aerosol species, to match the posterior AOT.

**3.3. JRAero**

The Japanese Reanalysis for Aerosol (JRAero) version 1.0 is produced by the Meteorological Research Institute (MRI) of the Japan Meteorological Agency. The global reanalysis product uses a global aerosol transport model named MASINGAR mk-2 (Model of Aerosol Species IN the Global AtmospheRe; Yukimoto et al., 2012), which consist an updated dust emission scheme (Yumimoto et al., 2017), when compared to the previous version of MASINGAR (Tanaka et al., 2003). MASINGAR mk-2 is coupled to an atmospheric general circulation model, also developed at MRI (Yoshimura and Yukimoto, 2008; Yukimoto et al., 2012), while the aerosol assimilation is done every 6 hours using a two-dimensional variational method (MASINGAR/2D-Var, similar to NAAPS-NAVDAS). Only the level 3 bias-corrected MODIS AOD, developed by the US Naval

Research Laboratory (NRL) and the University of North Dakota (Zhang & Reid, 2006 Hyer, et al., 2011; Shi et al., 2011), is assimilated into MASINGAR mk-2, and this data is largely unavailable over the deserts due to the stringent quality-control procedure (e.g. Yumimoto et al., 2017). Aerosol particles in the model are treated as externally mixed, with mineral dust carried in ten discrete particle bins (Yumimoto et al., 2017). The updated dust emission uses the wind erosion model developed by Shao et al. (1996), with erodibility factors for vegetation cover, snow cover, land-use type, and soil type (Tanaka and Chiba, 2005). Unlike MERRA-2 and NAAPS, both aerosol dry deposition and wet removal processes in MASINGAR mk-2 are model-dependent. Dry deposition in the model depends on the dry deposition velocity, which employs the resistance analog model (Seinfeld and Pandis, 2006), while the wet deposition process follows the parameterization of Giorgi & Chameides (1986) for in-cloud scavenging, and the procedure detailed in Tanaka & Chiba (2005) for below-cloud scavenging. JRAero is available for the period between 2011 and 2015, at 6 hours temporal resolution, and approximately 1.1º x 1.1º spatial resolution. We use the monthly averages of JRAero DAOD between 2011—2015 to construct the seasonal and climatological global DAOD values. Though the averaging period of 2011—2015, is different from other reanalysis product used, the spatial distribution of DAOD is largely consistent with other reanalysis products, albeit slightly smaller magnitude.

3.4. CAMSiRA

The Copernicus Atmosphere Monitoring Service (CAMS) interim Reanalysis (CAMSiRA) is a global reanalysis of atmospheric composition (Flemming et al., 2017). It uses a modified version of the European Centre for Medium-Range Weather Forecasts (ECMWF) Integrating Forecasting System for Composition (C-IFS) (Flemming et al., 2015). The aerosol model is based on the LMDZ model of Laboratoire de Météorologie Dynamique aerosol model (Reddy et al., 2005) that uses a bulk–bin scheme simulating desert dust, sea salt, organic carbon, black carbon, and sulfate aerosols (Morcrette et al., 2009). The wet and dry deposition are also modelled with different parameterizations. The wet deposition is based on Jacob et al. (2000) which account for sub-grid scale clouds and precipitation. Dry deposition is based on pre-calculated monthly mean deposition velocities following Wesley (1989). The C-IFS uses a four-dimensional variational (4D-VAR) data assimilation technique to combine satellite observations with chemistry-aerosol modelling. Aerosol optical depth is assimilated mainly from MODIS, with the variational bias correction

scheme developed at ECMWF (Inness et al., 2015). The mass mixing ratios of O3 and CO are also assimilated from various instruments as additional control variables. CAMSiRA is available for the period between 2011 and 2017, at 3 hours temporal resolution, and approximately 1.1°x1.1° spatial resolution. We use the monthly averages of CAMSiRA DAOD to construct the seasonal and climatological global DAOD values.

**4. Summary of measurements collected from literature and used for evaluation**

**4.1. D'Almeida (1987) – Ground Station – PSD only**

Aerosol particles are collected on microsorban-98 fiber filter, with size 20cm by 25 cm described in D'Almeida and Schutz (1983). This filter has a low flow resistance, and a high particle retention capacity. The filter is then dissolved in an organic liquid, such as xylene, to convert the dust particles into liquid suspensions. The resulting suspension is counted with scanning electron microscope (See the Fig. 1 in D'Almeida and Schutz (1983)). The procedure avoids charging effects on the sample surface, to guarantee unbiased magnification of the samples up to 30,000 times. The analysis was further corrected for collection efficiency of the filter. We use the average measurements that were taken over three sites between February-March 1979, and January-February 1982. These locations are: Matam (northeast Senegal) Timbuktu (Mali), and Agadez (Niger) and shown in Fig. 3 of D'Almeida (1987). Dust particles were measured for sizes larger than 100µm, but we use size distribution up to 20µm in this study. Since measurements are taken within the boundary layer, we select a representative height level between 0-100 m.

**4.2. Li et al., 1996 – Ground Station – MEE only**

Measurements are made over Barbados between 4 April to 3 May 1994 (main measurement period in April). Daily aerosol particles are collected using the Whatman-41 filter, and mineral dust components are determined by ashing the filter at 500 °C and weighing the residue. The resulting dust size distribution is mostly for particles of diameter $D \leq 10 \; \mu m$. Aerosol scattering is measured by nephelometer at 530 nm, and the resulting mass scattering efficiency is determined by linear regression method over the entire period of measurements.

**4.3.  Li et al., 2000 – Ground Station – MSE only**

Measurements were taken at a station on top of the Waliguan Mountain (3816 m atitude), in the Qinghai Province, China during October-November, 1997 and January 1998. Aerosol sizes up to diameter of D ≤ 18 μm were measured by a Micro Orifice Uniform Deposit Impactor used with Teflon filter. Measured $CaCO_3$ are assumed as proxy for dust particles, and consequently for dust volume distribution. Mass scattering efficiency is calculated using the Mie theory with density and index of refraction for $CaCO_3$ taken from Williams (1996). Values are reported at 550 nm wavelength (see their table 2).

**4.4.  Maring et al., 2000 – Ground Station – MSE only**

Dust properties are measured during July 1995 at the Global Atmospheric Watch station, located at Izana, Tenerife, Canary Island. Measurements took place at the station 2360 m above sea level, which is above the inversion level that is typically around 1200 m in summer. The dust size distribution is measured using a scanning mobility particle sizer and aerodynamic particle sizer, with diameter mostly up to about 10 μm (see their Fig. 7; it could also sample to 15 μm with stronger wind speed). Aerosol extinction was measured using nephelometer. The mass scattering efficiency is calculated using two methods, as the average for dusty and non-dusty periods: First, by calculating the linear regression between aerosol mass and its scattering (0.52 $m^2$ $g^{-1}$), and secondly by using Mie theory (0.48 $m^2$ $g^{-1}$). The values are reported for wavelength of 532 nm.

**4.5.  Andreae et al., 2002 – ARACHNE – MSE only**

Over a remote site in the Negev desert (Sde Boker, Israel), measurements of aerosol properties were conducted for a period of 2 years (Dec, 1995 –Oct, 1997) as part of Aerosol Radiation and Chemistry Experiment (ARACHNE) research program. For the entire period, light scattering was measured by nephelometer, but every week a 2-days and a 3-days samples are taken using a "Gent" PM10 stacked filter unit sampler to determine the concentration of the constituent species. The mass scattering efficiency is calculated as a multivariate linear regression of the light scattering coefficients on the coarse-mode, fine-mode, sulphate and dust concentrations. Dust mass scattering efficiency at 550 nm is thereafter obtained. For the value correction for non-Lambertian behavior and truncation errors of the nephelometer has been applied.

**4.6. Quinn et al., 2002 – INDOEX – MEE only**

As part of Indian Ocean Experiment (INDOEX) Intensive Field Phase (IFP), measurements of aerosol properties were made over the Arabia sea and the Indian Ocean on board the R/V Ronald H. Brown between February and March, 1999. The two-stage multi-jet cascade impactors (Berner et al., 1979) apportioned to differential mobility particle sizer and aerodynamic particle sizer are used for size distributions. From the elemental components (Al, Si, Ca, Fe, and Ti), dust is considered as inorganic oxidized material (IOM), and it is obtained by summing the oxides of the elements, in which each elemental mass concentration is multiplied by a molar correction factor (See their Equation 2). The mass extinction efficiency is calculated using Mie theory, and we use here values for particles with diameter $1.1 \leq D \leq 10$ μm, to avoid possible contamination by other aerosol in the sub-micron range (See their Fig. 10). Campaign-derived index of refraction is used. The values are reported for wavelength of 550 nm.

**4.7. Haywood et al., 2003 – SHADE – MEE only**

Dust particle measurements were taken during the Saharan Dust Experiment (SHADE) which took place between 19-28 September 2000 close to Sal, Cape Verde, off the coast of North Africa. The size distribution is determined using Passive Cavity Aerosol Spectrometer Probe 100X. The size distributions were not corrected for refractive index because they assumed that the refractive index of latex is approximately similar to that of dust. Due to instrument malfunction during the campaign, calculations of optical properties were largely limited to about 10μm. Mie theory is used to calculate the mass extinction efficiency at wavelength of 550 nm (see their table 2).

**4.8. Clarke et al. 2004 – ACE-Asia/TRACE-P – PSD and MSE**

Aerosol measurements were taken in the Sea of Japan (between Koran and Japan) in the spring (24 February to 10 April) of 2001, as part of the Asian Pacific Regional Aerosol Characterization Experiment (ACE-Asia) and NASA Transport and Chemical Evolution over the Pacific (TRACE-P). Similar instrumentations as the INDOEX campaign (Quinn et al., 2002) were used during ACE-Asia campaign. ACE-Asia used a laser optical particle counters (OPC) and condensation nuclei (CN) counters for aerosol size distribution. The OPC was operated at150°C and then at 300°C, to drive off low-temperature volatiles. In addition, light scattering of coarse and fine aerosol mode

was measured by two-wavelength TSI 563 nephelometers. Despite some differences in instrumentations in the ACE-Asia and TRACE-P, the authors show that measured aerosol sizing and optical properties agreed within instrument uncertainty at all altitudes. After the size distribution are normalized to emphasize the coarse dust (see their Fig. 5), we select the resulting reference size distribution as the representative size distribution. In addition, based on their Fig. 1, we choose the representative height level between surface and 6km. The mass scattering efficiency is calculated using the Mie theory. The wavelength is at 550 nm.

**4.9. Otto et al., 2007 – ACE-2 – PSD only**

Aerosol measurements were taken during Aerosol Characterisation Experiment (ACE-2) conducted about 50—200km off the coast of Northern Africa close to Canary Islands on 8[th] of July, 1997. The aerosol size distributions used data from five instruments, including Condensation Particle Counter (CPC), Differential Mobility Analyser (DMA), Optical Particle Counter (OPC), and Forward Scattering Spectrometer Probe (FSSP). Together, the instruments measured particles up to diameter of ~31μm (see their Table 1). We use reported size distribution, up to 20μm at four specific levels – 2700 m, 4000 m, 5500 m, 7000 m.

**4.10. Chou et al., 2008 & Osborne et al. 2008 – AMMA/DABEX – PSD and MEE**

Based in Niamey, Niger, aerosol measurements were made between 13 January and 3 February, 2006 over the West Africa Sahel region, as part of the Dust and Biomass-burning Experiment (DABEX), affiliated with the African Monsoon Multidisciplinary Analysis (AMMA). On board the UK BAe-146 research aircraft , aerosol size distribution are measured using the Passive Cavity Aerosol Spectrometer Probe 100-X (PCASP) with additional counter-flow virtual impactor (CVI) inlet to measure particles up to diameter of 10 μm. Because of the aerosol inlet configuration on the aircraft, the measurement of coarse dust were particularly problematic. Both groups of authors reported size distributions measured from 2 flights numbered B160 and B165 out of 14 flights. Dust size distribution is taken from Chou et al, 2008, while the mass extinction efficiency is taken from Osborne et al., 2008. Since most of the flight are below ~1500m above ground level, we select 0-1500m as the representative height level. The mass extinction efficiency is calculated

using Mie theory, at 550 nm wavelength. The mass extinction efficiency is calculated with log-normal fit to the measured dust size distribution, with assumed dust density of 2.65 g cm$^{-3}$ (see Table 4 in Osborne et al, 2008).

**4.11. McConnell et al., 2008 – DODO-1/ DODO-2 – PSD and MEE**

Based at Dakar, Senegal, measurements of dust properties are conducted as part of the Dust Outflow and Deposition to the Ocean project (DODO) off the coast of North Africa. The project occurred on two phases: One between 7 to 16 February 2006, called DODO-1, and the other between 22 to 28 August, 2006, called DODO-2. During DODO, a combination of wing-mounted Passive Cavity Aerosol Spectrometer Probe (PCASP), Droplet Measurement Technology cloud droplet probe (CDP-100), and bulk filters are used to measure dust size distribution up to diameter of 40 µm. We use the DODO-2 size distribution in this study based on their Fig. 7. Because the height level is given around 1 km altitude for the size distribution (see caption of Fig. 7), we limit our representative height level between 0-1 km. The mass extinction efficiency is calculated with Mie code, using the measured size distribution. Because coarse dust particles are not collected during DODO-1, we use the MEE value reported in Osborne et al., 2008 (see their Table 4) that include coarse dust collected during DABEX.

**4.12. Weinzierl et al., 2009 – SAMUM-1 – PSD only**

Based in Casablanca, Morocco, in situ dust particle size distribution measurements were taken onboard the German Center for Aviation and Space Flight (DLR) Falcon as part of the Saharan Mineral Dust Experiment (SAMUM-1) in Southern Morocco in May and June 2006. Three dust events were observed during the campaign on 16 to 22 May, 24 to 28 May, and 31 May to 5 June. We use the size distribution measured from a wing-mounted Forward Scattering Spectrometer Probe (FSSP) 300, and the composite size distribution from three Condensation Particle Counters (CPCs) heated with a thermal denuder (TD) at 250°C and a Grimm OPC (Optical Particle Counter). The FSSP-300 measured particles with diameters between 0.3 and 30 µm. The three CPCs measured non-volatile particles in nucleation, Aitken, and accumulation mode, respectively. With the Grimm OPC, non-volatile size distribution was derived for particles smaller than 2.5 µm.

Data are taken from their Fig. 8 which represents the composite size distribution for L02 on flight #060519a  and L07 on flight #060604a. For these flights, are respectively 4853m

 and 3703m above sea level, and therefore approximated to 3700-4900m in our study.

**4.13.   Wagner et al., 2009 – DARPO – PSD only**

Based in Casablanca, Morocco, in situ measurements were performed in May 2006 over Portugal as part of the Desert Aerosols over Portugal (DAPRO) project affiliated with SAMUM (see section 4.12), using essentially the same instrumentation and derivation as Weinzierl et al. (2009) with an additional high spectral resolution lidar. Measurements were conducted at 2300m and 3245m during a flight onboard Falcon aircraft over Évora on 27 May 2006, and size distribution data between 0.01 and 35 µm were presented (see their Fig. 9). Size distribution at the two different heights were very similar. In this study, we take the representative height range between 2300-5000m.

**4.14.   Kandler et al., 2009 – SAMUM-1 – PSD only**

During the 2006 SAMUM campaign, Kandler et al. conducted size distribution measurements by collecting dust samples at a ground station in Tinfou, Morocco where dust events occur often during summer. However, anthropogenic emissions still exerted a significant impact on particles smaller than 500 nm, despite the remote location of the ground station. Mineral dust dominated particles beyond 500 nm. Employing a combination Differential Mobility Particle Sizer (DMPS), Aerodynamic Particle Sizer (APS) and single-stage impactor (SSI), the authors measured and reported a size distribution under a higher concentration condition, named dust wind condition. The number distribution of particles larger than 500 nm varied by more than one order of magnitude, largely correlated to meteorological conditions. For particles larger than 10 µm, the variation was about three orders of magnitude. Since the station is at an elevation of approximately 684 m above sea level and the inlet of the sampling device ~4 m above ground level, we choose our representative height to be between 0-700m.

**4.15.   Kandler et al., 2011 – SAMUM-2 – PSD only**

A part of the SAMUM-2 campaign which aims to study more aged dust as opposed to fresh dust in SAMUM-1, the effort of Kandler et al. (2011) used the similar instrumentation as SAMUM-1 (Kandler et al., 2009) to measure dust size distribution at a ground station on Praia, Cape Verde in winter 2008. A notably higher concentration of clay minerals was found compared to SAMUM-1, as expected for aged dust. Size distributions from three dust phases were reported. As in SAMUM-1, it was found that wind speed had a significant impact on the distribution between 400 nm and 10 μm, and this strength of this impact increases rapidly beyond 10 μm. The presence of larger particles is highly correlated with mass concentration. Similar to Kandler et al., 2009, because the elevation of the station is approximately 100 m above sea level, we place our representative height levels between 0-110m.

**4.16. Ryder et al., 2013a and Ryder et al., 2013b – Fennec 2011 – PSD and MEE**

Both Ryder et al studies measured dust properties over and near western side of the North African desert on board the UK's BAe-146-301 Research Aircraft during the Fennec June 2011 campaign. While the Ryder et al., (2013a) study reported dust properties near the Canary Islands, the Ryder et al., (2013b) study reported dust properties farther inland over Mauritania and Mali. Below, we give brief description of the instruments used and measurements taken. For more details, please refer to their studies.

(Ryder et al., 2013b): Although 16 dedicated flights was conducted over Mauritania and Mali during the campaign, only 11 of those with consistent instrumentation were used. A suite of instruments is used to measure dust size distribution (see table 3 in Ryder et al., 2013b), namely wing-mounted Passive Cavity Aerosol Spectrometer Probe 100X (PCASP), Cloud Droplet Probe (CDP), and Cloud Imaging Probe (CIP). The measurement covers significant coarse-mode size range of dust particles, and were corrected for a refractive index appropriate for dust and for instrumental drift during the campaign. Details of the calibration and correction performed on each instrument can be found in (Ryder et al., 2013b). Since most of the measurements taken are below 2-3 km (see their Fig. 2), therefore in this study we use a representative height between 0-3 km. We use data taken from their Fig. 5b, which include the mean size distribution obtained using the PCASP, CDP and CIP.

(Ryder et al., 2013a): Although dust properties are taken near the Canary Islands, in-situ measurements reported in this study uses similar instrumentations as in (Ryder et al., 2013b). Because take-off and landing profiles observations, vertical distribution of the dust size distribution can be made. We obtain these dataset directly from the authors, and present size distribution at four levels – 2500, 4000, 5500, and 6000 m. As reported in the study, we used here the averaged MEE value of 0.31+/-0.08 between the calculate values for aged dust (0.23) and the SAL categories (0.39). Mie scattering code is used to calculate the mass scattering efficiency at wavelength of 550 nm.

**4.17. Jung et al., 2013 – BACEX - PSD only**

In situ measurements of aged dust size distribution was conducted onboard Center for Interdisciplinary Remotely Piloted Aircraft Studies (CIRPAS) Twin Otter research aircraft under the Barbados Aerosol Cloud Experiment (BACEX) in on 1 and 2 April 2010. Size distribution measured from Passive Cavity Aerosol Spectrometer Probe (PCASP) and the forward scattering section of a Cloud and Aerosol Spectrometer (CASF) covered particle diameters from 0.1 to 54 µm. Data taken while the aircraft was in clouds were excluded because PCASP is known to have low accuracy inside clouds. We use one measurement on 1 April within the Sahara air layer (SAL) at 2726m, and one on 2 April in the intermediate layer at 1289m. For comparison, the representative height is placed between 1250-2700m. The Mass extinction efficiency is calculated using Mie theory, at 550 nm wavelength.

**4.18. Weinzierl et al., 2017 – SALTRACE - PSD only**

Based in Barbados, Puerto Rico, and Cabo Verde, in situ aerosol size distribution measurements were conducted as part of the Saharan Aerosol Long-Range Transport and Aerosol–Cloud-Interaction Experiment (SALTRACE) in June 2013. The same air mass was first sampled over Cabo Verde at the altitude of 2.6km on 17 June 2013, and again over Barbados at 2.3km on 22 June 2013. Total number distribution below 1 µm was inverted from measurements from three Condensation Particle Counters (CPCs) between 0.005 and 2.5 µm, a Grimm Sky Optical Particle Counter (OPC) between 0.25 and 2.5 µm, and a wing-mounted Ultra-High Sensitivity Aerosol

Spectrometer Airborne (UHSAS-A) between 0.06 and 1 μm. Total number distribution above 1 μm was measured with Cloud and Aerosol Spectrometer with Depolarization (CAS-DPOL). Distribution in the full size range was parametrized with four lognormal distributions. The authors expected 20-μm particles to be removed after 3 days of transport, but 20% of the observed 20-μm particles in Cabo Verde survived in the second measurement above Barbados.

**4.19.  Ryder et al., 2018 - AER-D - PSD and MEE**

In-situ measurements were taken in August, 2015 close to Cape Verde, off the coast of Northern Africa properties during the beginning of trans-Atlantic transport of dust particles. These measurements were part of the AERosol Properties – Dust (AER-D) fieldwork campaign, which ran alongside the Ice in Clouds Experiment – Dust (ICE-D) project, and similarly used the UK's BAe-146-301 Research Aircraft. In addition to the instruments used during Fennec 2011 campaign, the AER-D campaign used cloud imaging probes (CIP15 and 2DS) for size distributions at d>10 microns. They use wing-mounted optical particle counters and shadow probes to measure dust sizes between 0.1 and 100 μm diameter, a nephelometer and an absorption photometer to measure dust optical properties, and an in-cabin filter collection system to collect dust samples. Data for size distribution was obtained directly from the authors. However, reported value of dust mass extinction efficiency, calculated using Mie code at 550 nm wavelength, was obtained from their paper.

---

## Author Response (AR2)

Comment from co-editor

The first part of paragraph 2.3.1 still needs to be improved.
Please concentrate you efforts on lines 29 to 35 of page 7 and lines 1 through 11 of page 8 to make this part easy to understand for the reader.
As an example, the following sentence is not grammatically correct and I am not sure to be able to make sense of it:
"This correction factor (alpha ) is estimated as the ratio of the K17 constraint on the fractional contribution of the particle bin to the simulated fractional contribution of the particle bin per unit global dust loading.'

Thank you for the comment.
We rewrote some parts of this section for clarity. Specifically, we now have equation 2 immediately after it is defined. To make it clearer, we wrote:

"For each model, we force the simulated globally-averaged dust size distribution to match the K17 constraint on the globally-averaged size distribution (see supplement Fig. S-1), such that:"

As a result, we move the description of the correction factor after the equation. To also make the description of the correction factor clearer, we rewrote this part to include the following sentences:

". In Eqn. 2, we multiply each simulated dust size distribution $\tilde{f}_{k,i}(x, y, z, D_{k,i})$ by a correction factor $\alpha_{k,i}$, that is estimated as the ratio of the fractional contributions of the K17 globally-averaged constraint to that obtained from the same model. This correction is done for each bin $i$ of each model $k$, defined between $D_{k,i-}$ and $D_{k,i+}$."

[revised manuscript text omitted]
})$ by a correction factor $\alpha_{k,i}$, that is estimated as the ratio of the fractional contributions of the K17 globally-averaged constraint to that obtained from the same model. This correction is done for each bin $i$ of each model $k$, defined between $D_{k,i-}$ and $D_{k,i+}$. The resulting corrected spatially-varying dust size distribution, $\hat{f}_{k,i}(x,y,z,D_{k,i})$ is normalized such that the discrete sum over each location and height equals unity, that is: $\sum_{i=1_k}^{N_k} \hat{f}_{k,i}(x,y,z,D_{k,i}) = 1$.

Each model simulation in the ensemble has a particle size range and spacing that differs from other models (see Table 1 and section 2.1 for details). In order to combine the corrected size distributions from the different models into a single estimate, and to quantify the uncertainty across the different models, each corrected size distribution must be in a consistent size range and spacing with other models. We therefore process the corrected size distributions over a given location as follows: (1) we correct and scale each model's lower and upper diameter limits to the common diameter range of 0.2 – 20 µm (see section 2.3.1.1); and (2) we estimate the sub-bin distribution for each model's bias-corrected size distribution by fitting a generalized analytical function, extending the Kok et al, (2017) theoretical expression of dust size distribution to the 3-D dataset (see section 2.3.1.2).

**2.3.1.1   Correcting model simulations to a common diameter range**

For all simulations in the model ensemble, we set the lower and upper diameter limits to common limits defined by $D_{\min} = 0.2\ \mu m$ and $D_{\max} = 20\ \mu m$, respectively. The lower diameter limit ($D_{\min}$) is based on the lowest common diameter included in all the model simulations used in our analysis (Table 1). In addition, possible contaminations by other aerosol species are significantly more likely below 0.2 $\mu m$ in measurements of dust aerosol particles (e.g. Dubovik et al., 2000). For these reasons, we set the lower diameter limit to $D_{\min} = 0.2 \mu m$, consistent with previous studies (e.g. Mahowald et al., 2014, Kok et al., 2017). Further, we set the upper diameter limit to $D_{\max} = 20\ \mu m$, because most global models generally do not incorporate dust particles beyond 20 $\mu m$ and also because the observational constraints on the size distribution from Kok et al. (2017) is limited to this maximum diameter. Although advances in airborne observations in recent years have led to measurements of larger dust particles with $D > D_{max}$ in the atmosphere which has shown that the contribution of $D > 20\ \mu m$ to shortwave and longwave extinctions are non-negligible (e.g. Ryder et al., 2013b, 2019; Weinzierl et al., 2009, 2017), there is still a scarcity of these measurements, such that an observational constraint on dust particles with $D > D_{max}$ would be very uncertain (e.g. Mahowald et al., 2014).

To correct each model simulation to the common diameter range of $[D_{\min}, D_{\max}]$, we first create a new particle bin for the lower and/or upper diameter limit, and then we use the K17 constraints on the globally-averaged size distribution (Eqn. 1) to estimate the equivalent fraction of dust mass in that bin. This dust mass fraction is estimated in a way that is consistent with the size distribution obtained earlier from Eqn. 2. Specifically, for simulations with a lower diameter limit ($D_{k,1_k-}$) less than $D_{\min}$, we estimate the equivalent dust mass fraction for the bin between $D_{\min}$ and $D_{k,1_k+}$ (where $D_{k,1_k+}$ is the upper diameter limit of bin 1; such that $D_{k,1_k+} > D_{\min}$ ) by scaling the mass in the nearest bin with a factor that depends on the globally-averaged size distribution. For instance, the first particle bin of the CESM model (Table 1) has a range of $\left[D_{k,1_k-}, D_{k,1_k+}\right] = 0.1 - 1.0$ μm, such that we create a new particle bin defined by $\left[D_{\min}, D_{k,1_k+}\right] = 0.2 - 1.0$ μm, and estimate the equivalent dust mass fraction in that new bin. For all model simulations, we can denote this procedure mathematically as:

$$\hat{f}_k\left(x, y, z, \left[D_{\min}, D_{k,1_k+}\right]\right) = \hat{f}_k\left(x, y, z, \left[D_{k,1_k-}, D_{k,1_k+}\right]\right) \cdot \delta_{D_{\min}} \tag{3}$$

[revised manuscript text omitted]